# Complementary Benefits of Contrastive Learning and Self-Training Under Distribution Shift

**Saurabh Garg***
Carnegie Mellon University
sgarg2@andrew.cmu.edu

**Amrith Setlur***
Carnegie Mellon University
asetlur@andrew.cmu.edu

**Zachary C. Lipton**
Carnegie Mellon University
zlipton@andrew.cmu.edu

**Sivaraman Balakrishnan**
Carnegie Mellon University
sbalakri@andrew.cmu.edu

**Virginia Smith**
Carnegie Mellon University
smithv@andrew.cmu.edu

**Aditi Raghunathan**
Carnegie Mellon University
aditirag@andrew.cmu.edu

## Abstract

Self-training and contrastive learning have emerged as leading techniques for incorporating unlabeled data, both under distribution shift (unsupervised domain adaptation) and when it is absent (semi-supervised learning). However, despite the popularity and compatibility of these techniques, their efficacy in combination remains surprisingly unexplored. In this paper, we first undertake a systematic empirical investigation of this combination, finding (i) that in domain adaptation settings, self-training and contrastive learning offer significant complementary gains; and (ii) that in semi-supervised learning settings, surprisingly, the benefits are not synergistic. Across eight distribution shift datasets (*e.g.*, BREEDs, WILDS), we demonstrate that the combined method obtains 3–8% higher accuracy than either approach independently. Finally, we theoretically analyze these techniques in a simplified model of distribution shift demonstrating scenarios under which the features produced by contrastive learning can yield a good initialization for self-training to further amplify gains and achieve optimal performance, even when either method alone would fail.

## 1 Introduction

Even under natural, non-adversarial distribution shifts, the performance of machine learning models often drops [65, 83, 47, 29]. While we might hope to retrain our models on labeled samples from the new distribution, this option is often unavailable due to the expense or impracticality of collecting new labels. Consequently, researchers have investigated the Unsupervised Domain Adaptation (UDA) setting. Here, given labeled source data and unlabeled out-of-distribution (OOD) target data, the goal is to produce a classifier that performs well on the target. Because UDA is generally underspecified [7], researchers have focused on two main paths: (i) domain adaptation papers that explore heuristics for incorporating the unlabeled target data, relying on benchmark datasets ostensibly representative of "real-world shifts" to adjudicate progress [72, 85, 47]; and (ii) theoretically motivated papers that explore structural assumptions under which UDA problems are well posed [77, 74]. This work engages with the former focusing on two popular methods: self-training and contrastive pretraining.

Self-training [75, 52, 79, 90, 86] and contrastive pretraining [13, 16, 93] were both proposed, initially, for traditional Semi-Supervised Learning (SSL) problems, where the labeled and unlabeled data are drawn from the same distribution. Here the central challenge is statistical: to exploit the unlabeled data to learn a better predictor than one would get by training on the (small) labeled data alone. More

---

*Equal contribution.

37th Conference on Neural Information Processing Systems (NeurIPS 2023).

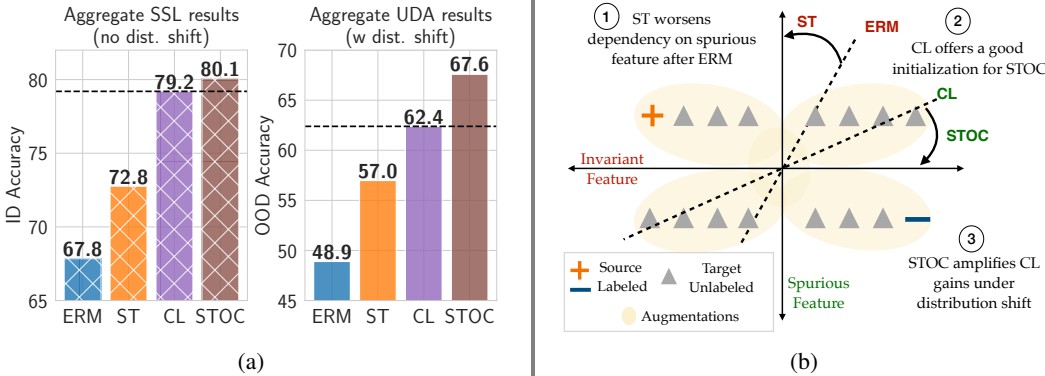

(a)                                                  (b)

Figure 1: *Self-training over Contrastive learning (STOC) improves over Contrastive Learning (CL) under distribution shift.* **(a)** We observe that in SSL settings, where labeled and unlabeled data are drawn from the same distribution, STOC offers negligible improvements over CL. In contrast, in UDA settings where there is distribution shift between labeled and unlabeled data, STOC offers gains over CL. Results aggregated across 8 benchmarks. Results on individual data in Table 1 and 2. **(b)** 2-D illustration of our simplified distribution setup, depicting decision boundaries learned by ERM and CL and how Self-Training (ST) updates those. ①, ②, and ③ summarize our theoretical results in Sec. 4.

recently, these methods have emerged as favored empirical approaches for UDA, demonstrating efficacy on many popular benchmarks [70, 30, 12, 76]. In self-training, one first learns a predictor using source labeled data. The predictor then produces pseudolabels for the unlabeled target data, and a new predictor is trained on the pseudolabeled data. Contrastive pretraining learns representations from unlabeled data by enforcing invariance to specified augmentations. These representations are subsequently used to learn a classifier. In UDA, the representations are trained on the union of the source and target data. In attempts to explain their strong empirical performance, several researchers have attempted analyses under various assumptions on the data, task, and inductive biases of the function class[87, 39, 73, 76, 12, 40, 38, 11]. Despite the strong strong results, there has been surprisingly little work (both empirically and theoretically) exploring when either might be expected to perform best and whether the benefits might be complementary.

In this paper, we investigate the complementary benefits of self-training and contrastive pretraining. Interestingly, we find that the combination yields significant gains in UDA despite producing negligible gains in SSL. In experiments across eight distribution shift benchmarks (*e.g.* BREEDs [72], FMoW [47], Visda [63]), we observe that re-using unlabeled data for self-training (with Fix-Match [79]) after learning contrastive representations (with SwAV [13]), yields $> 5\%$ average improvement on OOD accuracy in UDA as compared to $< 0.8\%$ average improvement in SSL (Fig. 1).

Next, we address the question *why the combination of self-training and contrastive learning* proves synergistic in distribution shift scenarios. To facilitate our analysis, we consider a simplified distribution shift setting that includes two types of features: (i) invariant features that perfectly predict the label; and (ii) domain-dependent features that are predictive of the label in just source. Our theoretical analysis reveals that self-training can achieve optimal target performance but requires a "good" enough classifier to start with. We observe that source-only ERM fails to provide a "good" initialization. On the other hand, contrastive pretraining on unlabeled data performs better than ERM but is still sub-optimal. This implies that contrastive pretraining ends up decreasing reliance on domain-dependent features (as compared to ERM) but doesn't completely eliminate them. Nevertheless, contrastive pretraining does provide a "good" initialization for self-training, *i.e.*, "good" initial pseudolabels on the target unlabeled data. As a result, self-training on top of contrastive learned features effectively unlearns the reliance on domain-dependent features and generalizes perfectly OOD. In contrast, for SSL settings (*i.e.*, in distribution), our analysis highlights that contrastive pretraining already acquires sufficient predictive features such that linear probing with (a small amount of) labeled data picks up those features and attains near-optimal ID generalization.

Finally, we connect our theoretical understanding of "good" representations from contrastive learning and improved linear transferability from self-training back to observed empirical gains. We linearly probe representations (fix representations and train only the linear head) learned by contrastive pretraining vs. no pretraining and find: (i) contrastive pretraining substantially improves the ceiling on

the target accuracy (performance of optimal linear probe) compared to ERM; (ii) self-training mainly improves linear transfer, *i.e.* OOD performance for the linear probe trained with source labeled data.

## 2 Setup and Preliminaries

**Task.** Our goal is to learn a predictor that maps inputs $x \in \mathcal{X} \subseteq \mathbb{R}^d$ to outputs $y \in \mathcal{Y}$. We parameterize predictors $f = h \circ \Phi : \mathbb{R}^d \mapsto \mathcal{Y}$, where $\Phi : \mathbb{R}^d \mapsto \mathbb{R}^k$ is a feature map and $h \in \mathbb{R}^k$ is a classifier that maps the representation to the final scores or logits. Let $P_S, P_T$ be the source and target joint probability measures over $\mathcal{X} \times \mathcal{Y}$ with $p_S$ and $p_T$ as the corresponding probability density (or mass) functions. The distribution over unlabeled samples from both the union of source and target is denoted as $P_U = (1/2) \cdot P_S(x) + (1/2) \cdot P_T(x)$.

We study two particular scenarios: (i) Unsupervised Domain Adaptation (UDA); and (ii) Semi-Supervised Learning (SSL). In UDA, we assume that the source and target distributions have the same label marginals $P_S(y) = P_T(y)$ (*i.e.*, no label proportion shift) and the same Bayes optimal predictor, *i.e.*, $\arg\max_y p_S(y \mid x) = \arg\max_y p_T(y \mid x)$. We are given labeled samples from the source, and unlabeled pool from the target. In contrast in SSL, there is no distribution shift, *i.e.*, $P_S = P_T = P_U$. Here, we are given a small number of labeled examples and a comparatively large amount of unlabeled examples, both drawn from the same distribution, which we denote as $P_T$.

Unlabeled data is typically much cheaper to obtain, and our goal in both these settings is to leverage this along with labeled data to achieve good performance on the target distribution. In the UDA scenario, the challenge lies in generalizing out-of-distribution, while in SSL, the challenge is to generalize in-distribution despite the paucity of labeled examples. A predictor $f$ is evaluated on distribution P via its accuracy, *i.e.*, $A(f, P) = \mathbb{E}_P(\arg\max f(x) = y)$.

**Methods.** We now introduce the algorithms used for learning from labeled and unlabeled data.

1. *Source-only ERM (ERM)*: A standard approach is to simply perform supervised learning on the labeled data by minimizing the empirical risk $\sum_{i=1}^n \ell(h \circ \Phi(x), y)$, for some classification loss $\ell : \mathbb{R} \times \mathcal{Y} \mapsto \mathbb{R}$ (*e.g.*, softmax cross-entropy) and labeled points $\{(x_i, y_i)\}_{i=1}^n$.

2. *Contrastive Learning (CL)*: We first use the unlabeled data to learn a feature extractor. In particular, the objective is to learn a feature extractor $\Phi_{cl}$ that maps augmentations (for e.g. crops or rotations) of the same input close to each other and far from augmentations of random other inputs [13, 16, 93]. Once we have $\Phi_{cl}$, we learn a linear classifier $h$ on top to minimize a classification loss on the labeled source data. We could either keep $\Phi_{cl}$ fixed or propagate gradients through.

   When clear from context, we also use CL to refer to just the contrastively pretrained backbone without training for downstream classification.

3. *Self-training (ST)*: This is a two-stage procedure, where the first stage performs source-only ERM by just looking at source-labeled data. In the second stage, we iteratively apply the current classifier on the unlabeled data to generate "pseudo-labels" and then update the classifier by minimizing a classification loss on the pseudolabeled data [52].

## 3 Self-Training Improves Contrastive Pretraining Under Distribution Shift

**Self-Training Over Contrastive learning (STOC).** Finally, rather than starting with a source-only ERM classifier, we propose to initialize self-training with a CL classifier, that was pretrained on unlabeled source and target data. ST uses that same unlabeled data again for pseudolabeling. As we demonstrate experimentally and theoretically, this combination of methods improves substantially over each independently.

**Datasets.** For both UDA and SSL, we conduct experiments across eight benchmark datasets: four BREEDs datasets [72]—Entity13, Entity30, Nonliving26, Living17; FMoW [47, 18] from WILDS benchmark; Officehome [85]; Visda [64, 63]; and CIFAR-10 [48]. Each of these datasets consists of domains, enabling us to construct source-target pairs (e.g., CIFAR10, we consider CIFAR10→CINIC shift [22]). In the UDA setup, we adopt the source and target domains standard to previous studies (details in App. C.2). Because the SSL setting lacks distribution shift, we do not need to worry about domain designations and default to using source alone. To simulate limited supervision in SSL, we sub-sample the original labeled training set to 10%.

Table 1: *Results in the UDA setup*. We report accuracy on target (OOD) data from which we only observe unlabeled examples during training. For benchmarks with multiple target distributions (*e.g.*, OH, Visda), we report avg accuracy on those targets. Results with source performance, individual target performance, and standard deviation numbers are in App. C.4.

| Method | Living17 | Nonliv26 | Entity13 | Entity30 | FMoW (2 tgts) | Visda (2 tgts) | OH (3 tgts) | CIFAR→ CINIC | Avg |
|---|---|---|---|---|---|---|---|---|---|
| ERM | 60.31 | 45.54 | 68.32 | 55.75 | 56.50 | 20.91 | 9.51 | 74.33 | 48.90 |
| ST | 71.29 | 56.79 | 77.93 | 66.37 | 56.79 | 38.03 | 10.47 | 78.19 | 56.98 |
| CL | 74.14 | 57.02 | 76.58 | 66.01 | 61.78 | 63.49 | 22.63 | 77.51 | 62.39 |
| STOC (ours) | **82.22** | **62.23** | **81.84** | **72.00** | **65.25** | **70.08** | **27.12** | **79.94** | **67.59** |

Table 2: *Results in the SSL setup*. We report accuracy on hold-out ID data. Recall that SSL uses labeled and unlabeled data from the same distribution during training. Refer to App. C.5 for ERM and ST.

| Method | Living17 | Nonliv26 | Entity13 | Entity30 | FMoW | Visda | OH | CIFAR | Avg |
|---|---|---|---|---|---|---|---|---|---|
| CL | 91.15 | 84.58 | 90.73 | 85.47 | 43.05 | 97.67 | 49.73 | 91.78 | 79.27 |
| STOC (ours) | 92.00 | 85.95 | 91.27 | 86.14 | 44.43 | 97.70 | 49.95 | 93.06 | 80.06 |

**Experimental Setup and Protocols.** SwAV [13] is the specific algorithm that we use for contrastive pretraining. In all UDA settings, unless otherwise specified, we pool all the (unlabeled) data from the source and target to perform SwAV. For self-training, we apply FixMatch [79], where the loss on source labeled data and on pseudolabeled target data are minimized simultaneously. For both methods, we fix the algorithm-specific hyperparameters to the original recommendations. For SSL settings, we perform SwAV and FixMatch on in-distribution unlabeled data. We experiment with Resnet18, Resnet50 [42] trained from scratch (*i.e.* random initialization). We do not consider off-the-shelf pretrained models (*e.g.*, on Imagenet [68]) to avoid confounding our conclusions about contrastive pretraining. However, we note that our results on most datasets tend to be comparable to and sometimes exceed those obtained with ImageNet-pretrained models. For source-only ERM, as with other methods (FixMatch, SwAV), we default to using strong augmentation techniques: random horizontal flips, random crops, augmentation with Cutout [24], and RandAugment [21]. Moreover, unless otherwise specified, we default to full finetuning with source-only ERM, both from scratch and after contrastive pretraining, and for ST with FixMatch. For UDA, given that the setup precludes access to labeled data from the target distribution, we use source hold-out performance to pick the best hyperparameters. During pretraining, early stopping is done according to lower values of pretraining loss. For more details on datasets, model architectures, and experimental protocols, see App. C.

**Results on UDA setup.** Both ST and CL individually improve over ERM across all datasets, with CL significantly performing better than ST on 5 out of 8 benchmarks (see Table 1). Even on datasets where ST is better than CL, their performance remains close. Combining ST and CL with STOC shows an 3–8% improvement over the best alternative, yielding an absolute improvement in average accuracy of 5.2%.

Note that by default, we train with CL on the combined unlabeled data from source and target. However, to better understand the significance of unlabeled target data in contrastive pretraining, we perform an ablation where the CL model was trained solely on unlabeled source data (refer to this as CL (source only); see App. C.4). We observe that ST on top of CL (source only) improves over ST (from scratch). However, the average performance of ST over CL (source only) is similar to that of standalone CL, maintaining an approximate 6% performance gap observed between CL and ST. This brings two key insights to the fore: (i) the observed benefit is not merely a result of the contrastive pretraining objective alone, but specifically CL with unlabeled target data helps; and (ii) both CL and ST leverage using target unlabeled data in a complementary nature.

**Results on SSL setup.** While CL improves over ST (as in UDA), unlike UDA, STOC doesn't offer any significant improvements over CL (see Table 2; ERM and ST results (refer to App. C.5). We conduct ablation studies with varying proportions of labeled data used for SSL, illustrating that there's considerable potential for improvement (see App. C.5). These findings highlight that the complementary nature of STOC over CL and ST individually is an artifact of distribution shift.

# 4 Theoretical Analysis and Intuitions

Our results on real-world datasets suggest that although self-training may offer little to no improvement over contrastive pretraining for in-distribution (*i.e.*, SSL) settings, it leads to substantial improvements when facing distribution shifts in UDA (Sec. 3). Why do these methods offer complementary gains, but only under distribution shifts? In this section, we seek to answer this question by first replicating all the empirical trends of interest in a simple data distribution with an intuitive story (Sec. 4.1). In this toy model, we formally characterize the gains afforded by contrastive pretraining and self-training both individually (Secs. 4.2, 4.3) and when used together (Sec. 4.4).

**Data distribution** We consider binary classification and model the inputs as consisting of two kinds of features: $x = [x_{in}, x_{sp}]$, where $x_{in} \in \mathbb{R}^{d_{in}}$ is the invariant feature that is predictive of the label across both source $P_S$ and target $P_T$ and $x_{sp} \in \mathbb{R}^{d_{sp}}$ is the spurious feature that is correlated with the label $y$ only on the source domain $P_S$ but uncorrelated with label $y$ in $P_T$. Formally, we sample $y \sim \text{Unif}\{-1, 1\}$ and generate inputs $x$ conditioned on y as follows:

$$P_S: \ x_{in} \sim \mathcal{N}(\gamma \cdot yw^\star, \Sigma_{in}) \ \ x_{sp} = y\mathbf{1}_{d_{sp}}$$
$$P_T: \ x_{in} \sim \mathcal{N}(\gamma \cdot yw^\star, \Sigma_{in}) \ \ x_{sp} \sim \mathcal{N}(\mathbf{0}, \Sigma_{sp}), \tag{1}$$

where $\gamma$ is the margin afforded by the invariant feature[2]. We set the covariance of the invariant features $\Sigma_{in} = \sigma_{in}^2 \cdot (\mathbf{I}_{d_{in}} - w^\star w^{\star\top})$. This makes the variance along the latent predictive direction $w^\star$ to be zero. Note that the spurious feature is also completely predictive of the label in the source data. In fact, when $d_{sp}$ is sufficiently large, $x_{sp}$ is more predictive (than $x_{in}$) of y in the source. In the target, $x_{sp}$ is distributed as a Gaussian with $\Sigma_{sp} = \sigma_{sp}^2 \mathbf{I}_{d_{sp}}$. We use $w_{in} = [w^\star, 0, ..., 0]^\top$ to refer to the invariant direction/feature, and $w_{sp} = [0, ..., 0, \mathbf{1}_{d_{sp}}/\sqrt{d_{sp}}]^\top$ for the spurious direction.

**Data for UDA vs. SSL** For convenience, we assume access to infinite unlabeled data and replace their empirical quantities with population counterparts. For SSL, we sample both finite labeled and infinite unlabeled data from the same distribution $P_T$, where spurious features are absent (to exclude easy-to-generalize features). For UDA, we assume infinite labeled data from $P_S$ and infinite unlabeled from $P_T$. Importantly, note that due to distribution shift, population access of $P_S$ doesn't trivialize the problem as "ERM" on infinite labeled source data *does not* achieve optimal performance on target.

**Methods and objectives** Recall from Section 2 that we learn linear classifiers $h$ over feature extractor $\Phi$. For our toy setup, we consider linear feature extractors i.e. $\Phi$ is a matrix in $\mathbb{R}^{d \times k}$ and the prediction is given by $\text{sgn}(h^\top \Phi x)$. We use the exponential loss $\ell(f(x), y) = \exp(-yf(x))$.

*Self-training.* ST performs ERM in the first stage using labeled data from the source, and then subsequently updates the head $h$ by iteratively generating pseudolabels on the unlabeled target:

$$\mathcal{L}_{st}(h; \Phi) \ := \ \mathbb{E}_{P_T(x)} \ell(h^\top \Phi x, \text{sgn}(h^\top \Phi(x)))$$
$$\text{Update: } h^{t+1} \ = \ \frac{h^t - \eta \nabla_h \mathcal{L}_{st}(h^t; \Phi)}{\|h^t - \eta \nabla_h \mathcal{L}_{st}(h^t; \Phi)\|_2} \tag{2}$$

For ERM and ST, we train both $h$ and $\Phi$ (equivalent to $\Phi$ being identity and training a linear head).

*Contrastive pretraining.* We obtain $\Phi_{cl} := \arg\min_\Phi \mathcal{L}_{cl}(\Phi)$ by minimizing the Barlow Twins objective [93], which prior works have shown is also equivalent to spectral contrastive and non-contrastive objectives [33, 11]. Given probability distribution $P_A(a \mid x)$ for input $x$, and marginal $P_A$, we consider a constrained form of Barlow Twins in (3) which enforces features of "positive pairs" $a_1, a_2$ to be close while ensuring feature diversity. We assume a strict regularization ($\rho = 0$) for the theory arguments in the rest of the paper, and in App. D.2 we prove that all our claims hold for small $\rho$ as well. For augmentations, we scale the magnitude of each co-ordinate uniformly by an independent amount, i.e., $a \sim P_A(\cdot \mid x) = \mathbf{c} \odot x$, where $\mathbf{c} \sim \text{Unif}[0, 1]^d$. We try to mirror practical settings where the augmentations are fairly "generic", not encoding information about which features are invariant or spurious, and hence perturb all features symmetrically.

$$\mathcal{L}_{cl}(\Phi) \ := \ \mathbb{E}_{x \sim P_U} \mathbb{E}_{a_1, a_2 \sim P_A(\cdot \mid x)} \|\Phi(a_1) - \Phi(a_2)\|_2^2$$
$$\text{s.t.} \ \ \left\| \mathbb{E}_{a \sim P_A} \left[ \Phi(a)\Phi(a)^\top \right] - \mathbf{I}_k \right\|_F^2 \leq \rho \tag{3}$$

---

[2]See App. D.1 for similarities and differences of our setup with prior works.

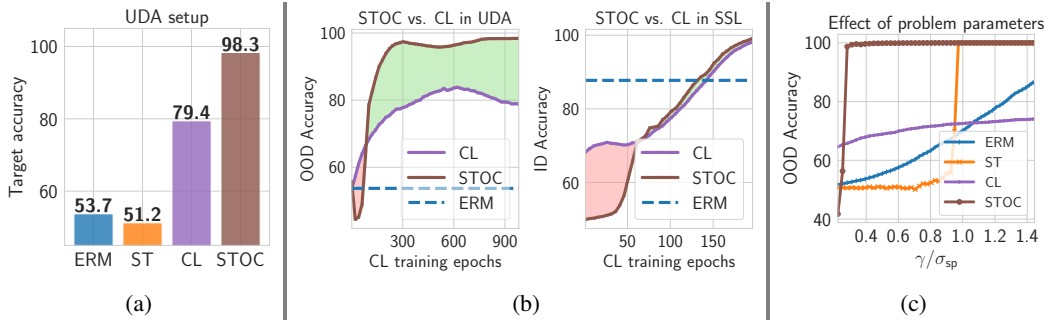

Figure 2: *Our simplified model of shift captures real-world trends and theoretical behaviors:* **(a)** Target (OOD) accuracy separation in the UDA setup (for problem parameters in Example 1). **(b)** Comparison of the benefits of STOC (ST over CL) over just CL in UDA and SSL settings, done across training iterations for contrastive pretraining. **(c)** Comparison between different methods in UDA setting, as we vary problem parameters $\gamma$ and $\sigma_{\text{sp}}$, connecting our theory results in Sec. 4.

Keeping the $\Phi_{\text{cl}}$ fixed, we then learn a linear classifier $h_{\text{cl}}$ over $\Phi_{\text{cl}}$ to minimize the exponential loss on labeled source data (refer to as *linear probing*). For STOC, keeping the $\Phi_{\text{cl}}$ fixed and initializing the linear head with the CL linear probe (instead of source only ERM), we perform ST with (2).

**Example 1.** *For the setup in* (1)*, we choose* $\gamma = 0.5$*,* $\sigma_{\text{sp}}^2 = 1.$*, and* $\sigma_{\text{in}}^2 = 0.05$ *with* $d_{\text{in}} = 5$ *and* $d_{\text{sp}} = 20$ *for our running example.* $\gamma/\sqrt{d_{\text{sp}}}$ *controls signal to noise ratio in the source such that spurious feature is easy-to-learn and the invariant feature is harder-to-learn.* $\sigma_2$ *controls the noise in target which we show later is critical in unlearning the spurious feature with CL.*

## 4.1 Simulations and Intuitive Story: A Comparative Study Between SSL and DA

**Our setup captures real-world trends in UDA setting.** Our toy setup (in Example 1) accentuates the behaviors observed on real-world datasets (Fig. 2(a)): (i) both ERM and ST yield close to random performance (though ST performs slightly worse than ERM); (ii) CL improves over ERM but still yields sub-optimal target performance; (iii) STOC then further improves over CL, achieving near-optimal target performance. Note that, a linear predictor can improve target performance only by reducing its dependence on spurious feature $x_{\text{sp}}$, and increasing it on invariant feature $x_{\text{in}}$ (along $w^\star$). Given this, we can explain our trends if we understand the following: (i) how ST reduces dependence on spurious feature when done after CL; (ii) why CL helps reduce but not completely eliminate the reliance of linear head on spurious features. Before we present intuitions, we ablate over a key problem parameter that affects both the target performance and conditions for ST to work.

**Effect of $\gamma/\sigma_{\text{sp}}$ on success of ST.** By increasing the ratio of margin $\gamma$ and variance of spurious feature on target $\sigma_{\text{sp}}$ (keeping others constant), the problem becomes easier because $\gamma$ directly affects the signal on $x_{\text{in}}$ and reducing $\sigma_{\text{sp}}$ helps ST to unlearn $x_{\text{sp}}$ (see App. D.3). In Fig. 2(c), we see that a phase transition occurs for ST, *i.e.*, after a certain threshold of $\gamma/\sigma_{\text{sp}}$, ST successfully recovers the optimal target predictor. This hints that ST has a binary effect, where beyond a certain magnitude of $\gamma/\sigma_{\text{sp}}$, ST can amplify the signal on domain invariant feature to obtain optimal target predictor. On the other hand, the performance of CL and ERM improve gradually where CL achieves high performance even at small ratios of $\gamma/\sigma_{\text{sp}}$. One way of viewing this trend with CL is that it magnifies the effective $\gamma/\sigma_{\text{sp}}$ in its representation space, because of which a linear head trained these representations have a good performance at low values of the ratio. Consequently, the *phase transition* of STOC occurs much sooner then that of ST. Finally, we note that for CL the rate of performance increase diminishes at high values of $\gamma/\sigma_{\text{sp}}$ because CL fails to reduce dependency along $x_{\text{sp}}$ beyond a certain point.

**An intuitive story.** We return to the question of why self-training improves over contrastive learning under distribution shift in our Example 1. When the classifier at initialization of ST relies more on spurious features, ST aggravates this dependency. However, as the problem becomes easier (with increasing $\gamma/\sigma_{\text{sp}}$), the source-only ERM classifier will start relying more on invariant rather than spurious feature. Once this ERM classifier is sufficiently accurate on the target, ST unlearns any dependency on spurious features achieving optimal target performance. In contrast, we observe that CL performs better than ERM but is still sub-optimal. This implies that CL ends up decreasing reliance on spurious features (as compared to ERM) but doesn't completely eliminate them. Combining ST and CL, a natural hypothesis explaining our trends is that CL provides a "favorable" initialization

for ST by sufficiently increasing signal on invariant features. Our empirical findings emphasize an intriguing contrast suggesting that ST and CL improve target performance in complementary ways.

**Why disparate behaviors for out-of-distribution vs. in distribution?** In the SSL setup, recall, there is no distribution shift. In Example 1, we sample $50k$ unlabeled data and $100$ labeled data from the same (target) distribution to simulate SSL setup. Substantiating our findings on real-world data, we observe that STOC provides a small to negligible improvement over CL (refer to App. D). To understand why such disparate behaviors emerge, recall that in the UDA setting, the main benefit of STOC lies in picking up reliance on "good" features for OOD data, facilitated by CL initialization. While contrastive pretraining uncovers features that are "good" for OOD data, it also learns more predictive source-only features (which are not predictive at all on target). As a result, linear probing with source-labeled data picks up these source-only features, leaving considerable room for improvement on OOD data with further self-training. On the other hand, in the SSL setting, the limited ID labeled data might provide enough signal to pick up features predictive on ID data, leaving little to no room for improvement for further self-training. Corroborating our intuitions, throughout the CL training in the toy setup, when CL doesn't achieve near-perfect generalization, the improvements provided by STOC for each checkpoint remain minimal. On the other hand, for UDA setup, after reaching a certain training checkpoint in CL, STOC yields significant improvement (Fig. 2(b)).

In the next sections, we formalize our intuitions and analyze why ST and CL offer complementary benefits when dealing with distribution shifts. Formal statements and proofs are in App. E.

### 4.2 Conditions for Success and Failure of Self-training over ERM from Scratch

In our results on Example 1, we observe that performing ST after ERM yields a classifier with near-random target accuracy. In Theorem 2, we characterize conditions under which ST fails and succeeds.

**Theorem 2** (Informal; Conditions for success and failure of ST over ERM). *The target accuracy of ERM classifier, is given by $0.5 \cdot \mathrm{erfc}\left(-\gamma^2/(\sqrt{2d_{\mathrm{sp}}} \cdot \sigma_{\mathrm{sp}})\right)$. Then ST performed in the second stage yields: (i) a classifier with $\approx 0.5$ target accuracy when $\gamma < 1/2\sigma_{\mathrm{sp}}$ and $\sigma_{\mathrm{sp}} \geqslant 1$; and (ii) a classifier with near-perfect target accuracy when $\gamma \geqslant \sigma_{\mathrm{sp}}$.*

The informal theorem above abstracts the exact dependency of $\gamma$, $\sigma_{\mathrm{sp}}$, and $d_{\mathrm{sp}}$ for the success and failure of ST over ERM. Our analysis highlights that while ERM learns a perfect predictor along $w_{\mathrm{in}}$ (with norm $\gamma$), it also learns to depend on $w_{\mathrm{sp}}$ (with norm $\sqrt{d_{\mathrm{sp}}}$) because of the perfect correlation of $x_{\mathrm{sp}}$ with labels on the source. Our conditions depict that when the $\gamma/\sigma_{\mathrm{sp}}$ is sufficiently small, then ST continues to erroneously enhance its reliance on the $x_{\mathrm{sp}}$ feature for target prediction, resulting in near-random target performance. Conversely, when $\gamma/\sigma_{\mathrm{sp}}$ is larger than 1, the signal in $x_{\mathrm{in}}$ is correctly used for predictor on the majority of target points, and ST eliminates the $x_{\mathrm{sp}}$ dependency, converging to an optimal target classifier.

Our proof analysis shows that if the ratio of the norm of the classifier along in the direction of $w^\star$ is smaller than $w_{\mathrm{sp}}$ by a certain ratio then the generated pseudolabels (incorrectly) use $x_{\mathrm{sp}}$ for its prediction further increasing the component along $w_{\mathrm{sp}}$. Moreover, normalization further diminishes the reliance along $w^\star$, culminating in a near-random performance. The opposite occurs when the ERM classifier achieves a signal along $w^\star$ that is sufficiently stronger than along $w_{\mathrm{sp}}$. Upon substituting the parameters used in Example 1, the ERM and ST performances as determined by Theorem 2 align with our empirical results, notably, ST performance on target being near-random.

### 4.3 CL Captures Both Features But Amplifies Invariant Over Spurious Features

Here we show that minimizing the contrastive loss (3) on unlabeled data from both $\mathrm{P_S}$ and $\mathrm{P_T}$ gives us a feature extractor $\Phi_{\mathrm{cl}}$ that has a higher inner product with the invariant feature over the spurious feature. First, we derive a closed form expression for $\Phi_{\mathrm{cl}}$ that holds for any linear backbone and augmentation distribution. Then, we introduce assumptions on the augmentation distribution (or equivalently on $w^\star$) and other problem parameters, that are sufficient to prove amplification.

**Proposition 3** (Barlow Twins solution). *The solution for (3) is $U_k^\top \Sigma_{\mathsf{A}}^{-1/2}$ where $U_k$ are the top $k$ eigenvectors of $\Sigma_{\mathsf{A}}^{-1/2} \widetilde{\Sigma} \Sigma_{\mathsf{A}}^{-1/2}$. Here, $\Sigma_{\mathsf{A}} := \mathbb{E}_{a \sim \mathrm{P_A}}[aa^\top]$ is the covariance over augmentations, and $\widetilde{\Sigma} := \mathbb{E}_{x \sim \mathrm{P_U}}[\widetilde{a}(x)\widetilde{a}(x)^\top]$ is the covariance matrix of mean augmentations $\widetilde{a}(x) := \mathbb{E}_{\mathrm{P_A}(a|x)}[a]$.*

The above result captures the effect of augmentations through the matrix $U_k$. If there were no augmentations, then $\Sigma_{\mathsf{A}} = \widetilde{\Sigma}$, implying that $U_k$ could then be any random orthonormal matrix. On

the other hand if augmentation distributions change prevalent covariances in the data, *i.e.*, $\Sigma_A$ is very different from $\widetilde{\Sigma}$, the matrix $U_k$ would bias the CL solution towards directions that capture significant variance in marginal distribution on augmented data, but have low conditional variance, when conditioned on original point $x$—precisely the directions with low invariance loss. Hence, we can expect that CL would learn components along both invariant $w_{\text{in}}$ and spurious $w_{\text{sp}}$ because: (i) these directions explain a large fraction of variance in the raw data; (ii) augmentations that randomly scale down dimensions would add little variance along $w_{\text{sp}}$ and $w_{\text{in}}$ compared to noise directions in their null space. On the other hand it is unclear which of these directions is amplified more in $\Phi_{\text{cl}}$. The following assumption and amplification result conveys that when the noise in target ($\sigma_{\text{sp}}$) is suficiently large, the CL solution amplifies the invariant feature over the spurious feature.

**Assumption 4** (Informal; Alignment of $w^\star$ with augmentations)**.** *We assume that $w^\star$ aligns with* $\mathrm{P}_A(\cdot \mid x)$, i.e., $\forall x$, $\mathbb{E}_{a|x}[a^\top w^\star] = 1/2 \cdot x^\top \mathrm{diag}(\mathbb{1}_d) w^\star$ *is high. Hence, we assume $w^\star = \mathbb{1}_{d_{\text{in}}}/\sqrt{d_{\text{in}}}$.*

One implication of Assumption 4 is that when $w^\star = \mathbb{1}_{d_{\text{in}}}/\sqrt{d_{\text{in}}}$, only the top two eigenvectors lie in the space spanned by $w_{\text{in}}$ and $w_{\text{sp}}$. To analyze our amplification with fewer eigenvectors from Proposition 3 while retaining all relevant phenomena, we assume $w^\star = \mathbb{1}_{d_{\text{in}}}/\sqrt{d_{\text{in}}}$ for mathematical convenience. While Assumption 4 permits a tighter theoretical analysis, our empirical results in Sec. 4.1 hold more generally for $w^\star \sim \mathcal{N}(0, \mathbf{I}_{d_{\text{in}}})$.

**Theorem 5** (Informal; CL recovers both $w_{\text{in}}$ and $w_{\text{sp}}$ but amplifies $w_{\text{in}}$)**.** *Under Assumption 4, the CL solution $\Phi_{\text{cl}} = [\phi_1, \phi_2, ..., \phi_k]$ satisfies $\phi_j^\top w_{\text{in}} = \phi_j^\top w_{\text{sp}} = 0 \; \forall j \geqslant 3$, $\phi_1 = c_1 w_{\text{in}} + c_3 w_{\text{sp}}$ and $\phi_2 = c_2 w_{\text{in}} + c_4 w_{\text{sp}}$. For constants $K_1, K_2 > 0$, $\gamma = K_1 K_2/\sigma_{\text{sp}}$, $d_{\text{sp}} = \sigma_{\text{sp}}^2/K_2^2$, $\forall \epsilon > 0$, $\exists \sigma_{\text{sp}0}$, such that for $\sigma_{\text{sp}} \geqslant \sigma_{\text{sp}0}$, $\left| c_1/c_3 - K_1 K_2^2 d_{\text{in}}/2L\sigma_{\text{in}}^2(d_{\text{in}}-1) \right| \leqslant \epsilon$, and $\left| |c_2/c_4| - L\sqrt{d_{\text{sp}}}/\gamma \right| \leqslant \epsilon$, where $L = 1 + K_2^2$.*

We analyze the amplification of $w_{\text{in}}/w_{\text{sp}}$ with contrastive learning in the regime where $\sigma_{\text{sp}}$ is large enough. In other words, if the target distribution has sufficient noise along the spurious feature, the augmentations prevent the CL solution from extracting components along $w_{\text{sp}}$. Thus, in our analysis, we first analyze the amplification factors asymptotically ($\sigma_{\text{sp}} \to \infty$), and then use the asymptotic behavior to draw conclusions for the regime where $\sigma_{\text{sp}}$ is large but finite.

Theorem 5 conveys two results: (i) CL recovers components along both $w_{\text{in}}$ and $w_{\text{sp}}$ through $\phi_1, \phi_2$; and (ii) it increases the norm along $w_{\text{in}}$ more than $w_{\text{sp}}$. The latter is evident because the margin separating labeled points along $w_{\text{in}}$ is now amplified by a factor of $|c_2/c_4| = \Omega(L\sqrt{d_{\text{sp}}}/\gamma)$ in $\phi_2$. Naturally, this will improve the target performance of a linear predictor trained over CL representations. At the same time, we also see that in $\phi_1$, the component along $w_{\text{sp}}$ is still significant ($c_1/c_3 = \mathcal{O}(1/L\sigma_{\text{in}}^2)$). Intuitively, CL prefers the invariant feature since augmentations amplify the noise along $w_{\text{sp}}$ in the target domain. At the same time, the variance induced by augmentations along $w_{\text{sp}}$ in source is still very small due to which the dependence on $w_{\text{sp}}$ is not completely alleviated. Due to the remaining components along $w_{\text{sp}}$, the target performance for CL can remain less than ideal. Both the above arguments on target performance are captured in Corollary 6.

**Corollary 6** (Informal; CL improves OOD error over ERM but is still imperfect)**.** *For $\gamma, \sigma_{\text{sp}}, d_{\text{sp}}$ defined as in Theorem 5, $\exists \sigma_{\text{sp}1}$ such that for all $\sigma_{\text{sp}} \geqslant \sigma_{\text{sp}1}$, the target accuracy of CL (linear predictor on $\Phi_{\text{cl}}$) is $\geqslant 0.5 \, \mathrm{erfc}\left( -L' \cdot \gamma/\sqrt{2}\sigma_{\text{sp}} \right)$ and $\leqslant 0.5 \, \mathrm{erfc}\left( -4L' \cdot \gamma/\sqrt{2}\sigma_{\text{sp}} \right)$, where $L' = K_2^2 K_1/\sigma_{\text{in}}^2(1-1/d_{\text{in}})$. When $\sigma_{\text{sp}1} > \sigma_{\text{in}}\sqrt{1 - 1/d_{\text{in}}}$, the lower bound on accuracy is strictly better than ERM from scratch.*

While $\Phi_{\text{cl}}$ is still not ideal for linear probing, in the next part we will see how $\Phi_{\text{cl}}$ can instead be sufficient for subsequent self-training to unlearn the remaining components along spurious features.

## 4.4 Improvements with Self-training Over Contrastive Learning

The result in the previous section highlights that while CL may improve over ERM, the linear probe continues to depend on the spurious feature. Next, we characterize the behavior STOC. Recall, in the ST stage, we iteratively update the linear head with (2) starting with the CL backbone and head.

**Theorem 7** (Informal; ST improves over CL)**.** *Under the conditions of Theorem 5 and $d_{\text{sp}} \leqslant K_1^2 \cdot K_2^{2/3}$, the target accuracy of ST over CL is lower bounded by $0.5 \cdot \mathrm{erfc}\left( - |c2/c4| \cdot \gamma/(\sqrt{2}\sigma_2) \right) \approx 0.5 \cdot \mathrm{erfc}\left( -L\sqrt{d_{\text{sp}}}/(\sqrt{2}\sigma_{\text{sp}}) \right)$ where $c_2$ and $c_4$ are the coefficients of feature $\phi_2$ along $w_{\text{in}}$ and $w_{\text{sp}}$ learned by BT.*

The above theorem states that when $\sqrt{d_{\text{sp}}}/\sigma_{\text{sp}} \gg 1$ the target accuracy of ST over CL is close to 1. In Example 1, the lower bound of the accuracy of ST over CL is $\text{erfc}\left(-\sqrt{10}\right) \approx 2$ showing near-perfect target generalization. Recall that Theorem 6 shows that CL yields a linear head that mainly depends on both the invariant direction $w_{\text{in}}$ and the spurious direction $w_{\text{sp}}$. At initialization, the linear head trained on the CL backbone has negligible dependence on $\phi_2$ (under conditions in Theorem 6). Building on that, the analysis in Theorem 7 captures that ST gradually reduces the dependence on $w_{\text{sp}}$ by learning a linear head that has a larger reliance on $\phi_2$, which has a higher "effective" margin on the target, thus increasing overall dependency on $w_{\text{in}}$.

**Theoretical comparison with SSL.** Our analysis until now shows that linear probing with source labeled data during CL picks up features that are more predictive of source label under distribution shift, leaving a significant room for improvement on OOD data when self-trained further. In UDA, the primary benefit of ST lies in picking up the features with a high "effective" margin on target data that are not picked up by linear head trained during CL. In contrast, in the SSL setting, the limited ID labeled data may provide enough signal in picking up high-margin features that are predictive on ID data, leaving little to no room for improvement for further ST. We formalize this intuition in App. E.

### 4.5 Reconciling Practice: Implications for Deep Non-Linear Networks

In this section, we experiment with deep non-linear backbone (*i.e.*, $\Phi_{\text{cl}}$). When we continue to fix $\Phi_{\text{cl}}$ during CL and STOC, the trends we observed with linear networks in Sec. 4.1 continue to hold. We then perform full fine-tuning with CL and STOC, i.e., propagate gradients even to $\Phi_{\text{cl}}$, as commonly done in practice. We present key takeaways here but detailed experiments are in App. D.4.

**Benefits of augmentation for self-training.** ST while updating $\Phi_{\text{cl}}$ can hurt due to overfitting issues when training with the finite sample of labeled and unlabeled data (drop by >10% over CL). This is due to the ability of deep networks to overfit on confident but incorrect pseudolabels on target data [94]. This exacerbates components along $w_{\text{sp}}$ and we find that augmentations (and other heuristics) typically used in practice (*e.g.* in FixMatch [79]) help avoid overfitting on incorrect pseudolabels.

**Can ERM and ST over contrastive pretraining improve features?** We find that self-training can also slightly improve features when we update the backbone with the second stage of STOC and when the CL backbone is early stopped sub-optimally (*i.e.* at an earlier checkpoint in Fig. 2(b)). This feature finetuning can now widen the gap between STOC and CL in SSL settings, as compared to the linear probing gap (as in 2). This is because STOC can now improve performance beyond just recovering the generalization gap for the linear head (which is typically small). However, STOC benefits are negligible when CL is not early stopped sub-optimally, *i.e.*, trained till convergence. Thus, it remains unclear if STOC and CL have complementary benefits for feature learning in UDA or SSL settings. Investigating this is an interesting avenue for future work.

## 5 Connecting Experimental Gains with Theoretical Insights

Our theory emphasizes that under distribution shift contrastive pretraining improves the representations for target data, while self-training primarily improves linear classifiers learned on top. To investigate different methods in our UDA setup, we study the representations learned by each of them. We fix the representations and train linear heads over them to answer two questions: (i) How good are the representations in terms of their *ceiling* of target accuracy (performance of the optimal linear probe)?—we evaluate this by training the classifier head on target labeled data (*i.e.*, target linear probe); and (ii) How well do heads trained on source generalize to target?—we assess this by training a head on source labeled data (source linear probe) and evaluate its difference with target linear probe. For both, we plot target accuracy. We make two *intriguing* observations Fig. 3:

**Does CL improve representations over ERM features?** Yes. We observe a substantial difference in accuracy ($\approx$ 14% gap) of target linear probes on backbones trained

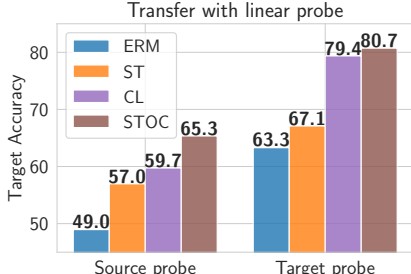

Figure 3: *Target accuracy with source and target linear probes*, which freezes backbones trained with various objectives and trains only the head in UDA setup. Avg. accuracy across all datasets. We observe that: (i) ST improves the linear transferability of source probes, and (ii) CL improves representations.

with contrastive pretraining (*i.e.* CL, STOC) and without it (*i.e.*, ERM, ST) highlighting that CL significantly pushes the performance ceiling over non-contrastive features. As a side, our findings also stand in contrast to recent studies suggesting that ERM features might be "good enough" for OOD generalization [67, 46]. Instead, the observed gains with contrastively pretrained backbones (*i.e.* CL, STOC) demonstrate that target unlabeled data can be leveraged to further improve over ERM features.

**Do CL features yield *perfect* linear transferability from source to target?** Recent works [40, 76] conjecture that under certain conditions CL representations, linear probes learned with source labeled data may transfer perfectly from source to target. However, we observe that this doesn't hold strictly in practice, and in fact, the linear transferability can be further improved with ST. We first note a significant gap between the performance of source linear probes and target linear probes illustrating that linear transferability is not perfect in practice. Moreover, while the accuracy of target linear probes doesn't change substantially between CL and STOC, the accuracy of the source linear probe improves significantly. Similar observations hold for ERM and ST, methods trained without contrastive pretraining. This highlights that ST performs "feature refinement" to improve source to target linear transfer (with relatively small improvements in their respective target probe performance). *The findings highlight the complementary nature of benefits on real-world data: ST improves linear transferability while CL improves representations.*

## 6 Connections to Prior Work

Our empirical results and our analyses offer a perspective that contrasts with the prior literature that argues for the individual optimality of contrastive pretraining and self-training. We outline the key differences from existing studies here, and delve into other related works in App. A.

**Limitations of prior work analyzing contrastive learning** Prior works [40, 76] analyzing CL first make assumptions on the consistency of augmentations with labels [39, 11, 73, 44], and specifically for UDA make stronger ones on the augmentation graph connecting examples from same domain or class more than cross-class/cross-domain ones. While this is sufficient to prove linear transferability, it is unclear if this holds in practice when augmentations are imperfect, *i.e.* if they fail to mask the spurious features completely—as corroborated by our findings in Sec. 5. We show why this also fails in our simplified setup in App. F.1.

**Limitations of prior work analyzing self-training** Prior research views self-training as consistency regularization, ensuring pseudolabels for original samples align with their augmentations [12, 87, 79]. This approach abstracts the role played by the optimization algorithm and instead evaluates the global minimizer of a population objective promoting pseudolabel consistency. It also relies on specific assumptions about class-conditional distributions to guarantee pseudolabel accuracy across domains. However, this framework doesn't address issues in iterative label propagation. For example, when augmentation distribution has long tails, the consistency of pseudolabels depends on the sampling frequency of "favorable" augmentations (for more discussion see App. F.2). Our analysis thus follows the iterative examination of self-training [17].

## 7 Conclusion

In this study, we highlight the synergistic behavior of self-training and contrastive pretraining under distribution shift. Shifts in distribution are commonplace in real-world applications of machine learning, and even under natural, non-adversarial distribution shifts, the performance of machine learning models often drops. By simply combining existing techniques in self-training and constrastive learning, we find that we can improve accuracy by 3–8% rather than using either approach independently. Despite these significant improvements, we note that one limitation of this combined approach is that performing self-training sequentially after contrastive pretraining increases the computation cost for UDA. The potential for integrating these benefits into one unified training paradigm is yet unclear, presenting an interesting direction for future exploration.

Beyond this, we note that our theoretical framework primarily confines the analysis to training the backbone and linear network independently during the pretraining and fine-tuning/self-training phases. Although our empirical observations apply to deep networks with full fine-tuning, we leave a more rigorous theoretical study of full fine-tuning for future work. Our theory also relies on a covariate shift assumption (where we assume that label distribution also doesn't shift). Investigating the complementary nature of self-training and contrastive pretraining beyond the covariate shift assumption would be another interesting direction for future work.

## Acknowledgements

SG acknowledges the JP Morgan AI Ph.D. Fellowship and Bloomberg Ph.D. Fellowship for their support.

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

# Appendix

## Appendix Table of Contents

## A   Other Related Works

**Unsupervised domain adaption.**   Without assumption on the nature of shift, UDA is underspecified [7]. This challenge has been addressed in various ways by researchers. One approach is to investigate additional structural assumptions under which UDA problems are well posed [77, 74]. Popular settings for which DA is well-posed include (i) *covariate shift* [96, 92, 19, 20, 35] where $p(x)$ can change from source to target but $p(y|x)$ remains invariant; and (ii) *label shift* [69, 53, 3, 1, 31, 95, 66, 30] where the label marginal $p(y)$ can change but $p(x|y)$ is shared across source and target. Principled methods with strong theoretical guarantees exists for adaptation under these settings when target

distribution's support is a subset of the source support. Other works [25, 6, 32, 28] extend the label shift setting to scenarios where previously unseen classes may appear in the target and $p(x|y)$ remains invariant among seen classes. A complementary line of research focuses on constructing benchmarks to develop heuristics for incorporating the unlabeled target data, relying on benchmark datasets ostensibly representative of "real-world shifts" to adjudicate progress [72, 85, 70, 62, 63]. As a result, various benchmark-driven heuristics have been proposed [54, 55, 82, 81, 99, 98, 26, 79]. Our work engages with the latter, focusing on two popular methods: self-training and contrastive pretraining.

**Domain generalization.** In domain generalization, the model is given access to data from multiple different domains and the goal is to generalize to a previously unseen domain at test time [10, 59]. For a survey of different algorithms for domain generalization, we refer the reader to Gulrajani and Lopez-Paz [37]. A crucial distinction here is that unlike the domain generalization setting, in DA problems, we have access to unlabeled examples from the test domain.

**Semi-supervised learning.** To learn from a small amount of labeled supervision, semi-supervised learning methods leverage unlabeled data alongside to improve learning models. One of the seminal works in SSL is the pseudolabeling method [75], where a classifier is trained on the labeled data and then used to classify the unlabeled data, which are then added to the training set. The work of Zhu and Ghahramani [100] built on this by introducing graph-based methods, and the transductive SVMs [43] presented an SVM-based approach. More recent works have focused on deep learning techniques, and similar to UDA, self-training and contrastive pretraining have emerged as two prominent choices. We delve into these methods in greater detail in the following paragraphs. For a discussion on other SSL methods, we refer interested readers to [15, 84, 91].

**Self-training.** Two popular forms of self-training are pseudolabeling [52] and conditional entropy minimization [34], which have been observed to be closely connected [8, 52, 79, 78]. Motivated by its strong performance in SSL and UDA settings [79, 89, 30, 78], several theoretical works have made attempts to understand its behavior [50, 87, 17]. [87, 12] aims to understand the behavior of the global minimizer of self-training objective by studying input consistency regularization, which enforces stability of the prediction for different augmentations of the unlabeled data. Our analysis of self-training is motivated by the work of Chen et al. [17] which explores the iterative behavior of self-training to unlearn spurious features. The setting of spurious features is of particular interest, since prior works have specifically analyzed the failures of out-of-distribution generalization in the presence of spurious features [60, 71].

**Contrastive learning.** An alternate line of work that uses unlabeled data for learning representations in the pretraining stage is contrastive learning [36, 61, 13, 16, 88]. Given an augmentation distribution, the main goal of contrastive objectives is to map augmentations drawn from the same input (positive pairs) to similar features, and force apart features corresponding to augmentations of different inputs (negative pairs) [13, 14, 41]. Prior works [11, 44, 38] have also shown a close relationship between contrastive [16, 39] and non-contrastive objectives [4, 93]. Consequently, in our analysis pertaining to the toy setup we focus on the mathematically non-contrastive objective Barlow Twins [93]. Using this pretrained backbone (either as an initialization or as a fixed feature extractor) a downstream predictor is learned using labeled examples. Several works [39, 73, 38, 2, 44] have analyzed the in-distribution generalization of the downstream predictor via label consistency arguments on the graph of positive pairs (augmentation graph). In contrast, we study the impact of contrastive learning under distribution shifts in the UDA setup. Other works [76, 40] that examine contrastive learning for UDA also conform to the augmentation graph view point, making additional assumptions that guarantee linear transferability. In our simplified setup involving spurious correlations, these abstract assumptions break easily when the augmentations are of a generic nature, akin to practice. Finally, some empirical works [58, 57] have found self-supervised objectives like contrastive pretraining to reduce dependence on spurious correlations. Corroborating their findings, we extensively evaluate the complementary benefits of contrastive learning and self-training on real-world datasets. Finding differing results in SSL and UDA settings, we further examine their behavior theoretically in our toy setup.

# B  More Details on Problem Setup

In this section, we elaborate on our setup and methods studied in our work.

**Unsupervised Domain Adaptation (UDA).** We assume that we are given labeled data from the *source* distribution and unlabeled data from a shifted, *target* distribution, with the goal of performing well on target data. We assume that the source and target distributions have the same label marginals $P_S(y) = P_T(y)$ (*i.e.*, no label proportion shift) and the same Bayes optimal predictor, *i.e.*, $\arg\max_y p_S(y \mid x) = \arg\max_y p_T(y \mid x)$. Here, even with infinite labeled source data, the challenge lies in generalizing out-of-distribution. In experiments, we assume access to finite data but in theory, we assume population access to labeled source and unlabeled target.

**Semi-Supervised Learning (SSL).** Here, there is no distribution shift, *i.e.*, $P_S = P_T = P_U$. We are given a small number of labeled examples and a comparatively large amount of unlabeled examples, both drawn from the same distribution. Without loss of generality, we denote this distribution with $P_T$. The goal in SSL is to generalize in-distribution. The challenge is primarily due to limited access to labeled data. Here, in experiments, we assume limited access to labeled data but a comparatively larger amount of unlabeled in-distribution data. In theory, we assume population access to unlabeled data but limited labeled examples.

**Methods.** As discussed in the main paper, we compare four methods for learning with labeled and unlabeled data. Table 8 summarizes the main methods and key differences between those methods in UDA and SSL setup. For exact implementation in our experiments, we refer reader to App. C.3.

## C  Additional Experiments and Details

### C.1  Additional setup and notation

Recall, our goal is to learn a predictor that maps inputs $x \in \mathcal{X} \subseteq \mathbb{R}^d$ to outputs $y \in \mathcal{Y}$. We parameterize predictors $f = h \circ \Phi : \mathbb{R}^d \mapsto \mathcal{Y}$, where $\Phi : \mathbb{R}^d \mapsto \mathbb{R}^k$ is a feature map and $h \in \mathbb{R}^k$ is a classifier that maps the representation to the final scores or logits. With $A : \mathcal{X} \to \mathcal{A}$, we denote the augmentation function that takes in an input $x$ and outputs an augmented view of the input $A(x)$. Unless specified otherwise, we perform full-finetuning in all of our experiments on real-world data. That is, we backpropagate gradients in both the linear head $h$ and the backbone $\phi$. For UDA, we denote source labeled points as $\{(x_i, y_i)\}_{i=1}^n$ and target unlabeled points as $\{(x_i')\}_{i=1}^m$. For SSL, we use the same notation for labeled and unlabeled in-distribution data.

### C.2  Dataset details

For both UDA and SSL, we conduct experiments across eight benchmark datasets. Each of these datasets consists of domains, enabling us to construct source-target pairs for UDA. The adopted source and target domains are standard to previous studies [76, 30, 70]. Because the SSL setting lacks distribution shift, we do not need to worry about domain designations and default to using source alone. To simulate limited supervision in SSL, we sub-sample the original labeled training set to 10%. Below provide exact details about the datasets used in our benchmark study.

- **CIFAR10** We use the original CIFAR10 dataset [48] as the source dataset. For target domains, we consider CINIC10 [22] which is a subset of Imagenet restricted to CIFAR10 classes and downsampled to $32 \times 32$.

- **FMoW** In order to consider distribution shifts faced in the wild, we consider FMoW-WILDs [47, 18] from WILDS benchmark, which contains satellite images taken in different geographical regions and at different times. We use the original train as source and OOD val and OOD test splits as target domains as they are collected over different time-period. Overall, we obtain 3 different domains (1 source and 2 targets).

- **BREEDs** We also consider BREEDs benchmark [72] in our setup to assess robustness to subpopulation shifts. BREEDs leverage class hierarchy in ImageNet [68] to re-purpose original classes to be the subpopulations and defines a classification task on superclasses. We consider distribution shift due to subpopulation shift which is induced by directly making the subpopulations present in the training and test distributions disjoint. BREEDs benchmark contains 4 datasets **Entity-13**, **Entity-30**, **Living-17**, and **Non-living-26**, each focusing on different subtrees and levels in the hierarchy. Overall, for each of the 4 BREEDs datasets (i.e., Entity-13, Entity-30, Living-

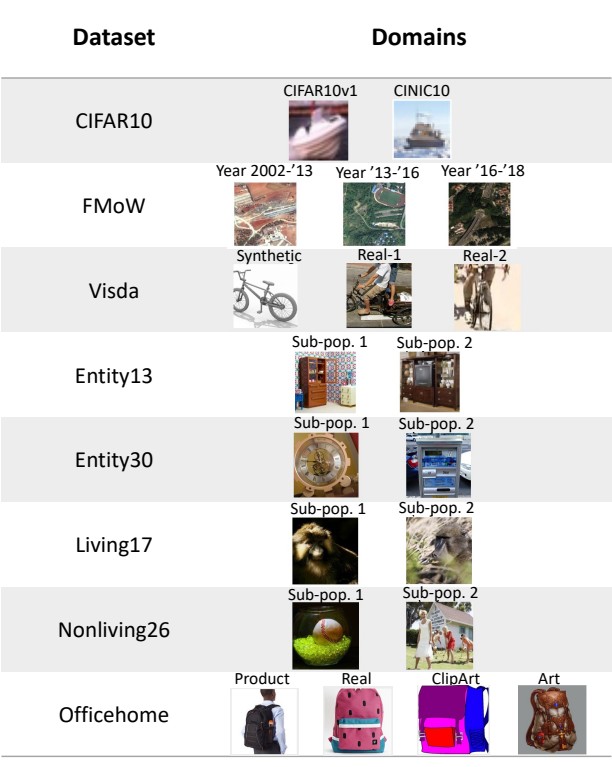

Figure 4: Examples from all the domains in each dataset.

17, and Non-living-26), we obtain one different domain which we consider as target. We refer to source and target as follows: BREEDs sub-population 1, BREEDs sub-population 2.

- **OfficeHome** We use four domains (art, clipart, product and real) from OfficeHome dataset [85]. We use the product domain as source and the other domains as target.

- **Visda** We use three domains (train, val and test) from the Visda dataset [64, 63]. While 'train' domain contains synthetic renditions of the objects, 'val' and 'test' domains contain real world images. To avoid confusing, the domain names with their roles as splits, we rename them as 'synthetic', 'Real-1' and 'Real-2'. We use the synthetic (original train set) as the source domain and use the other domains as target.

We summarize the information about source and target domains in Table 3.

**Train-test splits** We partition each source and target dataset into 80% and 20% i.i.d. splits. We use 80% splits for training and 20% splits for evaluation (or validation). We throw away labels for the 80% target split and only use labels in the 20% target split for final evaluation. The rationale behind splitting the target data is to use a completely unseen batch of data for evaluation. This avoids evaluating on examples where a model potentially could have overfit. over-fitting to unlabeled examples for evaluation. In practice, if the aim is to make predictions on all the target data (i.e., transduction), we can simply use the (full) target set for training and evaluation.

**Simulating SSL settings and limited supervision.** For SSL settings, we choose the in-distribution domain as the source domain. To simulate limited supervision in SSL, we sub-sample the original labeled training set to 10% and use all the original dataset as unlabeled data. For evaluation, we further split the original holdout set into two partitions (one for validation and the other to report final accuracy numbers).

### C.3 Method details

For implementation, we build on top of WILDs [70] and RLSbench [30] open source libraries.

| Dataset | Source | Target |
|---|---|---|
| CIFAR10 | CIFAR10v1 | CINIC10 |
| FMoW | FMoW (2002–'13) | FMoW (2013–'16), FMoW (2016–'18) |
| Entity13 | Entity13 (sub-population 1) | Entity13 (sub-population 2) |
| Entity30 | Entity30 (sub-population 1) | Entity30 (sub-population 2), |
| Living17 | Living17 (sub-population 1) | Living17 (sub-population 2), |
| Nonliving26 | Nonliving26 (sub-population 1) | Nonliving26 (sub-population 2), |
| Officehome | Product | Product, Art, ClipArt, Real |
| Visda | Synthetic (originally referred to as train) | Synthetic, Real-1 (originally referred to as val), Real-2 (originally referred to as test) |

Table 3: Details of source and target sets in each dataset considered in our testbed.

**ERM (Source only) training.** We consider Empirical Risk Minimization (ERM) on the labeled source data as a baseline. Since this simply ignores the unlabeled target data, we call this as source only training. As mentioned in the main paper, we perform source only training with data augmentations. Formally, we minimize the following ERM loss:

$$L_{\text{source only}}(f) = \frac{1}{n} \sum_{i=1}^{n} \ell(f(A(x_i), y_i)), \tag{4}$$

where $A$ is the stochastic data augmentation operation and $\ell$ is a loss function. For SSL, the ERM baseline only uses the small of labeled data available.

**Contrastive Learning (CL).** We perform contrastive pretraining on the unlabeled dataset to obtain the backbone $\phi_{\text{cl}}$. And then we perform full fine-tuning with source labeled data by initializing the backbone with $\phi_{\text{cl}}$. We use SwAV [13] for contrastive pretraining. The main idea behind SwAV is to train a model to identify different views of the same image as similar, while also ensuring that it finds different images to be distinct. This is accomplished through a *swapped* prediction mechanism, where the goal is to compute a code from an augmented version of the image and predict this code from other augmented versions of the same image. In particular, given two image features $\phi(x'_{a1})$ and $\phi(x'_{a2})$ from two different augmentations of the same image $x'$, i.e., $x'_{a1}, x'_{a2} \sim A(x')$, SwAV computes their codes $z_{a1}$ and $z_{a2}$ by matching the features to a set of $K$ prototypes $\{c_1, \cdots, c_K\}$. Then SwAV minimizes the following loss such that $\phi(x'_{a1})$ can compute codes $z_{a2}$ and $\phi(x'_{a2})$ can compute codes $z_{a1}$:

$$L_{\text{SwAV}}(\phi) = \sum_{i=1}^{m} \sum_{x'_{i,a1}, x'_{i,a2} \sim A(x'_i)} \ell'(\phi(x'_{i,a1}), z_{i,a2}) + \ell'(\phi(x'_{i,a2}), z_{i,a1}), \tag{5}$$

where $\ell'$ computes KL-divergence between codes computed with features (e.g. $\phi(x_{a1})$) and the code computed by another view (e.g. $z_{a2}$). For more details about the algorithm, we refer the reader to Caron et al. [13]. In all UDA settings, unless otherwise specified, we pool all the (unlabeled) data from the source and target to perform SwAV. For SSL, we leverage in-distribution unlabeled data.

We employ SimCLR [16] for the CIFAR10 dataset, aligning with previous studies that have utilized contrastive pretraining on the same dataset [51, 76]. The reason for this choice is that SwAV relies on augmentations that involve cropping images to a smaller resolution, making it more suitable for datasets with larger resolutions beyond $32 \times 32$.

**Self-Training (ST).** For self-training, we apply FixMatch [79], where the loss on labeled data and on pseudolabeled unlabeled data are minimized simultaneously. Sohn et al. [79] proposed FixMatch as a variant of the simpler Pseudo-label method [52]. This algorithm dynamically generates psuedolabels and overfits on them in each batch. FixMatch employs consistency regularization on the unlabeled data. In particular, while pseudolabels are generated on a weakly augmented view of

the unlabeled examples, the loss is computed with respect to predictions on a strongly augmented view. The intuition behind such an update is to encourage a model to make predictions on weakly augmented data consistent with the strongly augmented example. Moreover, FixMatch only overfits to the assigned labeled with weak augmentation if the confidence of the prediction with strong augmentation is greater than some threshold $\tau$. Refer to $A_{\text{weak}}$ as the weak-augmentation and $A_{\text{strong}}$ as the strong-augmentation function. Then, FixMatch uses the following loss function:

$$
\begin{aligned}
L_{\text{FixMatch}}(f) = {} & \frac{1}{n} \sum_{i=1}^{n} \ell(f(A_{\text{strong}}(x_i), y_i)) \\
& + \frac{\lambda}{m} \sum_{i=1}^{m} \ell(f(A_{\text{strong}}(x_i'), \widetilde{y}_i)) \cdot \mathbb{I}\left[\max_y f_y(A_{\text{strong}}(x_i')) \geqslant \tau\right],
\end{aligned}
$$

where $\widetilde{y}_i = \arg\max_y f_y(T_{\text{weak}}(x_i))$. For UDA, our unlabeled data is the union of source and target unlabeled data. For SSL, we only leverage in-distribution unlabeled data.

We adapted our implementation from Sagawa et al. [70] which matches the implementation of Sohn et al. [79] except for one detail. While Sohn et al. [79] augments labeled examples with weak augmentation, Sagawa et al. [70] proposed to strongly augment the labeled source examples.

**Self-Training Over Contrastive learning (STOC).** Finally, rather than performing FixMatch from a randomly initialized backbone, we initialize FixMatch with a contrastive pretrained backbone.

### C.4   Additional UDA experimemts

Table 4: *Results in the UDA setup*. We report accuracy on target (OOD) data from which we only observe unlabeled examples during training. For benchmarks with multiple target distributions (*e.g.*, OH, Visda), we report average accuracy on those targets.

| Method | Living17 | Nonliv26 | Entity13 | Entity30 | FMoW (2 tgts) | Visda (2 tgts) | OH (3 tgts) | CIFAR→ CINIC |
|---|---|---|---|---|---|---|---|---|
| ERM | $60.2_{\pm 0.1}$ | $45.4_{\pm 0.2}$ | $68.6_{\pm 0.1}$ | $55.7_{\pm 0.0}$ | $56.5_{\pm 0.1}$ | $20.8_{\pm 0.2}$ | $9.5_{\pm 0.2}$ | $74.3_{\pm 0.1}$ |
| ST | $71.1_{\pm 0.2}$ | $56.8_{\pm 0.1}$ | $78.0_{\pm 0.3}$ | $66.7_{\pm 0.1}$ | $56.9_{\pm 0.4}$ | $39.1_{\pm 0.1}$ | $11.1_{\pm 0.1}$ | $78.3_{\pm 0.3}$ |
| CL | $74.1_{\pm 0.2}$ | $57.4_{\pm 0.3}$ | $76.9_{\pm 0.2}$ | $66.6_{\pm 0.3}$ | $61.5_{\pm 0.5}$ | $63.2_{\pm 0.2}$ | $22.8_{\pm 0.1}$ | $77.5_{\pm 0.1}$ |
| STOC (ours) | $\mathbf{82.6}_{\pm 0.1}$ | $\mathbf{62.1}_{\pm 0.2}$ | $\mathbf{81.9}_{\pm 0.2}$ | $\mathbf{72.0}_{\pm 0.2}$ | $\mathbf{65.3}_{\pm 0.1}$ | $\mathbf{70.1}_{\pm 0.2}$ | $\mathbf{27.1}_{\pm 0.3}$ | $\mathbf{79.9}_{\pm 0.3}$ |

Table 5: *Results in the UDA setup with source only contrastive pretraining*. We report accuracy on target (OOD) data from which we only observe unlabeled examples during training. For benchmarks with multiple target distributions (*e.g.*, OH, Visda), we report average accuracy on those targets.

| Method | Living17 | Nonliv26 | Entity13 | Entity30 | FMoW (2 tgts) | Visda (2 tgts) | OH (3 tgts) | CIFAR→ CINIC |
|---|---|---|---|---|---|---|---|---|
| CL (source only) | $67.3_{\pm 0.1}$ | $49.1_{\pm 0.2}$ | $71.5_{\pm 0.1}$ | $58.5_{\pm 0.3}$ | $53.9_{\pm 0.1}$ | $33.3_{\pm 0.2}$ | $21.7_{\pm 0.1}$ | $77.7_{\pm 0.1}$ |
| STOC (source only) | $75.0_{\pm 0.2}$ | $58.4_{\pm 0.1}$ | $79.8_{\pm 0.3}$ | $67.5_{\pm 0.1}$ | $56.3_{\pm 0.4}$ | $42.7_{\pm 0.1}$ | $25.7_{\pm 0.1}$ | $77.8_{\pm 0.1}$ |
| CL | $74.1_{\pm 0.2}$ | $57.4_{\pm 0.3}$ | $76.9_{\pm 0.2}$ | $66.6_{\pm 0.3}$ | $61.5_{\pm 0.5}$ | $63.2_{\pm 0.2}$ | $22.8_{\pm 0.1}$ | $77.5_{\pm 0.1}$ |
| STOC | $\mathbf{82.6}_{\pm 0.1}$ | $\mathbf{62.1}_{\pm 0.2}$ | $\mathbf{81.9}_{\pm 0.2}$ | $\mathbf{72.0}_{\pm 0.2}$ | $\mathbf{65.3}_{\pm 0.1}$ | $\mathbf{70.1}_{\pm 0.2}$ | $\mathbf{27.1}_{\pm 0.3}$ | $\mathbf{79.9}_{\pm 0.3}$ |

## C.5 Additional SSL experimemts

Table 6: *Results in the SSL setup.* We report accuracy on hold-out ID data. Recall that SSL uses labeled and unlabeled data from the same distribution during training.

| Method | Living17 | Nonliv26 | Entity13 | Entity30 | FMoW | Visda | OH | CIFAR |
|---|---|---|---|---|---|---|---|---|
| ERM | $76.8_{\pm 0.1}$ | $64.9_{\pm 0.2}$ | $80.1_{\pm 0.0}$ | $70.4_{\pm 0.3}$ | $33.6_{\pm 0.4}$ | $99.2_{\pm 0.0}$ | $32.0_{\pm 0.2}$ | $85.5_{\pm 0.1}$ |
| ST | $85.4_{\pm 0.1}$ | $75.7_{\pm 0.2}$ | $85.4_{\pm 0.2}$ | $77.3_{\pm 0.1}$ | $33.6_{\pm 0.3}$ | $99.2_{\pm 0.1}$ | $32.0_{\pm 0.1}$ | $93.1_{\pm 0.1}$ |
| CL | $91.1_{\pm 0.5}$ | $84.6_{\pm 0.6}$ | $90.7_{\pm 0.4}$ | $85.5_{\pm 0.3}$ | $43.1_{\pm 0.2}$ | $97.6_{\pm 0.3}$ | $49.7_{\pm 0.2}$ | $91.7_{\pm 0.2}$ |
| STOC (ours) | $92.0_{\pm 0.1}$ | $85.8_{\pm 0.2}$ | $91.3_{\pm 0.3}$ | $86.1_{\pm 0.2}$ | $44.4_{\pm 0.1}$ | $97.7_{\pm 0.2}$ | $49.9_{\pm 0.2}$ | $93.06_{\pm 0.3}$ |

## C.6 Other experimental details

**Augmentations.** For weak augmentation, we leverage random horizontal flips and random crops of pre-defined size. For SwAV, we also perform multicrop augmentation as proposed in Caron et al. [13]. For strong augmentation, we apply the following transformations sequentially: random horizontal flips, random crops of pre-defined size, augmentation with Cutout [24], and RandAugment [21]. For the exact implementation of RandAugment, we directly use the implementation of Sohn et al. [79]. Unless specified otherwise, for all methods, we default to using strong augmentation techniques.

**Architectures.** In our work, we experiment with Resnet18, Resnet50 [42] trained from scratch (*i.e.* random initialization). We do not consider off-the-shelf pretrained models (*e.g.*, on Imagenet [68]) to avoid confounding our conclusions about contrastive pretraining. However, we note that our results on most datasets tend to be comparable to and sometimes exceed those obtained with ImageNet pretrained models. For BREEDs datasets, we employ Resnet18 architecture. For other datasets, we train a Resnet50 architecture.

Except for Resnets on CIFAR dataset, we used the standard pytorch implementation [27]. For Resnet on Cifar, we refer to the implementation here: `https://github.com/kuangliu/pytorch-cifar`. For all the architectures, whenever applicable, we add antialiasing [97]. We use the official library released with the paper.

**Hyperparameters.** For all the methods, we fix the algorithm-specific hyperparameters to the original recommendations. For UDA, given that the setup precludes access to labeled data from the target distribution, we use source hold-out performance to pick the best hyperparameters. During pretraining, early stopping is done according to lower values of pretraining loss.

We tune the learning rate and $\ell_2$ regularization parameter by fixing the batch size for each dataset that corresponds to the maximum we can fit to 15GB GPU memory. We default to using cosine learning rate schedule [56]. We set the number of epochs for training as per the suggestions of the authors of respective benchmarks. For SSL, we run both ERM and FixMatch for approximately 2000 epochs. Note that we define the number of epochs as a full pass over the labeled training source data. We summarize the learning rate, batch size, number of epochs, and $\ell_2$ regularization parameter used in our study in Table 7.

**Compute infrastructure.** Our experiments were performed across a combination of Nvidia T4, A6000, and V100 GPUs.

# D  Additional Results in Toy Setup

In this section we will first give more details on our simplified setup that captures both contrastive pretraining and self-training in the same framework. Then, we provide some additional empirical results that are not captured theoretically but mimic behaviors observed in real world settings, highlighting the richness of our setup.

## D.1  Detailed description of our simplified setup

In this subsection, we will first re-iterate the problem setup in Sec. 4 and provide some comparisons between our setup and those in closely related works. We will then describe the four methods: ERM,

| Dataset | Batch size | $\ell_2$ regularization set | Learning rate set |
|---|---|---|---|
| CIFAR10 | 200 | $\{0.001, 0.0001, 10^{-5}, 0.0\}$ | $\{0.2, 0.1, 0.05, 0.01, 0.003, 0.001\}$ |
| FMoW | 64 | $\{0.001, 0.0001, 10^{-5}, 0.0\}$ | $\{0.01, 0.003, 0.001, 0.0003, 0.0001\}$ |
| Entity13 | 256 | $\{0.001, 0.0001, 10^{-5}, 0.0\}$ | $\{0.4, 0.2, 0.1, 0.05, 0.02, 0.01, 0.005\}$ |
| Entity30 | 256 | $\{0.001, 0.0001, 10^{-5}, 0.0\}$ | $\{0.4, 0.2, 0.1, 0.05, 0.02, 0.01, 0.005\}$ |
| Entity30 | 256 | $\{0.001, 0.0001, 10^{-5}, 0.0\}$ | $\{0.4, 0.2, 0.1, 0.05, 0.02, 0.01, 0.005\}$ |
| Nonliving26 | 256 | $\{0.001, 0.0001, 10^{-5}, 0.0\}$ | $\{0.4, 0.2, 0.1, 0.05, 0.02, 0.01, 0.005\}$ |
| Officehome | 96 | $\{0.001, 0.0001, 10^{-5}, 0.0\}$ | $\{0.01, 0.003, 0.001, 0.0003, 0.0001\}$ |
| Visda | 96 | $\{0.001, 0.0001, 10^{-5}, 0.0\}$ | $\{0.03, 0.01, 0.003, 0.001, 0.0003\}$ |

Table 7: Details of the batch size, learning rate set and $\ell_2$ regularization set considered in our testbed.

ST, CL, and STOC, providing details on the exact estimates returned by these algorithms in the SSL and UDA settings.

**Data distribution.** We consider binary classification and model the inputs as consisting of two kinds of features: $x = [x_{\mathrm{in}}, x_{\mathrm{sp}}]$ where $x_{\mathrm{in}} \in \mathbb{R}^{d_{\mathrm{in}}}$ is the invariant feature that is predictive of the label across both source $\mathrm{P_S}$ and target $\mathrm{P_T}$ and $x_{\mathrm{sp}} \in \mathbb{R}^{d_{\mathrm{sp}}}$ is the spurious feature that is correlated with the label $y$ only on the source domain $\mathrm{P_S}$ but uncorrelated with label $y$ in $\mathrm{P_T}$. Here, $x_{\mathrm{in}} \in \mathbb{R}^{d_{\mathrm{in}}}$ determines the label using the ground truth classifier $w^\star \sim \mathrm{Unif}(\mathbb{S}^{d_{\mathrm{in}}-1})$, and $x_{\mathrm{sp}} \in \mathbb{R}^{d_{\mathrm{sp}}}$ is strongly correlated with the label on source but random noise on target. Formally, we sample $\mathrm{y} \sim \mathrm{Unif}\{-1, 1\}$ and generate inputs $x$ conditioned on y as follows

$$\mathrm{P_S}: \quad x_{\mathrm{in}} \sim \mathcal{N}(\gamma \cdot \mathrm{y} w^\star, \Sigma_{\mathrm{in}}) \quad x_{\mathrm{sp}} = \mathrm{y} \mathbf{1}_{d_{\mathrm{sp}}}$$
$$\mathrm{P_T}: \quad x_{\mathrm{in}} \sim \mathcal{N}(\gamma \cdot \mathrm{y} w^\star, \Sigma_{\mathrm{in}}) \quad x_{\mathrm{sp}} \sim \mathcal{N}(\mathbf{0}, \Sigma_{\mathrm{sp}}), \tag{6}$$

where $\gamma$ is the margin afforded by the invariant feature. We set covariance of the invariant features $\Sigma_{\mathrm{in}} = \sigma_{\mathrm{in}}^2 \cdot (\mathbf{I}_{d_{\mathrm{in}}} - w^\star w^{\star\top})$ to capture structure in the invariant feature that the variance is less along the latent predictive direction $w^\star$. Note that the spurious feature is completely predictive of the label in the source data, and is distributed as spherical Gaussian in the target data with $\Sigma_{\mathrm{sp}} = \sigma_{\mathrm{sp}}^2 \mathbf{I}_{d_{\mathrm{sp}}}$.

**Why is our simplified setup interesting?** In our setup, $x_{\mathrm{in}}$ is the hard to learn feature that generalizes from source to target. The hardness of learning this feature is determined by the value of the margin $\gamma$ and how it compares with size of the spurious feature ($\sqrt{d_{\mathrm{sp}}}$). Since, $\gamma/\sqrt{d_{\mathrm{sp}}}$ is small in our setup, $x_{\mathrm{in}}$ is much harder to learn on source data (even with population access) compared to the spurious feature $x_{\mathrm{sp}}$ which generalizes poorly from source to target. These two types of features have been captured in similar analysis on spurious correlations [71, 60] since it imitates pitfalls emanating from the presence of spurious features in real world datasets (*e.g.*, the easy to learn background feature in image classification problems). While this setup is simple, it is also expressive enough to elucidate both self-training and contrastive learning behaviors we observe in real world settings. Specifically, it captures the separation results we observe in Sec. 3.

**Differences of our setup with prior works.** While our distribution shift settings bears the above similarities it also has important differences with works analyzing self-training and contrastive pretraining individually. Chen et al. [17] analyze the iterative nature of self-training algorithm, where the premise is that we are given a classifier that not only has good performance on source data but in addition does not rely too much on the spurious feature. Under the strong condition of small norms along the spurious feature, they show that self-training can provably unlearn this small dependence when the target data along the spurious feature is random noise. This assumption is clearly violated in setups where the spurious correlation is strong (as in our toy setup), *i.e.*, the dependence on the spurious feature is rather large (much larger than that on the invariant feature) for any classifier that is trained directly on source data. Consequently, we show the need for "good" pretrained representations from contrastive pretraining over which if we train a linear predictor (using source labeled data), it will provably have a reduced "effective" dependence on the spurious feature.

Using an augmentation distribution similar to ours, Saunshi et al. [73] carried out contrastive pretraining analysis with the backbone belonging to a capacity constrained function class (similar analysis also in [40]). Our setup differs from this in two key ways: (i) we specifically consider a distribution

shift from source to target. Unlike their setting, it is not sufficient to make augmentations consistent with ground truth labels, since the predictor that uses just the spurious feature also assigns labels consistent with both ground truth predictions and augmentations on the source data; and (ii) our augmentation distribution assumes no knowledge of the invariant feature, which is why we augment all dimensions uniformly, as opposed to selectively augmenting a set of dimensions. In other words, we assume no knowledge of the structure of the optimal target predictor. For *e.g.*, if we had knowledge of the spurious dimensions we could have just selectively augmented those. Assuming knowledge of these perfect augmentations is not ideal for two reasons: (a) it makes the problem so easy that just training an ERM model on source data with these augmentations would already yield a good target predictor (which rarely happens in practice); and (b) in real-world datasets perfect augmentations for the downstream task are not known. Hence, we stick to generic augmentations in our setup.

## D.2   Discussion on self-training and contrastive learning objectives

| Method | UDA Setup | SSL Setup |
|---|---|---|
| **ERM**: | $h_{\mathrm{erm}} = \arg\min_h \mathbb{E}_{\mathrm{P_S}} \ell(h(x), y)$ | $h_{\mathrm{erm}} = \arg\min_h \frac{1}{n} \sum_{i=1}^n \ell(h(x_i), y_i)$ $\{(x_i, y_i)\}_{i=1}^n \sim \mathrm{P_T}^n$ |
| **ST**: | Starting from $h_{\mathrm{erm}}$ optimize over $h$ (to get $h_{\mathrm{st}}$): $\mathbb{E}_{\mathrm{P_T}(x)} \ell(h(x), \mathrm{sgn}(h(x)))$ | Starting from $h_{\mathrm{erm}}$ optimize over $h$ (to get $h_{\mathrm{st}}$): $\mathbb{E}_{\mathrm{P_T}(x)} \ell(h(x), \mathrm{sgn}(h(x)))$ |
| **CL**: | $\Phi_{\mathrm{cl}} = \arg\min_\phi \mathcal{L}_{\mathrm{cl}}(\Phi)$ Use $(\mathrm{P_S}(x) + \mathrm{P_T}(x))/2$ for $\mathcal{L}_{\mathrm{cl}}(\Phi)$ $h_{\mathrm{cl}} = \arg\min_h \mathbb{E}_{\mathrm{P_S}} \ell(h \circ \Phi_{\mathrm{cl}}(x), y)$ | $\Phi_{\mathrm{cl}} = \arg\min_\phi \mathcal{L}_{\mathrm{cl}}(\Phi)$ Use $\mathrm{P_T}(x)$ for $\mathcal{L}_{\mathrm{cl}}(\Phi)$ $h_{\mathrm{cl}} = \arg\min_h \frac{1}{n} \sum_{i=1}^n \ell(h \circ \Phi_{\mathrm{cl}}(x_i), y_i)$ |
| **STOC**: | Starting from $h_{\mathrm{cl}}$ optimize over $h$ (to get $h_{\mathrm{stoc}}$): $\mathbb{E}_{\mathrm{P_T}(x)} \ell(h \circ \Phi_{\mathrm{cl}}(x), \mathrm{sgn}(h \circ \Phi_{\mathrm{cl}}(x)))$ | Starting from $h_{\mathrm{cl}}$ optimize over $h$ (to get $h_{\mathrm{stoc}}$): $\mathbb{E}_{\mathrm{P_T}(x)} \ell(h \circ \Phi_{\mathrm{cl}}(x), \mathrm{sgn}(h \circ \Phi_{\mathrm{cl}}(x)))$ |

Table 8: **Description of methods for SSL vs. UDA**: For each method we provide exact objectives used for experiments and analysis in the SSL and UDA setups (pertaining to Sec. 4).

In text we will describe our objectives and methods for the UDA setup. In Table 8 we constrast the differences in the methods and objectives for SSL and UDA setups. Recall from Section 2 that we learn linear classifiers $h$ over features extractors $\Phi$. We consider linear feature extractor i.e. $\Phi$ is a matrix in $\mathbb{R}^{k \times d}$. For mathematical convenience, we assume access to infinite unlabeled data and hence replace the empirical quantities over unlabeled data with their population counterpart. In the UDA setting, we further assume access to infinite labeled data from the source. Note that due to distribution shift between source and target, "ERM" on infinite labeled data from the source does not necessarily achieve optimal performance on the target. For binary classification, we assume that the linear layer $h$ maps features to a scalar in $\mathbb{R}$ such that the prediction is $\mathrm{sgn}(h^\top \Phi x)$. We use the exponential loss $\ell(f(x), y) = \exp(-yf(x))$ as the classification loss.

*Contrastive pretraining.* We obtain $\Phi_{\mathrm{cl}} := \arg\min_\Phi \mathcal{L}_{\mathrm{cl}}(\Phi)$ by minimizing the Barlow Twins objective [93], which prior works have shown is also equivalent to spectral contrastive and non-contrastive objectives [33, 11]. In Sec. 4, we consider a constrained form of Barlow Twins in (3) which enforces representations of different augmentations $a_1, a_2$ of the same input $x$ to be close in representation space, while ensuring feature diversity by staying in the constraint set. We assume a strict constraint on regularization ($\rho = 0$) for the theoretical arguments in the rest of the main paper. In App. E.1.2 we prove that all our claims hold for small $\rho$ as well. In (7), we redefine the pretraining objective with a regularization term (instead of a constraint set) where $\kappa$ controls the strength of the regularization term, with higher values of $\kappa$ corresponding to stronger constraints on feature diversity. We then learn a linear classifier $h_{\mathrm{cl}}$ over $\Phi_{\mathrm{cl}}$ to minimize the exponential loss on labeled source data.

$$\mathcal{L}_{\mathrm{cl}}(\Phi) := \mathbb{E}_{x \sim \mathrm{P_U}} \mathbb{E}_{a_1, a_2 \sim \mathrm{P_A}(\cdot|x)} \|\Phi(a_1) - \Phi(a_2)\|_2^2 + \kappa \cdot \left\| \mathbb{E}_{a \sim \mathrm{P_A}} \left[ \Phi(a)\Phi(a)^\top \right] - \mathbf{I}_k \right\|_F^2 \quad (7)$$

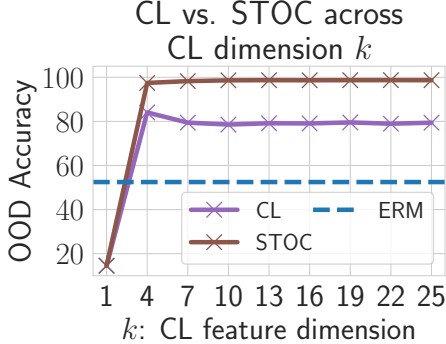
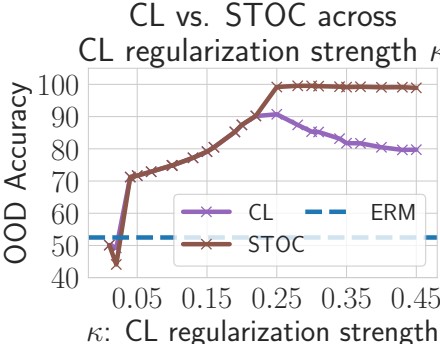

Figure 5: **Ablations on pretraining hyperparameters:** In the UDA setup we plot the performance of CL and STOC as we vary two pretraining hyper-parameters: *(left)* the output dimension ($k$) of the feature extractor $\Phi$; and *(right)* the strength ($\kappa$) of the regularizer in the Barlow Twins objective in (7). While ablating on $k$ we fix $\kappa = 0.5$, and while ablating on $\kappa$ we fix $k = 10$. Other problem parameters are taken from Example 1.

*Augmentations.* Data augmentations play a key role in contrastive pre-training (and also as we see later, state-of-the-art self-training variants like FixMatch). Given input $x \in \mathcal{X}$, let $P_A(a \mid x)$ denote the distribution over its augmentations, and $P_A$ denote the marginal distribution over all possible augmentations. We use the following simple augmentations where we scale the magnitude of each co-ordinate by a uniformly independent amount, *i.e.*,

$$a \sim P_A(\cdot \mid x) \equiv c \odot x \quad \text{where,} \quad c \sim \text{Unif}[0,1]^d. \tag{8}$$

The performance of different methods heavily depends on the assumptions we make on augmentations. We try to mirror practical settings where the augmentations are fairly "generic", not encoding any information about which features are invariant or spurious, and hence perturb all features symmetrically.

*Self-training.* ST performs ERM in the first stage using labeled data from the source, and then subsequently updates the head $h$ by iteratively generating pseudolabels on the unlabeled target:

$$\mathcal{L}_{\text{st}}(h; \Phi) := \mathbb{E}_{P_T(x)} \ell(h^\top \Phi x, \text{sgn}(h^\top \Phi(x))) \qquad \text{Update:} \ \ h^{t+1} = \frac{h^t - \eta \nabla_h \mathcal{L}_{\text{st}}(h^t; \Phi)}{\|h^t - \eta \nabla_h \mathcal{L}_{\text{st}}(h^t; \Phi)\|_2} \tag{9}$$

For convenience, we keep the feature backbone $\Phi$ fixed across the self-training iterations and only update the linear head on the pseudolabels.

*STOC(Self-training after contrastive learning).* Finally, we can combine the two unsupervised objectives where we do the self-training updates( 2) with $h_0 = h_{\text{cl}}$ and $\Phi_0 = \Phi_{\text{cl}}$ starting with the contrastive learning model rather than just source-only ERM. Here, we only update $h$ and fix $\Phi_{\text{cl}}$.

### D.3 Additional empirical results in our simplified setup

We conduct two ablations on the hyperparameters for contrastive pretraining. First, we vary the dimensionality $k$ of the linear feature extractor $\Phi \in \mathbb{R}^{k \times d}$. Second, we vary the regularization strength $\kappa$ that enforces feature diversity in the Barlow Twins objective (7). In Figure 5 we plot these ablations in the UDA setup.

**Varying feature dimension.** We find that CL recovers the full set of predictive features (*i.e.* both spurious and invariant) only when $k$ is large enough (Figure 5*(left)*). Since the dimensionality of the true feature is 5 in our Example 1, reducing $k$ below the true feature dimension hurts CL. Once $k$ crosses a certain threshold, CL features completely capture the projection of the invariant feature $w_{\text{in}}$. After this point, it amplifies the component along $w_{\text{in}}$. It retains the amplification over the spurious feature $w_{\text{sp}}$ even as we increase $k$. This is confirmed by our finding that further increasing $k$ does not hurt CL performance. This is also inline with our theoretical observations, where we find that for

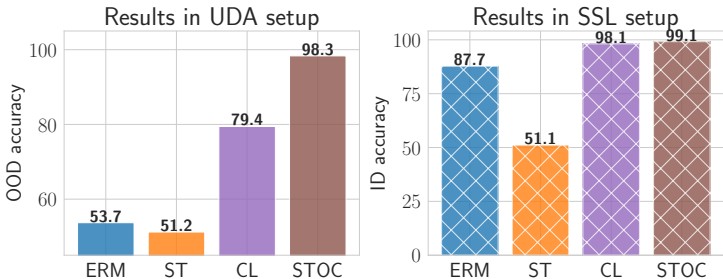

Figure 6: **Results with linear backbone:** We plot the OOD accuracy for ERM, CL, ST and STOC in the UDA setup and ID accuracy in the SSL setup when the feature extractor $\Phi$ is a linear network. Note, that the feature extractor is still fixed during CL and STOC.

suitable $w^\star$, the subspace spanned by $w_{\mathrm{in}}$ and $w_{\mathrm{sp}}$ are contained in a low rank space (as low as rank 2) of the contrastive representations (Theorem 5). Once CL has amplified the dependence along $w_{\mathrm{in}}$ STOC improves over CL by unlearning any remaining dependence on the spurious $w_{\mathrm{sp}}$. The above arguments for the CL trend also explain why the performance of STOC continues to remain $\approx 100\%$ as we vary $k$.

**Varying regularization strength.** In our main theoretical arguments we consider the constrained form of the Barlow Twins objective (3) with a strict constraint of $\rho = 0$ (we relax this theoretically as well, see E.1.2). For our experiments, we optimize the regularized version of this objective (7), where the constraint term now appears as a regularizer which enforces feature diversity, *i.e.* the features learned through contrastive pretraining span orthogonal parts of the input space (as governed under the metric defined by augmentation covariance matrix $\Sigma_A$). If $\kappa$ is very low, then trivial solutions exist for the Barlow Twins objective. For *e.g.*, $\phi \approx \mathbf{0}$ (zero vector) achieves very low invariance loss. When $\kappa < 0.05$, we find that CL recovers these trivial solutions (Figure 5*(right)*). Hence, both CL and STOC perform poorly. As we increase $\kappa$ the performance of both CL and STOC improve, mainly because the features returned by $\Phi_{\mathrm{cl}}$ now comprise of the predictive directions $w_{\mathrm{in}}$ and $w_{\mathrm{sp}}$, as predictive by our theoretical arguments for $\rho = 0$ (which corresponds to large $\kappa$). On the other hand, when $\kappa$ is too high optimization becomes hard since $\kappa$ directly effects the Lipschitz constant of the loss function. Hence, the performance of CL drops by some value. Note that this does not effect the performance of STOC since CL continues to amplify $w_{\mathrm{in}}$ over $w_{\mathrm{sp}}$ even if it is returning suboptimal solutions with respect to the optimization loss of the pretraining objective.

### D.4 Reconciling Practice: Experiments with deep networks in toy setup

In this section we delve into the details of Sec. 4.5, *i.e.*, we analyze performance of different methods when we make some design choices that imitate practice. First, we look at experiments involving a deep non-linear backbone $\Phi$. Here, the non-linear $\Phi$ is learned during contrastive pretraining and

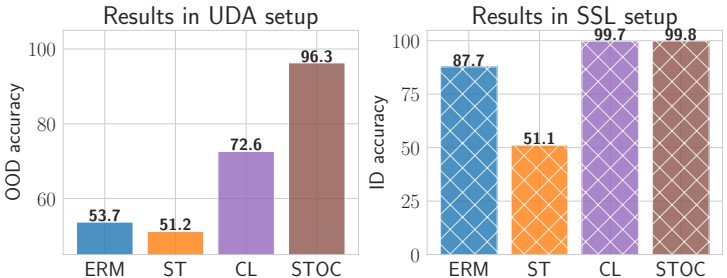

Figure 7: **Results with non-linear backbone:** We plot the OOD accuracy for ERM, CL, ST and STOC in the UDA setup and ID accuracy in the SSL setup when the feature extractor $\Phi$ is a non-linear one-hidden layer network with ReLU activations. Note, that the feature extractor is still fixed during CL and STOC.

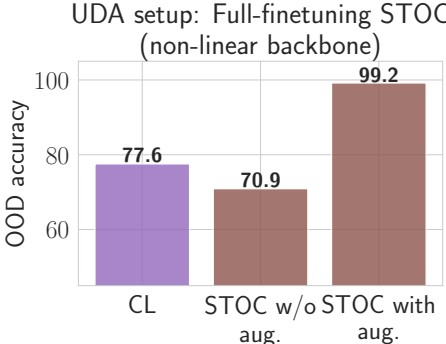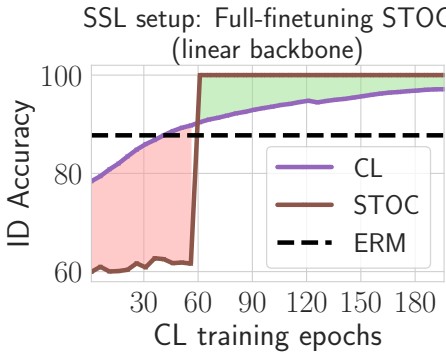

Figure 8: **Finetuning the contrastive representations during STOC:** We propagate gradients to the feature backbone $\Phi$ when running STOC algorithm. Note that CL still fixes the contrastive representations when learning a fixed linear head over it. On the *(left)* we show results in UDA setup where we compare the performance of STOC with and without augmentations (along with other practical design choices like confidence thresholds and continuing to optimize source loss as done in FixMatch) when the feature backbone is non-linear. On the *(right)* we show results for STOC and CL in the SSL setup when the feature backbone is linear.

fixed for CL and STOC. Then, we investigate trends when we continue to propagate gradients onto $\Phi$ during STOC (we call this full-finetuning). Unlike previous cases, this allows features to be updated.

**Results with non-linear feature extractor $\Phi$.** In Fig. 7 we plot the performance of the four methods when we use a non-linear feature extractor during contrastive pretraining. This feature extractor is a one-hidden layer neural network (hidden dimension is 500) with ReLU activations. We find that the trends observed with linear backbones in Fig. 6 are also replicated with the non-linear one. Specifically, we note that STOC improves over CL under distribution shifts, whereas CL is already close to optimal when there are no distribution shifts. We also see that CL and ST individually are subpar. In SSL, we see a huge drop in the performance of ST (over ERM) mainly because we only fit on pseudolabels during ST. This is different from practice where we continue to optimize loss on labeled data points while fitting the pseudolabels. Consequently, when we continue to optimize performance on source labeled data the performance of ST in SSL setup is improves from $51.1\% \rightarrow 72.6\%$.

**Results with full fine-tuning.** Up till this point, we have only considered the case (for both SSL and UDA) where we fix the contrastive learned features when running CL and STOC, *i.e.*, we only optimized the linear head $h$. Now, we shall consider the setting where gradients are propagated to $\Phi$ during STOC. Note that we still fix the representations for training the linear head during CL. Results for this setting are in Figure 8. We show two interesting trends that imitate real world behaviors.

*STOC benefits from augmentations during full-finetuning:* In the UDA setup we find that ST while updating $\Phi_{cl}$ can hurt due to overfitting issues when training with the finite sample of labeled and unlabeled data (drop by $> 7\%$ over CL). This is due to overfitting on confident but incorrect pseudolabels on target data. This can exacerbate components along spurious feature $w_{sp}$ from source. One reasoning behind this is that deep neural networks can perfectly memorize them on finite unlabeled target data [94]. Heuristics typically used in practice (*e.g.* in FixMatch [79]) help avoid overfitting on incorrect pseudolabels: (i) confidence thresholding; to pick confident pseudolabel examples; (ii) pseudolabel a different augmented input than the one on which the self-training loss is optimized; and (iii) optimize source loss with labeled data simultaneously when fitting pseudolabels. Intuitively, thresholding introduces a curriculum where we only learn confident examples in the beginning whose pseudolabels are mainly determined by component along the invariant feature $w_{in}$. Augmentations prevent the neural network from memorizing incorrect pseudolabels and optimizing source loss prevents forgetting of features learned during CL. When we implement these during full-finetuning in STOC we see that STOC now improves over CL (by $> 20\%$).

*Can we improve contrastive pretraining features during STOC?* We find that self-training can also improve features learned during contrastive pretraining when we update the full backbone during STOC (see Figure 8*(right)*). Specifically, in the SSL setup we find that STOC can now improve substantially over CL. Recall, that when we fixed $\Phi_{\text{cl}}$ this was not possible (see E.3 and Fig. 2(b)). This is mainly because STOC can now improve performance beyond just recovering the generalization gap for the linear head (which is typically small). This feature improvement is observed even when we fully finetune a linear feature extractor. Similar trends are also observed with the non-linear backbone. But, it becomes harder to identify a good stopping criterion for CL training. Thus, it remains unclear if STOC and CL have complementary benefits for feature learning in UDA or SSL settings. Investigating this is an interesting avenue for future work.

# E   Formal Statements from Sec. 4

Recall from Section 2 that we learn linear classifiers $h$ over features extractors $\Phi$. We consider linear feature extractor i.e. $\Phi$ is a matrix in $\mathbb{R}^{d \times k}$ and the linear layer $h : \mathbb{R}^k \to \mathbb{R}$ with a prediction as $\text{sgn}(h^\top \Phi x)$. We use the exponential loss $\ell(f(x), y) = \exp(-yf(x))$.

## E.1   Analysis of ERM and ST: Formal Statement of Theorem 2

For ERM and ST, we train both $h$ and $\Phi$. This is equivalent to $\Phi = I_{d \times d}$ being identity and training a linear head $h$. Recall that the ERM classifier is obtained by minimizing the population loss on labeled source data:

$$h_{\text{ERM}} = \arg\min_h \mathbb{E}_{(x,y) \sim P_S} [\ell(x, y)] . \tag{10}$$

We split Theorem 2 into Theorem 8 and Theorem 9. Before we characterize the ERM solution, we recall some additional notation. Define $w_{\text{in}} = [w^\star, 0, ..., 0]^\top$, and $w_{\text{sp}} = [0, ..., 0, \mathbf{1}_{d_{\text{sp}}}/\sqrt{d_{\text{sp}}}]^\top$. The following proposition characterizes $h_{\text{ERM}}$ and 0-1 error of the classifier on target:

**Theorem 8** (ERM classifier and its error on target). *ERM classifier obtained as in* (10) *is given by*

$$\frac{h_{ERM}}{\|h_{ERM}\|_2} = \frac{\gamma \cdot w_{\text{in}} + \sqrt{d_{\text{sp}}} \cdot w_{\text{sp}}}{\sqrt{\gamma^2 + d_{\text{sp}}}} .$$

*The target accuracy of $h_{ERM}$ is given by* $0.5 \cdot \text{erfc}\left(-\gamma^2/(\sqrt{2d_{\text{sp}}} \cdot \sigma_{\text{sp}})\right)$.

*Proof.* To prove this theorem, we first derive a closed-form expression for the ERM classifier and then use Lemma 29 to derive its 0-1 error on target. For Gaussian data with the same covariance matrices for class conditional $P_S(x|y = 1)$ and $P_S(x|y = 0)$, Bayes decision rule is given by the Fisher's linear discriminant direction (Chapter 4; Bishop [9]):

$$h(x) = \begin{cases} 1, & \text{if } h^\top x > 0 \\ 0, & \text{otherwise} \end{cases}$$

where $h = 2 \cdot \gamma(w_{\text{in}}) + 2 \cdot \sqrt{d_{\text{sp}}}(w_{\text{sp}})$. Plugging $h$ in Lemma 29 we get the desired result. $\square$

ST performs ERM in the first stage using labeled data from the source, and then subsequently updates the head $h$ by iteratively generating pseudolabels on the unlabeled target:

$$\mathcal{L}_{\text{st}}(h) := \mathbb{E}_{P_T(x)} \ell(h^\top x, \text{sgn}(h^\top x)) . \tag{11}$$

Starting with $h_{\text{ST}}^0 = h_{\text{ERM}}/\|h_{\text{ERM}}\|_2$ (the classifier obtained with ERM) we perform the following iterative procedure for self-training:

$$h_{\text{ST}}^{t+1} = \frac{h_{\text{ST}}^t - \eta \nabla_h \mathcal{L}_{\text{st}}(h_{\text{ST}}^t)}{\|h_{\text{ST}}^t - \eta \nabla_h \mathcal{L}_{\text{st}}(h_{\text{ST}}^t)\|_2} \tag{12}$$

Next, we characterize ST solution:

**Theorem 9** (ST classifier and its error on target). *Starting with ERM solution, ST will lead to:*

(i) (Necessary condition) $h_{ST}^t = w_{\text{sp}}$ as $t \to \infty$, such that the target accuracy is 50% for all $\sigma_{\text{sp}} \geqslant 1$ and $\gamma \leqslant \frac{1}{2\sqrt{\sigma_{\text{sp}}}}$.

(ii) (Sufficient condition) $h_{ST}^t = w_{\text{in}}$ as $t \to \infty$, such that the target accuracy is 100% when the problem parameters $\gamma, \sigma_{\text{sp}}$ satisfy: $\gamma \geqslant \sigma_{\text{sp}}$.

*Proof.* The proof can be divided into two parts: (i) deriving closed-form expressions for updates on $h_{ST}^t$ in terms of $h_{ST}^{t-1}$ and (ii) obtaining conditions under which the component along $w_{\text{in}}$ monotonically increases or decreases with $t$ after re-normalizing the norm of updated $h$. For notation convenience, we denote $h_{ST}$ with $h$ in the rest of the proof.

**Part-1.** First, the loss of self-training with classifier $h := [h_{\text{in}}, h_{\text{sp}}]$ where $h_{\text{in}} \in \mathbb{R}^{d_{\text{in}}}$ and $h_{\text{sp}} \in \mathbb{R}^{d_{\text{sp}}}$ is given by:

$$\mathcal{L}_{\text{st}}(h) = \mathbb{E}_{\mathrm{P}_{\mathsf{T}}(x)}\left[\ell(h^\top x, \text{sgn}(h^\top x))\right] \tag{13}$$

$$= \mathbb{E}_{\mathrm{P}_{\mathsf{T}}(x)}\left[\exp\left(-\text{sign}(h^\top x) \cdot (h^\top x)\right)\right] \tag{14}$$

$$= \mathbb{E}_{\mathrm{P}_{\mathsf{T}}(x)}\left[\exp\left(-\left|h^\top x\right|\right)\right] \tag{15}$$

$$= \mathbb{E}_{\mathrm{P}_{\mathsf{T}}(x)}\left[\exp\left(-\left|h_{\text{in}}^\top x_{\text{in}} + h_{\text{sp}}^\top x_{\text{sp}}\right|\right)\right] \tag{16}$$

$$= \mathbb{E}_{y \sim \mathcal{U}\{-1,1\}, z \sim \mathcal{N}(0,1)}\left[\exp\left(-\left|\gamma \cdot y \cdot h_{\text{in}}^\top w^\star \right.\right.\right.$$
$$\left.\left.\left. + \left[\sigma_{\text{in}}(\|h_{\text{in}}\|_2^2 - (h_{\text{in}}^T w^\star)^2) + \sigma_{\text{sp}} \cdot \|h_{\text{sp}}\|_2\right] \cdot z\right|\right)\right]. \tag{17}$$

$$= \mathbb{E}_{z \sim \mathcal{N}(0,1)}\left[\exp\left(-\left|\gamma \cdot h_{\text{in}}^\top w^\star + \left[\sigma_{\text{in}}(\|h_{\text{in}}\|_2^2 - (h_{\text{in}}^T w^\star)^2) + \sigma_{\text{sp}} \cdot \|h_{\text{sp}}\|_2\right] \cdot z\right|\right)\right], \tag{18}$$

where (16) to (17) is implied by simply replacing the definition of target distribution and (17) to (18) is implied by the symmetry of the function with respect to $y$ and $-y$ due to the symmetry of the absolute function and Gaussian distribution. For a classifier $h^t$, we denote $\mu_t = \gamma \cdot h_{\text{in}}^t{}^\top w^\star$ and $\sigma_t = \left[\sigma_{\text{in}}(\|h_{\text{in}}^t\|_2^2 - (h_{\text{in}}^t{}^T w^\star)^2) + \sigma_{\text{sp}} \cdot \|h_{\text{sp}}^t\|_2\right]$. With this notation, we can re-write the loss in (18) as $\mathcal{L}_{\text{st}}(h^t) = \mathbb{E}_{z \sim \mathcal{N}(0,\sigma_t^2)}\left[\exp\left(-\left|\mu_t + z\right|\right)\right]$.

Now we derive a closed-form expression of $\mathcal{L}_{\text{st}}(h^t)$ in Lemma 30:

$$\mathcal{L}_{\text{st}}(h^t) = \frac{1}{2}\left(\exp\left(\frac{\sigma_t^2}{2} - \mu_t\right) \cdot \text{erfc}\left(-\frac{\mu_t}{\sqrt{2}\sigma_t} + \frac{\sigma_t}{\sqrt{2}}\right) + \exp\left(\frac{\sigma_t^2}{2} + \mu_t\right) \cdot \text{erfc}\left(\frac{\mu_t}{\sqrt{2}\sigma_t} + \frac{\sigma_t}{\sqrt{2}}\right)\right). \tag{19}$$

Define the Mill's ratio as $\text{r}(x) = \exp\left(x^2/2\right) \cdot \text{erfc}\left(x/\sqrt{2}\right) \cdot \sqrt{\pi/2}$ as in Baricz [5]. We will frequently use standard properties of the Mill's ratio. We list them in Lemma 21 for completeness. Define:

$$\alpha_1(\mu_t, \sigma_t) = -\exp\left(\frac{\sigma_t^2}{2} - \mu_t\right) \cdot \text{erfc}\left(-\frac{\mu_t}{\sqrt{2}\sigma_t} + \frac{\sigma_t}{\sqrt{2}}\right) + \exp\left(\frac{\sigma_t^2}{2} + \mu_t\right) \cdot \text{erfc}\left(\frac{\mu_t}{\sqrt{2}\sigma_t} + \frac{\sigma_t}{\sqrt{2}}\right),$$

$$= \sqrt{\frac{2}{\pi}}\exp\left(-\frac{\mu_t^2}{2\sigma_t^2}\right)\left[\text{r}\left(\sigma_t + \frac{\mu_t}{\sigma_t}\right) - \text{r}\left(\sigma_t - \frac{\mu_t}{\sigma_t}\right)\right] \tag{20}$$

$$\alpha_2(\mu_t, \sigma_t) = \exp\left(\frac{\sigma_t^2}{2} - \mu_t\right) \cdot \text{erfc}\left(-\frac{\mu_t}{\sqrt{2}\sigma_t} + \frac{\sigma_t}{\sqrt{2}}\right) + \exp\left(\frac{\sigma_t^2}{2} + \mu_t\right) \cdot \text{erfc}\left(\frac{\mu_t}{\sqrt{2}\sigma_t} + \frac{\sigma_t}{\sqrt{2}}\right)$$

$$- \frac{2\sqrt{2}}{\sigma_t \sqrt{\pi}}\exp\left(-\frac{\mu_t^2}{2\sigma_t^2}\right)$$

$$= \sqrt{\frac{2}{\pi}}\exp\left(-\frac{\mu_t^2}{2\sigma_t^2}\right)\left[\text{r}\left(\sigma_t + \frac{\mu_t}{\sigma_t}\right) + \text{r}\left(\sigma_t - \frac{\mu_t}{\sigma_t}\right) - \frac{2}{\sigma_t}\right]. \tag{21}$$

Let $\widetilde{h}^{t+1}$ denote the un-normalized gradient descent update at iterate $t + 1$. We have:

$$\widetilde{h}^{t+1} = h^t - \eta \cdot \frac{\partial \mathcal{L}_{\text{st}}(h^t)}{\partial h}. \tag{22}$$

Now we will individually argue about the update of $\widetilde{h}^{t+1}$ along the first $d_{\mathrm{in}}$ dimensions and the last $d_{\mathrm{sp}}$ dimensions. First, we have:

$$
\begin{aligned}
\widetilde{h}_{\mathrm{in}}^{t+1} &= h_{\mathrm{in}}^{t} - \eta \cdot \frac{\partial \mathcal{L}_{\mathrm{st}}(h^t)}{\partial h_{\mathrm{in}}} \\
&= h_{\mathrm{in}}^{t} - \frac{\eta}{2}\left(-\exp\left(\frac{\sigma_t^2}{2} - \mu_t\right) \cdot \mathrm{erfc}\left(-\frac{\mu_t}{\sqrt{2}\sigma_t} + \frac{\sigma_t}{\sqrt{2}}\right)\right. \\
&\qquad\qquad\left. +\exp\left(\frac{\sigma_t^2}{2} + \mu_t\right) \cdot \mathrm{erfc}\left(\frac{\mu_t}{\sqrt{2}\sigma_t} + \frac{\sigma_t}{\sqrt{2}}\right)\right) \cdot \gamma \cdot w^\star \\
&\qquad - \frac{\eta}{2}\left(\exp\left(\frac{\sigma_t^2}{2} - \mu_t\right) \cdot \mathrm{erfc}\left(-\frac{\mu_t}{\sqrt{2}\sigma_t} + \frac{\sigma_t}{\sqrt{2}}\right)\right. \\
&\qquad\qquad +\exp\left(\frac{\sigma_t^2}{2} + \mu_t\right) \cdot \mathrm{erfc}\left(\frac{\mu_t}{\sqrt{2}\sigma_t} + \frac{\sigma_t}{\sqrt{2}}\right) \\
&\qquad\qquad\left. -\frac{2\sqrt{2}}{\sigma_t\sqrt{\pi}}\exp\left(-\frac{\mu_t^2}{2\sigma_t^2}\right)\right) \cdot (2h_{\mathrm{in}}^{t} - 2(h_{\mathrm{in}}^{t\top}w^\star)w^\star) \cdot \sigma_{\mathrm{in}}^2 \\
&= h_{\mathrm{in}}^{t} - \frac{\eta}{2} \cdot \alpha_1(\mu_t,\sigma_t) \cdot \gamma \cdot w^\star - \frac{\eta}{2} \cdot \alpha_2(\mu_t,\sigma_t) \cdot (2h_{\mathrm{in}}^{t} - 2(h_{\mathrm{in}}^{t\top}w^\star)w^\star) \cdot \sigma_{\mathrm{in}}^2 . \qquad (23)
\end{aligned}
$$

Notice that the update of $h_{\mathrm{in}}^{t+1}$ is split into two components, one along $w^\star$ and the other along the orthogonal component $2h_{\mathrm{in}}^{t} - 2(h_{\mathrm{in}}^{t\top}w^\star)w^\star$. We will now argue that since at initialization, the component along $(I - w^\star w^{\star\top})$ is zero then it will remain zero. In particular, we have:

$$
h_{\mathrm{in}}^{0\top}(I - w^\star w^{\star\top}) \propto w^{\star\top}(I - w^\star w^{\star\top}) = 0 . \qquad (24)
$$

With (23), we can argue that if $(I - w^\star w^{\star\top})h_{\mathrm{in}}^{t} = 0$, then $(I - w^\star w^{\star\top})\widetilde{h}_{\mathrm{inv}}^{t+1} = 0$ implying that $(I - w^\star w^{\star\top})\widetilde{h}_{\mathrm{in}}^{t} = 0$ for all $t > 0$. Hence, we have:

$$
\begin{aligned}
\widetilde{h}_{\mathrm{inv}}^{t+1} &= h_{\mathrm{in}}^{t} - \eta \cdot \frac{\partial \mathcal{L}_{\mathrm{st}}(h^t)}{\partial h_{\mathrm{in}}} \\
&= h_{\mathrm{in}}^{t} - \frac{\eta}{2} \cdot \alpha_1(\mu_t,\sigma_t) \cdot \gamma \cdot w^\star . \qquad (25)
\end{aligned}
$$

Second, we have the update $\widetilde{h}_{\mathrm{sp}}^{t+1}$ given by:

$$
\begin{aligned}
\widetilde{h}_{\mathrm{sp}}^{t+1} &= h_{\mathrm{sp}}^{t} - \eta \cdot \frac{\partial \mathcal{L}_{\mathrm{st}}(h^t)}{\partial h_{\mathrm{sp}}} \\
&= h_{\mathrm{sp}}^{t} - \frac{\eta}{2}\left(\exp\left(\frac{\sigma_t^2}{2} - \mu_t\right) \cdot \mathrm{erfc}\left(-\frac{\mu_t}{\sqrt{2}\sigma_t} + \frac{\sigma_t}{\sqrt{2}}\right)\right. \\
&\qquad\qquad\left. +\exp\left(\frac{\sigma_t^2}{2} + \mu_t\right) \cdot \mathrm{erfc}\left(\frac{\mu_t}{\sqrt{2}\sigma_t} + \frac{\sigma_t}{\sqrt{2}}\right) - \frac{2\sqrt{2}}{\sigma_t\sqrt{\pi}}\exp\left(-\frac{\mu_t^2}{2\sigma_t^2}\right)\right) \cdot h_{\mathrm{sp}}^{t} \cdot \sigma_{\mathrm{sp}}^2 \\
&= h_{\mathrm{sp}}^{t} - \frac{\eta}{2} \cdot \alpha_2(\mu_t,\sigma_t) \cdot h_{\mathrm{sp}}^{t} \cdot \sigma_{\mathrm{sp}}^2 . \qquad (26)
\end{aligned}
$$

Re-writing the expressions (25) and (26) for the update of $\widetilde{h}^{t+1}$, we have:

$$
\widetilde{h}_{\mathrm{in}}^{t+1} = h_{\mathrm{in}}^{t}\left(1 - \frac{\eta}{2} \cdot \alpha_1(\mu_t,\sigma_t) \cdot \gamma^2/\mu_t\right) . \qquad (27)
$$

$$
\widetilde{h}_{\mathrm{sp}}^{t+1} = h_{\mathrm{sp}}^{t}\left(1 - \frac{\eta}{2} \cdot \alpha_2(\mu_t,\sigma_t) \cdot \sigma_{\mathrm{sp}}^2\right) . \qquad (28)
$$

Here, we replace $h_{\mathrm{sp}}^{t} = \mu_t \cdot w^\star/\gamma$ in (25) to get (27). Updates in (27) and (28) show that $\widetilde{h}_{\mathrm{inv}}^{t+1}$ remains in the direction of $h_{\mathrm{in}}^{t}$ and $\widetilde{h}_{\mathrm{sp}}^{t+1}$ remains in the direction of $h_{\mathrm{sp}}^{t}$.

**Part-2.** Now we will derive conditions under which $h_{\text{in}}^t$ and $h_{\text{sp}}^t$ will show monotonic behavior for necessary and sufficient conditions. We will first argue the condition under which ST will provably fail and converge to a classifier with a random target performance. For this, at every $t$, if we have:

$$\frac{\left\|\widetilde{h}_{\text{sp}}^{t+1}\right\|_2}{\left\|\widetilde{h}^{t+1}\right\|_2} > \left\|h_{\text{sp}}^t\right\|_2 , \tag{29}$$

then we can argue that as $t \to \infty$, we have $\left\|h_{\text{sp}}^t\right\|_2 = 1$ and hence, the ST classifier will have random target performance. Thus, we will focus on conditions, under which the norm on $\left\|h_{\text{sp}}^t\right\|_2$ increases with $t$. Re-writing (29), we have:

$$\left\|\widetilde{h}_{\text{sp}}^{t+1}\right\|_2 > \left\|\widetilde{h}^{t+1}\right\|_2 \cdot \left\|h_{\text{sp}}^t\right\|_2 \tag{30}$$

$$\left\|\widetilde{h}_{\text{sp}}^{t+1}\right\|_2 > \left(\left\|\widetilde{h}_{\text{sp}}^{t+1}\right\|_2 + \left\|\widetilde{h}_{\text{in}}^{t+1}\right\|_2\right) \cdot \left\|h_{\text{sp}}^t\right\|_2 \tag{31}$$

$$\left\|\widetilde{h}_{\text{sp}}^{t+1}\right\|_2 \cdot \left(1 - \left\|h_{\text{sp}}^t\right\|_2\right) > \left\|\widetilde{h}_{\text{in}}^{t+1}\right\|_2 \cdot \left\|h_{\text{sp}}^t\right\|_2 \tag{32}$$

$$\frac{\left\|\widetilde{h}_{\text{sp}}^{t+1}\right\|_2}{\left\|h_{\text{sp}}^t\right\|_2} > \frac{\left\|\widetilde{h}_{\text{in}}^{t+1}\right\|_2}{\left\|h_{\text{in}}^t\right\|_2} . \tag{33}$$

Plugging in (27) and (28) into (33), we get:

$$\left|1 - \frac{\eta}{2} \cdot \alpha_2(\mu_t, \sigma_t) \cdot \sigma_{\text{sp}}^2\right| > \left|1 - \frac{\eta}{2} \cdot \alpha_1(\mu_t, \sigma_t) \cdot \gamma^2/\mu_t\right| . \tag{34}$$

For small enough $\eta$, we have the necessary condition for the failure of ST as:

$$\alpha_2(\mu_t, \sigma_t) \cdot \sigma_{\text{sp}}^2 < \alpha_1(\mu_t, \sigma_t) \cdot \gamma^2/\mu_t . \tag{35}$$

Now we show in Lemma 11 and Lemma 10 that if the conditions assumed in the theorem continue to hold, then we can success and failure respectively.

$\square$

**Lemma 10** (Necessary conditions for ST). *Define $\alpha_1$ and $\alpha_2$ as in* (20) *and* (21) *respectively. If $\sigma_{\text{sp}} \geqslant 1$ and $\gamma \leqslant \frac{1}{2\sqrt{\sigma_{\text{sp}}}}$, then we have for all $t$:*

$$\alpha_2(\mu_t, \sigma_t) \cdot \frac{\sigma_{\text{sp}}^2 \cdot \mu_t}{\gamma^2} \leqslant \alpha_1(\mu_t, \sigma_t) . \tag{36}$$

*Proof.* We upper bound and lower bound $\alpha_1$ and $\alpha_2$ by using the properties of $r(\cdot)$. Recall:

$$\alpha_1(\mu_t, \sigma_t) = \sqrt{\frac{2}{\pi}} \exp\left(-\frac{\mu_t^2}{2\sigma_t^2}\right) \left[r\left(\sigma_t + \frac{\mu_t}{\sigma_t}\right) - r\left(\sigma_t - \frac{\mu_t}{\sigma_t}\right)\right] . \tag{37}$$

and

$$\alpha_2(\mu_t, \sigma_t) = \sqrt{\frac{2}{\pi}} \exp\left(-\frac{\mu_t^2}{2\sigma_t^2}\right) \left[r\left(\sigma_t + \frac{\mu_t}{\sigma_t}\right) + r\left(\sigma_t - \frac{\mu_t}{\sigma_t}\right) - \frac{2}{\sigma_t}\right] . \tag{38}$$

We now use Taylor's expansion on $r(\cdot)$ and we get:

$$r(\sigma_t) + r'(\sigma_t) \cdot \left(\frac{\mu_t}{\sigma_t}\right) \leqslant r\left(\sigma_t + \frac{\mu_t}{\sigma_t}\right) \leqslant r(\sigma_t) + r'(\sigma_t) \cdot \left(\frac{\mu_t}{\sigma_t}\right) + R''\left(\frac{\mu_t}{\sigma_t}\right)^2 \tag{39}$$

and similarly, we get:

$$r(\sigma_t) - r'(\sigma_t) \cdot \left(\frac{\mu_t}{\sigma_t}\right) \leqslant r\left(\sigma_t - \frac{\mu_t}{\sigma_t}\right) \leqslant r(\sigma_t) - r'(\sigma_t) \cdot \left(\frac{\mu_t}{\sigma_t}\right) + R''\left(\frac{\mu_t}{\sigma_t}\right)^2 \tag{40}$$

where $R'' = \mathrm{r}''\left(\sigma_0\right)$. This is because $\mathrm{r}''\left(\cdot\right)$ takes positive values and is a decreasing function in $\sigma_t$ (refer to Lemma 21). We now lower bound $\alpha_1(\mu_t, \sigma_t)$ and upper bound $\alpha_2(\mu_t, \sigma_t)$:

$$\frac{\alpha_1(\mu_t, \sigma_t)}{\sqrt{\frac{2}{\pi}}\exp\left(-\frac{\mu_t^2}{2\sigma_t^2}\right)} \geqslant 2\mathrm{r}'\left(\sigma_t\right) \cdot \left(\frac{\mu_t}{\sigma_t}\right) - R''\left(\frac{\mu_t}{\sigma_t}\right)^2 \tag{41}$$

$$\frac{\alpha_2(\mu_t, \sigma_t)}{\sqrt{\frac{2}{\pi}}\exp\left(-\frac{\mu_t^2}{2\sigma_t^2}\right)} \leqslant 2\mathrm{r}\left(\sigma_t\right) + 2 \cdot R''\left(\frac{\mu_t}{\sigma_t}\right)^2 \tag{42}$$

Substituting the lower bound and upper bound in (36) gives us the following as stricter a necessary condition (i.e., (43) implies (36)):

$$\left[2\mathrm{r}\left(\sigma_t\right) + 2 \cdot R''\left(\frac{\mu_t}{\sigma_t}\right)^2 - \frac{2}{\sigma_t}\right] \cdot \frac{\sigma_{\mathrm{sp}}^2 \cdot \mu_t}{\gamma^2} \leqslant 2\mathrm{r}'\left(\sigma_t\right) \cdot \left(\frac{\mu_t}{\sigma_t}\right) - R''\left(\frac{\mu_t}{\sigma_t}\right)^2 \tag{43}$$

$$\iff \left[2\mathrm{r}\left(\sigma_t\right) + 2 \cdot R''\left(\frac{\mu_t}{\sigma_t}\right)^2 - \frac{2}{\sigma_t}\right] \cdot \frac{\sigma_{\mathrm{sp}}^2}{\gamma^2} \leqslant 2\mathrm{r}'\left(\sigma_t\right) \cdot \left(\frac{1}{\sigma_t}\right) - R''\left(\frac{\mu_t}{\sigma_t^2}\right) \tag{44}$$

$$\iff \left[\mathrm{r}\left(\sigma_t\right) + R''\left(\frac{\mu_t}{\sigma_t}\right)^2 - \frac{1}{\sigma_t}\right] \cdot \frac{\sigma_{\mathrm{sp}}^2}{\gamma^2} \leqslant \mathrm{r}\left(\sigma_t\right) - \frac{1}{\sigma_t} - \frac{R''}{2}\left(\frac{\mu_t}{\sigma_t^2}\right) \tag{45}$$

$$\iff \left[R''\left(\frac{\mu_t}{\sigma_t}\right)^2\right] \cdot \frac{\sigma_{\mathrm{sp}}^2}{\gamma^2} + \frac{R''}{2}\left(\frac{\mu_t}{\sigma_t^2}\right) \leqslant \left(\mathrm{r}\left(\sigma_t\right) - \frac{1}{\sigma_t}\right) \cdot \left(1 - \frac{\sigma_{\mathrm{sp}}^2}{\gamma^2}\right) \tag{46}$$

$$\iff \left[R''\left(\frac{\mu_t^2}{\sigma_t}\right)\right] \cdot \frac{\sigma_{\mathrm{sp}}^2}{\gamma^2} + \frac{R''}{2}\left(\frac{\mu_t}{\sigma_t}\right) \leqslant \left(\sigma_t\mathrm{r}\left(\sigma_t\right) - 1\right) \cdot \left(1 - \frac{\sigma_{\mathrm{sp}}^2}{\gamma^2}\right) \tag{47}$$

Now, we will argue the monotonicity of LHS and RHS in (47). Observe that LHS is increasing in $\mu_t$ and decreasing in $\sigma_t$ and RHS is decreasing in $\sigma_t$ as $\left(\sigma_t\mathrm{r}\left(\sigma_t\right) - 1\right)$ is increasing (and the multiplier is negative). Moreover, if (47) holds true for maximum value of RHS and minimum of LHS, then we would have (36). Thus substituting $\mu_t = \gamma$ and $\sigma_t = \sigma_0$ in LHS and $\sigma_t = \sigma_{\mathrm{sp}}$ in RHS, we get:

$$\left[R''\left(\frac{\gamma^2}{\sigma_0}\right)\right] \cdot \frac{\sigma_{\mathrm{sp}}^2}{\gamma^2} + \frac{R''}{2}\left(\frac{\gamma}{\sigma_0}\right) \leqslant \left(\sigma_{\mathrm{sp}}\mathrm{r}\left(\sigma_{\mathrm{sp}}\right) - 1\right) \cdot \left(1 - \frac{\sigma_{\mathrm{sp}}^2}{\gamma^2}\right) \tag{48}$$

$$\iff R'' \cdot \frac{\sigma_{\mathrm{sp}}^2}{\sigma_0} + \frac{R''}{2}\left(\frac{\gamma}{\sigma_0}\right) \leqslant \left(\sigma_{\mathrm{sp}}\mathrm{r}\left(\sigma_{\mathrm{sp}}\right) - 1\right) \cdot \left(1 - \frac{\sigma_{\mathrm{sp}}^2}{\gamma^2}\right) \tag{49}$$

$$\tag{50}$$

Taking $\gamma \leqslant \frac{1}{2\sqrt{\sigma_{\mathrm{sp}}}}$ and substituting $R'' = \mathrm{r}''\left(\sigma_0\right)$:

$$(5/4) \cdot \mathrm{r}''\left(\sigma_0\right) \cdot \sigma_{\mathrm{sp}} \leqslant \left(\sigma_{\mathrm{sp}}\mathrm{r}\left(\sigma_{\mathrm{sp}}\right) - 1\right) \cdot \left(1 - 4 \cdot \sigma_{\mathrm{sp}}^3\right) \tag{51}$$

Analytically solving the above expression, we get that (51) is satisfied for all values of $\sigma_{\mathrm{sp}} \geqslant 1$ when $d_{\mathrm{sp}} \geqslant 1$. For example, the expression in (51) is also satisfied for the problem parameter used in the running example of the main paper.

$\square$

As a remark, we note that in the proof of Lemma 10, the conditions derived are loose because of the relaxations made to simply the proof. In principle, the proof (and hence the conditions) can be tightened by carefully propagating second-order terms (which depend on $\sigma_t$) in (40).

**Lemma 11** (Sufficiency conditions for ST). *Define $\alpha_1$ and $\alpha_2$ as in* (20) *and* (21) *respectively. If $\sigma_{\mathrm{sp}} \leqslant \gamma$, then we have for all $t$:*

$$\alpha_2(\mu_t, \sigma_t) \cdot \frac{\sigma_{\mathrm{sp}}^2 \cdot \mu_t}{\gamma^2} \geqslant \alpha_1(\mu_t, \sigma_t). \tag{52}$$

*Proof.* We upper bound and lower bound $\alpha_1$ and $\alpha_2$ by using the properties of $\mathrm{r}\,(\cdot)$. Recall:

$$\alpha_1(\mu_t, \sigma_t) = \sqrt{\frac{2}{\pi}} \exp\left(-\frac{\mu_t^2}{2\sigma_t^2}\right) \left[\mathrm{r}\left(\sigma_t + \frac{\mu_t}{\sigma_t}\right) - \mathrm{r}\left(\sigma_t - \frac{\mu_t}{\sigma_t}\right)\right]. \tag{53}$$

and

$$\alpha_2(\mu_t, \sigma_t) = \sqrt{\frac{2}{\pi}} \exp\left(-\frac{\mu_t^2}{2\sigma_t^2}\right) \left[\mathrm{r}\left(\sigma_t + \frac{\mu_t}{\sigma_t}\right) + \mathrm{r}\left(\sigma_t - \frac{\mu_t}{\sigma_t}\right) - \frac{2}{\sigma_t}\right]. \tag{54}$$

We now use Taylor's expansion on $\mathrm{r}\,(\cdot)$ and we get:

$$\mathrm{r}\,(\sigma_t) + \mathrm{r}'\,(\sigma_t) \cdot \left(\frac{\mu_t}{\sigma_t}\right) \leqslant \mathrm{r}\left(\sigma_t + \frac{\mu_t}{\sigma_t}\right) \leqslant \mathrm{r}\,(\sigma_t) + \mathrm{r}'\,(\sigma_t) \cdot \left(\frac{\mu_t}{\sigma_t}\right) + \mathrm{r}''\,(\sigma_t) \cdot \left(\frac{\mu_t}{\sigma_t}\right)^2 \tag{55}$$

and similarly, we get:

$$\mathrm{r}\,(\sigma_t) - \mathrm{r}'\,(\sigma_t) \cdot \left(\frac{\mu_t}{\sigma_t}\right) + \mathrm{r}''\,(\sigma_t) \cdot \left(\frac{\mu_t}{\sigma_t}\right)^2 \leqslant \mathrm{r}\left(\sigma_t - \frac{\mu_t}{\sigma_t}\right) \leqslant \mathrm{r}\,(\sigma_t) - \mathrm{r}'\,(\sigma_t) \cdot \left(\frac{\mu_t}{\sigma_t}\right) + R''\left(\frac{\mu_t}{\sigma_t}\right)^2 \tag{56}$$

where $R'' = \mathrm{r}''\,(\sigma_0)$. This is because $\mathrm{r}''\,(\cdot)$ takes positive values and is a decreasing function in $\sigma_t$ (refer to Lemma 21). We now lower bound $\alpha_1(\mu_t, \sigma_t)$ and upper bound $\alpha_2(\mu_t, \sigma_t)$:

$$\frac{\alpha_1(\mu_t, \sigma_t)}{\sqrt{\frac{2}{\pi}} \exp\left(-\frac{\mu_t^2}{2\sigma_t^2}\right)} \leqslant 2\mathrm{r}'\,(\sigma_t) \cdot \left(\frac{\mu_t}{\sigma_t}\right) \tag{57}$$

$$\frac{\alpha_2(\mu_t, \sigma_t)}{\sqrt{\frac{2}{\pi}} \exp\left(-\frac{\mu_t^2}{2\sigma_t^2}\right)} \geqslant 2\mathrm{r}\,(\sigma_t) + \mathrm{r}''\,(\sigma_t) \cdot \left(\frac{\mu_t}{\sigma_t}\right)^2 - \frac{2}{\sigma_t} \tag{58}$$

Substituting the lower bound and upper bound in (52) gives us the following as stricter a sufficient condition (i.e., (59) implies (52)):

$$\left[2\mathrm{r}\,(\sigma_t) + \mathrm{r}''\,(\sigma_t) \cdot \left(\frac{\mu_t}{\sigma_t}\right)^2 - \frac{2}{\sigma_t}\right] \cdot \frac{\sigma_{\mathrm{sp}}^2 \cdot \mu_t}{\gamma^2} \geqslant 2\mathrm{r}'\,(\sigma_t) \cdot \left(\frac{\mu_t}{\sigma_t}\right) \tag{59}$$

$$\iff \left[2\mathrm{r}\,(\sigma_t) + \mathrm{r}''\,(\sigma_t) \cdot \left(\frac{\mu_t}{\sigma_t}\right)^2 - \frac{2}{\sigma_t}\right] \geqslant 2\mathrm{r}'\,(\sigma_t) \cdot \left(\frac{\mu_t}{\sigma_t}\right) \cdot \frac{\gamma^2}{\sigma_{\mathrm{sp}}^2 \cdot \mu_t} \tag{60}$$

$$\iff 2\mathrm{r}\,(\sigma_t) + \mathrm{r}''\,(\sigma_t) \cdot \left(\frac{\mu_t}{\sigma_t}\right)^2 - \frac{2}{\sigma_t} - 2\mathrm{r}'\,(\sigma_t) \cdot \left(\frac{\mu_t}{\sigma_t}\right) \cdot \frac{\gamma^2}{\sigma_{\mathrm{sp}}^2 \cdot \mu_t} \geqslant 0 \tag{61}$$

$$\iff 2\mathrm{r}\,(\sigma_t) \cdot \sigma_t + \mathrm{r}''\,(\sigma_t) \cdot \frac{\mu_t^2}{\sigma_t} - 2 - 2\mathrm{r}'\,(\sigma_t) \cdot \frac{\gamma^2}{\sigma_{\mathrm{sp}}^2} \geqslant 0 \tag{62}$$

$$\iff 2\mathrm{r}'\,(\sigma_t) + \mathrm{r}''\,(\sigma_t) \cdot \frac{\mu_t^2}{\sigma_t} - 2\mathrm{r}'\,(\sigma_t) \cdot \frac{\gamma^2}{\sigma_{\mathrm{sp}}^2} \geqslant 0 \tag{63}$$

$$\iff \mathrm{r}''\,(\sigma_t) \cdot \frac{\mu_t^2}{\sigma_t} + 2\mathrm{r}'\,(\sigma_t) \cdot \left[1 - \frac{\gamma^2}{\sigma_{\mathrm{sp}}^2}\right] \geqslant 0 \tag{64}$$

Hence, when $\left[1 - \frac{\gamma^2}{\sigma_{\mathrm{sp}}^2}\right] \leqslant 0$, we have condition in (64) hold true as $\mathrm{r}'\,(\sigma_t)$ is always negative. Hence, the condition $\gamma \geqslant \sigma_{\mathrm{sp}}$ gives us the necessary condition. $\square$

### E.1.1 Proof of Proposition 3

For convenience, we first restate the Proposition 3 which gives us a closed form solution for (3) when $\rho = 0$. Then, we provide the proof, focusing first on the case of $k = 1$, and then showing that extension to $k > 1$ is straightforward and renders the final form in the proposition that follows.

**Proposition 12** (Barlow Twins solution). *The solution for* (3) *is* $U_k^\top \Sigma_{\mathsf{A}}^{-1/2}$ *where* $U_k$ *are the top* $k$ *eigenvectors of* $\Sigma_{\mathsf{A}}^{-1/2} \widetilde{\Sigma} \Sigma_{\mathsf{A}}^{-1/2}$. *Here,* $\Sigma_{\mathsf{A}} := \mathbb{E}_{a \sim P_{\mathsf{A}}}[aa^\top]$ *is the covariance over augmentations, and* $\widetilde{\Sigma} := \mathbb{E}_{x \sim P_{\mathsf{U}}}[\widetilde{a}(x)\widetilde{a}(x)^\top]$ *is the covariance matrix of mean augmentations* $\widetilde{a}(x) := \mathbb{E}_{P_{\mathsf{A}}(a|x)}[a]$.

*Proof.* We will use $\phi(x)$ to denote $\phi^\top x$ where $\phi \in \mathbb{R}^d$. Throughout the proof, we use $a$ to denote augmentation and $x$ to denote the input. We will use $P_{\mathsf{A}}(a \mid x)$ as the probability measure over the space of augmentations $\mathcal{A}$, given some input $x \in \mathcal{X}$ (with corresponding density) $p_{\mathsf{A}}(\cdot \mid x)$. Next, we use $p_{\mathsf{A}}(\cdot)$ to denote the density associate with the marginal probability measure over augmentations: $P_{\mathsf{A}} = \int_{\mathcal{X}} P_{\mathsf{A}}(a \mid x)\mathrm{d}P_{\mathsf{U}}$. Finally, the joint distribution over positive pairs $A_+(a_1, a_2) = \int_{\mathcal{X}} P_{\mathsf{A}}(a_1 \mid x)P_{\mathsf{A}}(a_2 \mid x)\mathrm{d}P_{\mathsf{U}}$, gives us the positive pair graph over augmentations.

Before we solve the optimization problem in (3) for $\Phi \in \mathbb{R}^{k \times d}$ for any general $k$, let us first consider the case where $k = 1$, *i.e.* we only want to find a single linear projection $\phi$. The constraint $\rho = 0$, transfers onto $\phi$ in the following way:

$$\mathbb{E}_{a \sim P_{\mathsf{A}}}[\phi(a)^2] = 1 \quad \equiv \quad \phi^\top \Sigma_A \phi = 1 \tag{65}$$

Under the above constraint we want to minimize the invariance loss, which according to Lemma 22 is given by $2 \cdot \int_{\mathcal{A}} \phi(a)L(\phi)(a) \, \mathrm{d}P_{\mathsf{A}}$, where $L(\phi)(\cdot)$ is the following linear operator.

$$L(\phi)(a) = \phi(a) - \int_{\mathcal{A}} \frac{A_+(a, a')}{p_{\mathsf{A}}(a)} \cdot \phi(a') \, \mathrm{d}a'. \tag{66}$$

Based on the definition of the operator, we can reformulate the constrained optimization for contrastive pretraining as:

$$\underset{\phi : \phi^\top \Sigma_A \phi = 1}{\arg\min} \int_{\mathcal{A}} \phi(a) \cdot L(\phi)(a) \, \mathrm{d}P_{\mathsf{A}} \tag{67}$$

$$\implies \underset{\phi : \phi^\top \Sigma_A \phi = 1}{\arg\min} \ \mathbb{E}_{a \sim P_{\mathsf{A}}}[\phi(a)^2] - \int_{\mathcal{A}} \int_{\mathcal{A}} \phi(a) \cdot \phi(a') \cdot A_+(a, a') \, \mathrm{d}a \mathrm{d}a' \tag{68}$$

$$\implies \underset{\phi : \phi^\top \Sigma_A \phi = 1}{\arg\min} \ \mathbb{E}_{a \sim P_{\mathsf{A}}}[\phi(a)^2] - \int_{\mathcal{X}} \int_{\mathcal{A}} \int_{\mathcal{A}} p_{\mathsf{A}}(a \mid x)p_{\mathsf{A}}(a' \mid x) \cdot \phi(a)\phi(a') \, \mathrm{d}P_{\mathsf{U}} \tag{69}$$

$$\implies \underset{\phi : \phi^\top \Sigma_A \phi = 1}{\arg\min} \ \mathbb{E}_{a \sim P_{\mathsf{A}}}[\phi(a)^2] - \int_{\mathcal{X}} [\widetilde{\phi}(x)]^2 \, \mathrm{d}P_{\mathsf{U}}, \tag{70}$$

where $\widetilde{\phi}(x) = \mathbb{E}_{a \sim P_{\mathsf{A}}(\cdot|x)}\phi(x) = \mathbb{E}_{c \sim \mathrm{Unif}[0,1]^d}[\phi^\top(c \odot x)]$. Note that,

$$\widetilde{\phi}(x)^2 = \left(\mathbb{E}_{c \sim \mathrm{Unif}[0,1]^d}[\phi^\top(c \odot x)]\right)^2 \tag{71}$$

$$= \phi^\top (\mathbb{E}_{c \sim \mathrm{Unif}[0,1]^d}[c \odot x])(\mathbb{E}_{c \sim \mathrm{Unif}[0,1]^d}[c \odot x])^\top \phi \tag{72}$$

$$\implies \int_{\mathcal{X}} [\widetilde{\phi}(x)]^2 \, \mathrm{d}P_{\mathsf{U}} = \phi^\top \widetilde{\Sigma} \phi \tag{73}$$

Further, since $\mathbb{E}_{a \sim P_{\mathsf{A}}}[\phi(a)^2] = \phi^\top \Sigma \phi$ we can now rewrite our main optimization problem for $k = 1$ as:

$$\underset{\phi : \phi^\top \Sigma_A \phi = 1}{\arg\min} \ \phi^\top \Sigma_A \phi - \phi^\top \widetilde{\Sigma} \phi \tag{74}$$

$$= \underset{\phi : \phi^\top \Sigma_A \phi = 1}{\arg\max} \ \phi^\top \widetilde{\Sigma} \phi \tag{75}$$

Recall that in our setup both $\widetilde{\Sigma}$ and $\Sigma_A$ are positive definite and invertible matrices. To solve the above problem, let's consider a re-parameterization: $\phi' = \Sigma_A^{1/2}\phi$, thus $\phi^\top \Sigma_A \phi = 1$, is equivalent to the constraint $\|\phi'\|_2^2 = 1$. Based on this re-parameterization we are now solving:

$$\underset{\|\phi'\|_2^2 = 1}{\arg\max} \quad \phi'^{\top} \Sigma_A^{-1/2} \cdot \widetilde{\Sigma} \cdot \Sigma_A^{-1/2} \phi', \tag{76}$$

which is nothing but the top eigenvector for $\Sigma_A^{-1/2} \cdot \widetilde{\Sigma} \cdot \Sigma_A^{-1/2}$.

Now, to extend the above argument from $k = 1$ to $k > 1$, we need to care of one additional form of constraint in the form of feature diversity: $\phi_i^{\top} \Sigma_A \phi_j = 0$ when $i \neq j$. But, we can easily redo the reformulations above and arrive at the following optimization problem:

$$\underset{\substack{\|\phi_i'\|_2^2 = 1, \ \forall i \\ \phi_i'^{\top} \phi_j' = 0, \ \forall i \neq j}}{\arg\max} \quad \left[\phi_1', \phi_2', \ldots, \phi_k'\right]^{\top} \Sigma_A^{-1/2} \cdot \widetilde{\Sigma} \cdot \Sigma_A^{-1/2} \left[\phi_1', \phi_2', \ldots, \phi_k'\right], \tag{77}$$

where $\phi_i' = \Sigma_A^{1/2} \phi_i$. The above is nothing but the top $k$ eigenvectors for the matrix $\Sigma_A^{-1/2} \cdot \widetilde{\Sigma} \cdot \Sigma_A^{-1/2}$. This completes the proof of Proposition 12. $\qquad\square$

### E.1.2  Analysis with $\rho > 0$ in Contrastive Pretraining Objective (3)

In (3) we considered the strict version of the optimization problem where $\rho = 0$. Here, we will consider the following optimization problem that we optimize for our experiments in the simplified setup:

$$\mathcal{L}_{\mathrm{cl}}(\Phi, \kappa) := \mathbb{E}_{x \sim \mathrm{P_U}} \mathbb{E}_{a_1, a_2 \sim \mathrm{P_A}(\cdot|x)} \|\Phi(a_1) - \Phi(a_2)\|_2^2 + \kappa \cdot \left\|\mathbb{E}_{a \sim \mathrm{P_A}}\left[\Phi(a)\Phi(a)^{\top}\right] - \mathbf{I}_k\right\|_F^2, \tag{78}$$

where $\kappa > 0$ is some finite constant (note that every $\rho$ corresponds to some $\kappa$ and particularly $\rho = 0$, corresponds to $\kappa = \infty$). Let $\Phi^{\star}$ be the solution for (3) with $\rho = 0$, *i.e.* the solution described in Proposition 3. Now, we will show that in practice we can provably recover something close to $\Phi^{\star}$ when $\kappa$ is large enough.

**Theorem 13** (Solution for (78) is approximately equal to $\Phi^{\star}$). *If $\widehat{\Phi}$ is some solution that achieves low values of the objective $\mathcal{L}_{\mathrm{cl}}(\Phi, \kappa)$ in (78), i.e., $\mathcal{L}_{\mathrm{cl}}(\widehat{\Phi}, \kappa) \leqslant \epsilon$, then there exists matrix $W \in \mathbb{R}^{k \times k}$ such that:*

$$\mathbb{E}_{a \sim \mathrm{P_A}} \|W \cdot \Phi^{\star}(a) - \widehat{\Phi}(a)\|_2^2 \leqslant \frac{k\epsilon}{2\gamma_{k+1}},$$

$$where, \ \gamma_{k+1} \geqslant \frac{2\gamma_1^2}{k\epsilon} \cdot \left(1 - \sqrt{\frac{\epsilon}{\kappa}}\right) - \frac{\gamma_1}{k},$$

*where $\gamma_{k+1}$ is the the $k + 1^{th}$ eigenvalue for $\mathbf{I}_d - \Sigma_A^{-1/2} \widetilde{\Sigma} \Sigma_A^{-1/2}$. Here, $\lambda_1 \leqslant \lambda_2 \leqslant \ldots \leqslant \lambda_d$.*

*Proof.* Since we know that $\mathcal{L}_{\mathrm{cl}}(\widehat{\Phi}, \kappa) \leqslant \epsilon$, we can individually bound the invariance loss and the regularization term:

$$\mathbb{E}_{x \sim \mathrm{P_U}} \mathbb{E}_{a_1, a_2 \sim \mathrm{P_A}(\cdot|x)} \|\widehat{\Phi}(a_1) - \widehat{\Phi}(a_2)\|_2^2 \leqslant \epsilon \tag{79}$$

$$\left\|\mathbb{E}_{a \sim \mathrm{P_A}}\left[\widehat{\Phi}(a)\widehat{\Phi}(a)^{\top}\right] - \mathbf{I}_k\right\|_F^2 \leqslant \frac{\epsilon}{\kappa} \tag{80}$$

Thus,

$$\forall i \in [k]: \quad 1 - \sqrt{\frac{\epsilon}{\kappa}} \leqslant \widehat{\phi}_i^{\top} \Sigma_A \widehat{\phi}_i \leqslant 1 + \sqrt{\frac{\epsilon}{\kappa}} \tag{81}$$

$$\forall i \in [k]: \quad \mathbb{E}_{x \sim \mathrm{P_U}} \mathbb{E}_{a_1, a_2 \sim \mathrm{P_A}(\cdot|x)} (\widehat{\phi}_i^{\top} a_1 - \widehat{\phi}_i^{\top} a_2)^2 \leqslant \epsilon \tag{82}$$

Let $\phi_1^\star, \phi_2^\star, \phi_3^\star, \ldots, \phi_d^\star$ be the solution returned by the analytical solution for $\rho = 0$, *i.e.* the solution in Proposition 3. Now, since $\Phi^\star$ would span $\mathbb{R}^d$ when $\Sigma_A$ is full rank, we can denote:

$$\widehat{\phi}_i = \sum_{j=1}^{d} \eta_i^{(j)} \phi_j^\star \tag{83}$$

Now from Lemma 22, the invariance loss for $\widehat{\phi}_i$ can be written using the operator $L(\phi)(a) = \phi(a) - \int_{\mathcal{A}} \frac{A_+(a,a')}{p_A(a)} \phi(a') \, \mathrm{d}a'$:

$$\text{Invariance Loss}(\widehat{\phi}_i) := \mathbb{E}_{x \sim P_U} \mathbb{E}_{a_1, a_2 \sim P_A(\cdot|x)} (\widehat{\phi}_i^\top a_1 - \widehat{\phi}_i^\top a_2)^2 \tag{84}$$

$$= 2 \cdot \mathbb{E}_{a \sim P_A} [\widehat{\phi}_i(a) L(\widehat{\phi}_i)(a)] \tag{85}$$

$$= 2 \cdot \mathbb{E}_{a \sim P_A} \left[ \left( \sum_{j=1}^{d} \eta_i^{(j)} \phi_i^\star \right) L \left( \sum_{j=1}^{d} \eta_i^{(j)} \phi_j^\star \right) (a) \right] \tag{86}$$

$$= 2 \cdot \mathbb{E}_{a \sim P_A} \left[ \left( \sum_{j=1}^{d} \eta_i^{(j)} \phi_j^\star \right) \left( \sum_{j=1}^{d} \eta_i^{(j)} L(\phi_j^\star)(a) \right) \right] \tag{87}$$

$$= 2 \cdot \sum_{j=1}^{d} \left( \eta_i^{(j)} \right)^2 \mathbb{E}_{a \sim P_A} \left[ \phi_j^\star(a) L(\phi_j^\star)(a) \right] \tag{88}$$

$$+ 2 \cdot \sum_{m=1, n=1, m \neq n}^{d} \eta_i^{(m)} \eta_i^{(n)} \mathbb{E}_{a \sim P_A} \left[ \phi_m^\star(a) L(\phi_n^\star)(a) \right] \tag{89}$$

Since, $\phi_i^\star(\cdot)$ are eigenfunctions of the operator $L$ [38], we can conclude that:

$$\sum_{m=1, n=1, m \neq n}^{d} \eta_i^{(m)} \eta_i^{(n)} \mathbb{E}_{a \sim P_A} \left[ \phi_m^\star(a) L(\phi_n^\star)(a) \right] = 0,$$

and if $\gamma_1 \leqslant \gamma_2 \leqslant \gamma_3 \ldots \leqslant \gamma_d$ are the eigenvalues for $\phi_1^\star, \phi_2^\star, \phi_3^\star, \ldots, \phi_d^\star$ under the decomposition of $L(\phi)(\cdot)$ then:

$$\mathbb{E}_{x \sim P_U} \mathbb{E}_{a_1, a_2 \sim P_A(\cdot|x)} (\widehat{\phi}_i^\top a_1 - \widehat{\phi}_i^\top a_2)^2 = 2 \cdot \sum_{j=1}^{d} \gamma_j \left( \eta_i^{(j)} \right)^2 \tag{90}$$

Recall, we are also aware of a condition on the regularization term: $1 - \sqrt{\frac{\epsilon}{\kappa}} \leqslant \widehat{\phi}_i^\top \Sigma_A \widehat{\phi}_i \leqslant 1 + \sqrt{\frac{\epsilon}{\kappa}}$.

$$\widehat{\phi}_i^\top \Sigma_A \widehat{\phi}_i = \left( \sum_{j=1}^{d} \eta_i^{(j)} \phi_j^\star \right)^\top \Sigma_A \left( \sum_{j=1}^{d} \eta_i^{(j)} \phi_j^\star \right) = \sum_{j=1}^{d} \left( \eta_i^{(j)} \right)^2 \tag{91}$$

$$\implies 1 - \sqrt{\frac{\epsilon}{\kappa}} \leqslant \sum_{j=1}^{d} \left( \eta_i^{(j)} \right)^2 \leqslant 1 + \sqrt{\frac{\epsilon}{\kappa}} \quad \forall i. \tag{92}$$

In order to show that the projection of $\widehat{\phi}_i$ on $\Phi^*$ is significant, we need to argue that the term $\sum_{j=k+1}^{d} \left( \eta_i^{(j)} \right)^2$ is small. The argument for this begins with the condition on invariance loss, and the fact that $\gamma_1 \leqslant \gamma_2 \leqslant \ldots \leqslant \gamma_k \leqslant \gamma_{k+1} \leqslant \ldots \leqslant \gamma_d$:

$$\frac{\epsilon}{2} \geqslant \sum_{j=k+1}^{d} \left( \eta_i^{(j)} \right)^2 \gamma_j \geqslant \gamma_{k+1} \cdot \left( \sum_{j=k+1}^{d} \left( \eta_i^{(j)} \right)^2 \right) \tag{93}$$

$$\implies \sum_{j=k+1}^{d} \left( \eta_i^{(j)} \right)^2 \leqslant \frac{\epsilon}{2\gamma_{k+1}} \tag{94}$$

Extending the above result $\forall i$ by simply adding the bounds completes the claim of our first result in Theorem 13. Next, we will lower bound the eigenvalue $\gamma_{k+1}$. Recall that, $\sum_{j=1}^{k} \left( \eta_i^{(j)} \right)^2 \geqslant 1 - \sqrt{\frac{\epsilon}{\kappa}} - \frac{\epsilon}{2\gamma_{k+1}}$. Thus,

$$\gamma_1 \cdot \left( 1 - \sqrt{\frac{\epsilon}{\kappa}} - \frac{\epsilon}{2\gamma_{k+1}} \right) \leqslant \sum_{j=1}^{k} \gamma_j \left( \eta_i^{(j)} \right)^2 \leqslant k\gamma_{k+1} \cdot \frac{\epsilon}{2\gamma_1} \tag{95}$$

We assume that all eigenvalues are strictly positive, which is true under our augmentation distribution. Given, $\gamma_{k+1} \geqslant \gamma_1$, we can rearrange the above to get:

$$\gamma_{k+1} \geqslant \frac{2\gamma_1^2}{k\epsilon} \cdot \left( 1 - \sqrt{\frac{\epsilon}{\kappa}} \right) - \frac{\gamma_1}{k} \tag{96}$$

This completes the claim of our second result in Theorem 13. $\qquad\square$

### E.1.3 Proof of Theorem 5

In this section, we prove our main theorem about the recovery of both spurious $w_{\text{sp}}$, invariant $w_{\text{in}}$ features by the contrastive learning feature backbone, and also the amplification of the invariant over the spurious feature (where amplification is defined relatively with respect to what is observed in the data distribution alone). We begin by defining some quantities needed for analysis, that are fully determined by the choice of problem parameters for the model in (6).

From Section 4, we recall the definitions of $w_{\text{in}} := [w^\star, 0, \ldots, 0]$ and $w_{\text{sp}} := [0, \ldots 0, w']$ where $w' = \mathbf{1}_{d_{\text{sp}}}/\sqrt{d_{\text{sp}}}$. Let us now define $u_1, u_2$ as the top two eigenvectors of $\Sigma_A$ with eigenvalues $\lambda_1, \lambda_2 > 0$, (note that in our problem setup both $\Sigma_A$ and $\widetilde{\Sigma}$ are full rank positive definite matrices), and $\tau := \sqrt{\lambda_1/\lambda_2}$. Next we define $\alpha$ as the angle between $u_1$ and $w_{\text{in}}$, *i.e.*, $\cos(\alpha) = u_1^\top w_{\text{in}}$. Based on the definitions of $\alpha$ and $\tau$, both of which are fully determined by the eigen decomposition of the post-augmentation feature covariance matrix $\Sigma_A$, we now restate Theorem 5:

**Theorem 14** (Formal; CL recovers both invariant $w_{\text{in}}$ and spurious $w_{\text{sp}}$ but amplifies $w_{\text{in}}$). *Under Assumption 4 ($w^\star = \mathbf{1}_{d_{\text{in}}}/\sqrt{d_{\text{in}}}$), the CL solution $\Phi_{\text{cl}} = [\phi_1, \phi_2, ..., \phi_k]$ satisfies $\phi_j^\top w_{\text{in}} = \phi_j^\top w_{\text{sp}} = 0$ $\forall j \geqslant 3$. For $\tau, \alpha$ as defined above, the solution for $\phi_1, \phi_2$ is:*

$$\begin{bmatrix} w^\star \cdot \cot(\alpha)/\tau, & w^\star \\ w' \cdot 1/\tau, & w' \cdot \cot(\alpha) \end{bmatrix} \cdot \begin{bmatrix} \cos\theta, & \sin\theta \\ \sin\theta, & -\cos\theta \end{bmatrix},$$

*where $0 \leqslant \alpha, \theta \leqslant \pi/2$. Let us redefine $\phi_1 = c_1 w_{\text{in}} + c_3 w_{\text{sp}}$ and $\phi_2 = c_2 w_{\text{in}} + c_4 w_{\text{sp}}$.*

*For constants $K_1, K_2 > 0$, $\gamma = K_1 K_2/\sigma_{\text{sp}}$, $d_{\text{sp}} = \sigma_{\text{sp}}^2/K_2^2$, $\forall \epsilon > 0$, $\exists \sigma_{\text{sp}0}$, such that for $\sigma_{\text{sp}} \geqslant \sigma_{\text{sp}0}$:*

$$\frac{K_1 K_2^2 d_{\text{in}}}{2L\sigma_{\text{in}}^2(d_{\text{in}} - 1)} + \epsilon \geqslant \frac{c_1}{c_3} \geqslant \frac{K_1 K_2^2 d_{\text{in}}}{2L\sigma_{\text{in}}^2(d_{\text{in}} - 1)} - \epsilon$$

$$\frac{L\sqrt{d_{\text{sp}}}}{\gamma} + \epsilon \geqslant \left| \frac{c_2}{c_4} \right| \geqslant \frac{L\sqrt{d_{\text{sp}}}}{\gamma} - \epsilon,$$

*where $L = 1 + K_2^2$.*

*Proof.* We will first show that the only components of interest are $\phi_1, \phi_2$. Then, we will prove conditions on the amplification of $w_{\text{in}}$ over $w_{\text{sp}}$ in $\phi_1, \phi_2$. Following is the proof overview:

    I. When $w^\star = \mathbf{1}_{d_{\text{in}}}/\sqrt{d_{\text{in}}}$, from the closed form expressions for $\Sigma_A$ and $\widetilde{\Sigma}$, show that the solution returned by solving the Barlow Twins objective depends on $w_{\text{in}}$ and $w_{\text{sp}}$ only through the first two components $\phi_1, \phi_2$.

II. For the components $\phi_1, \phi_2$, we will show that the dependence along $w_{\text{in}}$ is amplified compared to $w_{\text{sp}}$ when the target data sufficiently denoises the spurious feature (*i.e.*, $\sigma_{\text{sp}}$ is sufficiently large).

**Part-I:**

We can divide the space $\mathbb{R}^d$ into two subspaces that are perpendicular to each other. The first subspace is $\mathcal{W} = \{b_1 \cdot w_{\text{in}} + b_2 \cdot w_{\text{sp}} : b_1, b_2 \in \mathbb{R}\}$, *i.e.* the rank 2 subspace spanned by $w_{\text{in}}$ and $w_{\text{sp}}$. The second subspace is $\mathcal{W}_\perp$ where $\mathcal{W}_\perp = \{u \in \mathbb{R}^d : u^\top w_{\text{in}} = 0, u^\top w_{\text{sp}} = 0\}$. Then, from Lemma 23 we can conclude that the matrix $\Sigma_A$ can be written as:

$$\Sigma_A = \Sigma_{A_\mathcal{W}} + \Sigma_{A_{\mathcal{W}_\perp}}$$
$$\Sigma_{A_\mathcal{W}} = \frac{1}{4}\begin{bmatrix} \left(\gamma^2(1 + 1/3d_{\text{in}}) + \sigma_{\text{in}}^2/3(1 - 1/d_{\text{in}})\right) \cdot w^\star w^{\star\top}, & \gamma\sqrt{d_{\text{sp}}}/2 \cdot w^\star w'^\top \\ \gamma\sqrt{d_{\text{sp}}}/2 \cdot w' w^{\star\top}, & \left(d_{\text{sp}}/2 + 4/3 \cdot \sigma_{\text{sp}}^2 + 1/6\right) \cdot w' w'^\top \end{bmatrix}, \quad (97)$$

where $\Sigma_{A_{\mathcal{W}_\perp}} := \mathbb{E}_{a \sim P_A}\left[\Pi_{\mathcal{W}_\perp}(a)(\Pi_{\mathcal{W}_\perp}(a))^\top\right]$ is the covariance matrix in the null space of $\mathcal{W}$, and $\Pi_{\mathcal{W}_\perp}(a)$ is the projection of augmentation $a$ into the null space of $\mathcal{W}$, *i.e.* the covariance matrix in the space of non-predictive (noise) features. Similarly we can define:

$$\widetilde{\Sigma} = \widetilde{\Sigma}_\mathcal{W} + \widetilde{\Sigma}_{\mathcal{W}_\perp}$$
$$\widetilde{\Sigma}_\mathcal{W} = \frac{1}{4}\begin{bmatrix} \gamma^2 \cdot w^\star w^{\star\top}, & \gamma\sqrt{d_{\text{sp}}}/2 \cdot w^\star w'^\top \\ \gamma\sqrt{d_{\text{sp}}}/2 \cdot w' w^{\star\top}, & \left(d_{\text{sp}}/2 + \sigma_{\text{sp}}^2/2\right) \cdot w' w'^\top \end{bmatrix} \quad (98)$$

Here again $\widetilde{\Sigma}_{\mathcal{W}_\perp} := \mathbb{E}_{x \sim P_U}\left[\Pi_{\mathcal{W}_\perp}(\mathbb{E}_{c \sim \text{Unif}[0,1]^d}(c \odot x))(\Pi_{\mathcal{W}_\perp}(\mathbb{E}_{c \sim \text{Unif}[0,1]^d}(c \odot x)))^\top\right]$ is the co-variance matrix of mean augmentations after they are projected onto the null space of predictive features. The above decomposition also follows from result in Lemma 23.

From Proposition 3, the closed form expression for the solution returned by optimizing the Barlow Twins objective in (3) is $U^\top \Sigma_A^{-1/2}$ where $U$ are the top-k eigenvectors of:

$$\Sigma_A^{-1/2} \cdot \widetilde{\Sigma} \cdot \Sigma_A^{-1/2} \quad (99)$$

When $w^\star = \mathbf{1}_{d_{\text{in}}}/\sqrt{d_{\text{in}}}$, then $\Sigma_{A_{\mathcal{W}_\perp}} = \widetilde{\Sigma}_{\mathcal{W}_\perp} + B$ where $B$ is a diagonal matrix with diagonal given by $\frac{1}{3} \cdot \text{diag}(\widetilde{\Sigma}_{\mathcal{W}_\perp})$. Further, since $\text{diag}(\widetilde{\Sigma}_{\mathcal{W}_\perp}) = p \cdot \mathbb{1}_d$ for some constant $p > 0$, the eigenvectors of $\widetilde{\Sigma}_{\mathcal{W}_\perp}$ and $\Sigma_{A_{\mathcal{W}_\perp}}$ are exactly the same. Hence, when we consider the SVD of the expression $\Sigma_A^{-1/2}\widetilde{\Sigma}\Sigma_A^{-1/2}$, the matrices $\Sigma_{A_{\mathcal{W}_\perp}}$ and $\widetilde{\Sigma}_{\mathcal{W}_\perp}$ have no effect on the SVD components that lie along the span of the predictive features. In fact, we only need to consider two rank 2 matrices (first terms in (98), (97)) and only do the SVD of $\Sigma_{A_\mathcal{W}}^{-1/2} \cdot \widetilde{\Sigma}_\mathcal{W} \cdot \Sigma_{A_\mathcal{W}}^{-1/2}$.

There are only two eigenvectors of $\Sigma_{A_\mathcal{W}}^{-1/2} \cdot \widetilde{\Sigma}_\mathcal{W} \cdot \Sigma_{A_\mathcal{W}}^{-1/2}$. We use $\lambda_1, \lambda_2$ to denote the eigenvalues of $\Sigma_{A_\mathcal{W}}$, and $[\cos(\alpha)w^\star, \sin(\alpha)w']^\top$, $[\sin(\alpha)w^\star, -\cos(\alpha)w']^\top$ for the corresponding eigenvectors. Similarly, we use $\widetilde{\lambda}_1, \widetilde{\lambda}_2$ to denote the eigenvalues of $\widetilde{\Sigma}_\mathcal{W}$, and $[\cos(\beta)w^\star, \sin(\beta)w']^\top$, $[\sin(\beta)w^\star, -\cos(\beta)w']^\top$ for the corresponding eigenvectors. Let $\text{SVD}_U(\cdot)$ denote the operation of obtaining the singular vectors of a matrix. Then, to compute the components of the final expression: $\text{SVD}_U(\Sigma_A^{-1/2}\widetilde{\Sigma}\Sigma_A^{-1/2})^\top\Sigma_A^{-1/2}$ that lies along the span of predictive features (in $\mathcal{W}$), we need only look at the decomposition of the following matrix:

$$\begin{bmatrix} \cos\theta , & \sin(\theta) \\ \sin\theta , & -\cos(\theta) \end{bmatrix} = \text{SVD}_U\left(\begin{bmatrix} 1/\sqrt{\lambda_1}, & 0 \\ 0, & 1/\sqrt{\lambda_2} \end{bmatrix} \cdot \begin{bmatrix} \cos(\alpha - \beta), & \sin(\alpha - \beta) \\ \sin(\alpha - \beta), & -\cos(\alpha - \beta) \end{bmatrix} \cdot \begin{bmatrix} \sqrt{\widetilde{\lambda}_1}, & 0 \\ 0, & \sqrt{\widetilde{\lambda}_2} \end{bmatrix}\right)$$
$$(100)$$

Based on the above definitions of $\theta, \alpha, \lambda_1, \lambda_2$, we can then formulate $\phi_1$ and $\phi_2$ in the following way:

$$[\phi_1, \phi_2] = \begin{bmatrix} w^\star \cdot \frac{\cos(\alpha)}{\sqrt{\lambda_1}}, & w^\star \cdot \frac{\sin(\alpha)}{\sqrt{\lambda_2}} \\ w' \cdot \frac{\sin(\alpha)}{\sqrt{\lambda_1}}, & w' \frac{-\cos(\alpha)}{\sqrt{\lambda_2}} \end{bmatrix} \cdot \begin{bmatrix} \cos\theta &, & \sin(\theta) \\ \sin\theta &, & -\cos(\theta) \end{bmatrix} \tag{101}$$

To summarize, using arguments in Lemma 23 and the fact that $w^\star = \mathbf{1}_{d_{\text{in}}}/\sqrt{d_{\text{in}}}$, we can afford to focus on just two rank two matrices $\Sigma_{A_\mathcal{W}}, \widetilde{\Sigma}_\mathcal{W}$ in the operation: $\text{SVD}_U(\Sigma_A^{-1/2})\widetilde{\Sigma}\Sigma_A^{-1/2}$. The other singular vectors from the SVD only impact directions that span $\mathcal{W}_\perp$, and the singular vectors obtained by considering only the rank 2 matrices lie only in the space of $\mathcal{W}$.

**Part-II:**

From the previous part we obtained forms of $\phi_1, \phi_2$ in terms of: $\lambda_1, \lambda_2, \alpha, \theta$, all of which are fully specified by the SVD of $\Sigma_{A_\mathcal{W}}$ and $\widetilde{\Sigma}_\mathcal{W}$. If we define $\tau := \frac{\sqrt{\lambda_1}}{\sqrt{\lambda_2}}$, we can evaluate $c_1, c_2, c_3, c_4$ as:

$$c_1 = \frac{\cot(\alpha)}{\tau} + \tan(\theta) \tag{102}$$

$$c_2 = -1 + \frac{\cot(\alpha)\tan(\theta)}{\tau} \tag{103}$$

$$c_3 = \frac{1}{\tau} - \cot(\alpha)\tan(\theta) \tag{104}$$

$$c_4 = \frac{\tan(\theta)}{\tau} + \cot(\alpha) \tag{105}$$

Now, we are ready to begin proofs for our claims on the amplification factors, *i.e.* on the ratios $c_1/c_3$, $|c_2/c_4|$.

We will first prove some limiting conditions for $c_1/c_3$, followed by those on $|c_2/c_4|$. For each of these conditions we will rely on the forms for $c_1, c_2, c_3, c_4$ derived in the previous part, in terms of $\alpha, \theta, \tau$ (where $0 \leqslant \alpha, \theta \leqslant \pi/2$). We will also rely on some lemmas that characterize the asymptotic behavior of $\alpha, \theta$ and $\tau$ as we increase $\sigma_{\text{sp}}$. We defer the full proof of these helper lemmas to later sections.

**Asymptotic behavior of $c_1/c_3$.**

From Lemma 25 and Lemma 26, when $\gamma = K_1/\sqrt{z}$ and $\sigma_{\text{sp}} = K_2\sqrt{z}$, then:

$$\lim_{z \to \infty} \frac{c_1}{c_3} = \frac{\cot\alpha + \tau\tan\theta}{1 - \tau\cot\alpha\tan\theta} = \lim_{z \to \infty} \tau\tan\theta = \frac{K_1 K_2^2}{(1 + K_2^2)2\sigma_{\text{in}}^2(1 - 1/d_{\text{in}})}, \tag{106}$$

where we apply Moore-Osgood when applying limits on intermediate forms. We can do this since $\tau\tan\theta$ approaches a constant, and each of $\cot\alpha, \tau$ and $\tan\theta$ are continuous and smooth functions of $z$ (see Lemma 24).

**Asymptotic behavior of $|c_2/c_4|$.**

When we consider the limiting behavior of $c_2/c_4 z$, as we increase $z$ or equivalently $\sigma_{\text{sp}}$ when $\gamma = K_1/\sqrt{z}$ and $\sigma_{\text{sp}} = K_2\sqrt{z}$, then we get:

$$\lim_{z \to \infty} \left| \frac{c_2}{c_4 z} \right| = \left| \frac{-1 + \cot(\alpha)\tan(\theta)}{\frac{\tan(\theta)z}{\tau} + \cot(\alpha)z} \right|. \tag{107}$$

From Lemma 26, $\cot\alpha\tan\theta \to 0$. Next, if we consider $\lim_{z\to\infty} z\tan\theta/\tau = \lim_{z\to\infty} \tau\tan\theta \cdot z/\tau^2$. For $z/\tau^2$, we invoke Lemma 28, which states that when $\gamma = K_1/\sqrt{z}$ and $\sigma_{\text{sp}} = K_2\sqrt{z}$, then:

$$\lim_{z \to \infty} \frac{z}{\tau^2} = \frac{2\sigma_{\text{in}}^2/3(1 - 1/d_{\text{in}})}{1 + 4/3K_2^2}. \tag{108}$$

Further, in our bound on $c_1/c_3$, we derived that $\tau\tan\theta \to K_1 K_2^2/(1+K_2^2)2\sigma_{\text{in}}^2(1-1/d_{\text{in}})$. Once again using Moore-Osgood we can plug this along with (108) to get:

$$\lim_{z \to \infty} \frac{\tan(\theta)z}{\tau} = \frac{K_1 K_2^2}{(1 + K_2^2)(3 + 4K_2^2)}. \tag{109}$$

Finally, from Lemma 27, when $\gamma = K_1/\sqrt{z}$ and $\sigma_{\mathrm{sp}} = K_2\sqrt{z}$, then:

$$\lim_{z \to \infty} \frac{z}{\tan \alpha} = \frac{K_1}{(1 + 4/3 K_2^2)}. \tag{110}$$

Plugging, 109 and 110 into 107 we get the following limit:

$$\lim_{z \to \infty} \left| \frac{c_2}{c_4 z} \right| = \frac{1 + K_2^2}{K_1}. \tag{111}$$

Since $z = K_1\sqrt{d_{\mathrm{sp}}}/\gamma$,

$$\lim_{z \to \infty} \left| \frac{c_2 \gamma}{c_4 K_1 \sqrt{d_{\mathrm{sp}}}} \right| = \frac{1 + K_2^2}{K_1} \implies \lim_{z \to \infty} \left| \frac{c_2 \gamma}{c_4 \sqrt{d_{\mathrm{sp}}}} \right| = 1 + K_2^2 \tag{112}$$

Since both $c_1/c_3$ and $|c_2/c_4|$ are continuous functions of $z$, with $\liminf_{z \to \infty}$ and $\limsup_{z \to \infty}$ converging to the limits in 106 and 107 for both quantities respectively, we conclude that $\forall \epsilon > 0$ there exists $\sigma_{\mathrm{sp}0}$ such that for all $\sigma_{\mathrm{sp}} \geqslant \sigma_{\mathrm{sp}0}$, the following is true:

$$\frac{K_1 K_2^2 d_{\mathrm{in}}}{2L\sigma_{\mathrm{in}}^2(d_{\mathrm{in}} - 1)} + \epsilon \geqslant \frac{c_1}{c_3} \geqslant \frac{K_1 K_2^2 d_{\mathrm{in}}}{2L\sigma_{\mathrm{in}}^2(d_{\mathrm{in}} - 1)} - \epsilon \tag{113}$$

$$\frac{(1 + K_2^2)\sqrt{d_{\mathrm{sp}}}}{\gamma} + \epsilon \geqslant \left| \frac{c_2}{c_4} \right| \geqslant \frac{(1 + K_2^2)\sqrt{d_{\mathrm{sp}}}}{\gamma} - \epsilon, \tag{114}$$

This completes both Part-I and Part-II of the proof for Theorem 5.

$$\square$$

### E.1.4 Proof of Corollary 6

**Corollary 15** (CL improves OOD error over ERM but is still imperfect). *For $\gamma, \sigma_{\mathrm{sp}}, d_{\mathrm{sp}}$ defined as in Theorem 5, $\exists \sigma_{\mathrm{sp}1}$ such that $\forall \sigma_{\mathrm{sp}} \geqslant \sigma_{\mathrm{sp}1}$, the target accuracy of CL (linear predictor on $\Phi_{\mathrm{cl}}$) is $\geqslant 0.5\,\mathrm{erfc}\left(-L' \cdot \gamma/\sqrt{2}\sigma_{\mathrm{sp}}\right)$ and $\leqslant 0.5\,\mathrm{erfc}\left(-4L' \cdot \gamma/\sqrt{2}\sigma_{\mathrm{sp}}\right)$, where $L' = K_2^2 K_1/\sigma_{\mathrm{in}}^2(1 - 1/d_{\mathrm{in}})$. When $\sigma_{\mathrm{sp}1} > \sigma_{\mathrm{in}}\sqrt{1 - 1/d_{\mathrm{in}}}$, the lower bound on accuracy is strictly better than ERM from scratch.*

*Proof.* Recall from Theorem 14, all $\phi_j$, for $j \geqslant 3$, lie in the null space of $w_{\mathrm{in}}$ and $w_{\mathrm{sp}}$. Since, the predictive features are strictly contained in the rank two space spanned by $w_{\mathrm{in}}$ and $w_{\mathrm{sp}}$, without loss of generality we can restrict ourselves to the case where $k = 2$, and when doing training a head $h = [h_1, h_2]^\top \in \mathbb{R}^2$ over contrastive pretrained representations using source labeled data, we get the following max margin solution:

$$h_1 = c_1 \cdot \gamma + c_3 \cdot \sqrt{d_{\mathrm{sp}}}$$
$$h_2 = c_2 \cdot \gamma + c_4 \cdot \sqrt{d_{\mathrm{sp}}} \tag{115}$$

Without loss of generality we can divide both $h_1$ and $h_2$ by $h_1$ and get the final classifier to be $\phi_1 + \frac{h_2}{h_1} \cdot \phi_2$:

$$(c_1 w_{\mathrm{in}} + c_3 w_{\mathrm{sp}}) + \frac{h_2}{h_1} \cdot (c_2 w_{\mathrm{in}} + c_4 w_{\mathrm{sp}})$$

$$= (c_1 w_{\mathrm{in}} + c_3 w_{\mathrm{sp}}) + \frac{(c_2 \gamma + c_4 \sqrt{d_{\mathrm{sp}}})}{(c_1 \gamma + c_3 \sqrt{d_{\mathrm{sp}}})} \cdot (c_2 w_{\mathrm{in}} + c_4 w_{\mathrm{sp}}) \tag{116}$$

From Lemma 29, we can derive the target accuracy of the classifier $h$ on top of CL representations to be the following:

$$0.5 \operatorname{erfc}\left(-\frac{c_1 + \beta c_2}{c_3 + \beta c_4} \cdot \frac{\gamma}{\sqrt{2}\sigma_{\mathrm{sp}}}\right) \tag{117}$$

where $\beta = {(c_2\gamma + c_4\sqrt{d_{\mathrm{sp}}})}/{(c_1\gamma + c_3\sqrt{d_{\mathrm{sp}}})}$.

Substituting $\beta$ into the expression $\frac{c_1 + \beta c_2}{c_3 + \beta c_4}$ we get:

$$\frac{c_1^2\gamma + c_1 c_3 \sqrt{d_{\mathrm{sp}}} + c_2^2\gamma + c_2 c_4 \sqrt{d_{\mathrm{sp}}}}{c_1 c_3 \gamma + c_3^2 \sqrt{d_{\mathrm{sp}}} + c_2 c_4 \gamma + c_4^2 \sqrt{d_{\mathrm{sp}}}} \tag{118}$$

We first substitute expressions for $c_1, c_2, c_3, c_4$ from (102), (103), (104) and (105) in the above expression. Then for $\gamma = K_1/\sqrt{z}, \sigma_{\mathrm{sp}} = K_2\sqrt{z}$, we substitute the expressions for $\cot\alpha, \tan\theta$, and $\tau = \lambda_1/\lambda_2$ with their corresponding closed form expressions (as functions of $z$) from Lemma 24. On the resulting expression we apply do repeated applications of L'Hôpital's rule to get the following result:

$$\lim_{z\to\infty} \frac{c_1^2\gamma + c_1 c_3 \sqrt{d_{\mathrm{sp}}} + c_2^2\gamma + c_2 c_4 \sqrt{d_{\mathrm{sp}}}}{c_1 c_3 \gamma + c_3^2 \sqrt{d_{\mathrm{sp}}} + c_2 c_4 \gamma + c_4^2 \sqrt{d_{\mathrm{sp}}}} = \frac{2K_2^2 K_1}{\sigma_{\mathrm{in}}^2(1 - 1/d_{\mathrm{in}})} \tag{119}$$

Based on $\gamma, d_{\mathrm{sp}}, \sigma_{\mathrm{sp}}$ defined in Theorem 5, and (119) we can conclude that $\exists \sigma_{\mathrm{sp}_1}$ such that for all $\sigma_{\mathrm{sp}} \geqslant \sigma_{\mathrm{sp}_1}$:

$$\frac{4K_2^2 K_1}{\sigma_{\mathrm{in}}^2(1 - 1/d_{\mathrm{in}})} \geqslant \frac{c_1^2\gamma + c_1 c_3 \sqrt{d_{\mathrm{sp}}} + c_2^2\gamma + c_2 c_4 \sqrt{d_{\mathrm{sp}}}}{c_1 c_3 \gamma + c_3^2 \sqrt{d_{\mathrm{sp}}} + c_2 c_4 \gamma + c_4^2 \sqrt{d_{\mathrm{sp}}}} \geqslant \frac{K_2^2 K_1}{\sigma_{\mathrm{in}}^2(1 - 1/d_{\mathrm{in}})} \tag{120}$$

Finally, applying (120) to Lemma 29, we conclude the following: When $\gamma = {K_1 K_2}/{\sigma_{\mathrm{sp}}}, d_{\mathrm{sp}} = \sigma_{\mathrm{sp}}^2/K_2^2$, there exists $\sigma_{\mathrm{sp}_1}$, such that for any $\sigma_{\mathrm{sp}} \geqslant \sigma_{\mathrm{sp}_1}$, target accuracy of CL is at least $0.5 \operatorname{erfc}\left(-L' \cdot \frac{\gamma}{\sqrt{2}\sigma_{\mathrm{sp}}}\right)$ and at most $0.5 \operatorname{erfc}\left(-4L' \cdot \frac{\gamma}{\sqrt{2}\sigma_{\mathrm{sp}}}\right)$, where $L' = \frac{K_2^2 K_1}{\sigma_{\mathrm{in}}^2(1 - 1/d_{\mathrm{in}})}$.

**Comparison with ERM.** Recall from Theorem 8 the performance of ERM classifier (trained from scratch) is $0.5 \operatorname{erfc}\left(-\gamma^2/\sqrt{2 d_{\mathrm{sp}}}\sigma_{\mathrm{sp}}\right)$. The lower bound on the performance of classifier over CL representations is strictly better than ERM when:

$$\frac{\gamma}{\sqrt{d_{\mathrm{sp}}}} < L'$$

$$\impliedby \frac{K_2^2 K_1}{\sigma_{\mathrm{in}}^2(1 - 1/d_{\mathrm{in}})} > \frac{\gamma}{\sqrt{d_{\mathrm{sp}}}} \impliedby \frac{K_2^2 K_1}{\sigma_{\mathrm{in}}^2(1 - 1/d_{\mathrm{in}})} > \frac{K_1 K_2^2}{\sigma_{\mathrm{sp}}^2}$$

$$\impliedby \sigma_{\mathrm{sp}} > \sigma_{\mathrm{in}}\sqrt{1 - 1/d_{\mathrm{in}}} \impliedby \sigma_{\mathrm{sp}_1} > \sigma_{\mathrm{in}}\sqrt{1 - 1/d_{\mathrm{in}}}.$$

This completes our proof of Corollary 6.

$\square$

### E.2 Analysis of STOC: Formal Statement of Theorem 7

Recall ERM solution over contrastive pretraining. We showed that without loss of generality when $k$ (the output dimensionality of $\Phi$) is greater than 2, we can restrict $k$ to 2 and the $\Phi$ can be denoted as $[\phi_1, \phi_2]^\top$ where $\phi_1 = c_1 w^\star + c_3 w_{\mathrm{sp}}$ and $\phi_2 = c_2 w^\star + c_4 w_{\mathrm{sp}}$. The ERM solution of the linear head is then given by $h_1, h_2 \in \mathbb{R}$:

$$h_1 = c_1 \cdot \gamma + c_3 \cdot \sqrt{d_{\mathrm{sp}}}, \text{ and } h_2 = c_2 \cdot \gamma + c_4 \cdot \sqrt{d_{\mathrm{sp}}}. \tag{121}$$

STOC performs self-training of the linear head over the CL solution. Before introducing the result, we need some additional notation. Let $h^t$ denote the solution of the linear head at iterate $t$. Without loss of generality, assume that the coefficients in $\phi_1 = c_1 w_{\mathrm{in}} + c_3 w_{\mathrm{sp}}$ and $\phi_2 = c_2 w_{\mathrm{in}} + c_4 w_{\mathrm{sp}}$ are such that $c_2$ is positive and $c_1, c_3$, and $c_4$ are negative. Moreover, for simplicity of exposition, assume that $|c_4| > |c_3|$.

**Theorem 16.** *Under the conditions of Corollary 15 and when $\frac{\gamma^2}{\sigma_{sp}} \geqslant \left[\frac{-c_3-c_4}{(c_2+c_1)\cdot|c_1|}\right] \vee \left[\frac{c_4}{c_1\cdot c_2}\right]$, the target accuracy of ST over CL is lower bounded by $0.5 \cdot \text{erfc}\left(-|c_2/c_4| \cdot \gamma/(\sqrt{2}\sigma_2)\right) \geqslant 0.5 \cdot \text{erfc}\left(-L \cdot \sqrt{d_{sp}}/(\sqrt{2}\sigma_{sp})\right)$ with $L \geqslant 1$.*

Before proving Theorem 16, we first connect the condition $\frac{\gamma^2}{\sigma_{sp}} \geqslant \left[\frac{-c_3-c_4}{(c_2+c_1)\cdot|c_1|}\right] \vee \left[\frac{c_4}{c_1\cdot c_2}\right]$ with the result obtained with contrastive learning.

**Remark 1.**   We first argue that $\left[\frac{-c_3-c_4}{(c_2+c_1)\cdot|c_1|}\right]$ term dominates and hence, if we have $\frac{\gamma^2}{\sigma_{sp}} \geqslant \left[\frac{-c_3-c_4}{(c_2+c_1)\cdot|c_1|}\right]$, then we get the result in Theorem 16. First, recall that as $\sigma_{sp}$ increases, we have $\left|\frac{c_3}{c_1}\right|$ converge to $\frac{2L\sigma_{in}^2(d_{in}-1)}{K_1K_2^2d_{in}}$, $c_2 \to 1$ and $\frac{c_1}{c_2} \to 0$. Using these limits, we get:

$$\frac{\gamma^2}{\sigma_{sp}} = \frac{K_1^2}{K_2 \cdot z^{3/2}} \geqslant \frac{2L\sigma_{in}^2(d_{in}-1)}{K_1K_2^2d_{in}} . \tag{122}$$

which reduces the following condition: $d_{sp} \leqslant K_1^2 K_2^{2/3} \cdot \left(\frac{d_{in}}{2L\sigma_{in}^2(d_{in}-1)}\right)^{2/3}$.

*Proof.* First, we create an outline of the proof. We argue about the updates of $h^t$ showing that both $h_1^t$ and $h_2^t$ increase with $|h_2^t|$ becoming greater than $|h_1^t|$ for some large $t$. Then we show that $|h_2^t| \geqslant |h_1^t|$ is sufficient to obtain near-perfect target generalization.

**Part 1.**   Recall the loss of used for self-training of $h$:

$$\mathcal{L}_{st}(h) = \mathbb{E}_{P_T(x)}\left[\ell(h^\top \Phi x, \text{sgn}(h^\top \Phi x))\right] \tag{123}$$

$$= \mathbb{E}_{P_T(x)}\left[\exp\left(-|h^\top \Phi x|\right)\right] \tag{124}$$

$$= \mathbb{E}_{z\sim\mathcal{N}(0,1)}\left[\exp\left(-|c_1\gamma h_1 + c_2\gamma h_2 + (c_3\sigma_{sp}h_1 + c_4\sigma_{sp}h_2)\cdot z|\right)\right] . \tag{125}$$

Define $\mu_t = c_1\gamma h_1^t + c_2\gamma h_2^t$ and $\sigma_t = c_3\sigma_{sp}h_1^t + c_4\sigma_{sp}h_2^t$. With this notation, we can re-write the loss in (125) as $\mathcal{L}_{st}(h^t) = \mathbb{E}_{z\sim\mathcal{N}(0,\sigma_t^2)}\left[\exp\left(-|\mu_t + z|\right)\right]$.

Similar to the the treatment in Theorem 9, we now derive a closed-form expression of $\mathcal{L}_{st}(h^t)$ in Lemma 30:

$$\mathcal{L}_{st}(h^t) = \frac{1}{2}\left(\exp\left(\frac{\sigma_t^2}{2} - \mu_t\right)\cdot\text{erfc}\left(-\frac{\mu_t}{\sqrt{2}\sigma_t} + \frac{\sigma_t}{\sqrt{2}}\right) + \exp\left(\frac{\sigma_t^2}{2} + \mu_t\right)\cdot\text{erfc}\left(\frac{\mu_t}{\sqrt{2}\sigma_t} + \frac{\sigma_t}{\sqrt{2}}\right)\right) . \tag{126}$$

Define:

$$A_1(\mu_t, \sigma_t) = \exp\left(\frac{\sigma_t^2}{2} - \mu_t\right)\cdot\text{erfc}\left(-\frac{\mu_t}{\sqrt{2}\sigma_t} + \frac{\sigma_t}{\sqrt{2}}\right)$$

$$= \sqrt{\frac{2}{\pi}}\exp\left(-\frac{\mu_t^2}{2\sigma_t^2}\right)\text{r}\left(\sigma_t - \frac{\mu_t}{\sigma_t}\right) , \tag{127}$$

$$A_2(\mu_t, \sigma_t) = \exp\left(\frac{\sigma_t^2}{2} + \mu_t\right)\cdot\text{erfc}\left(\frac{\mu_t}{\sqrt{2}\sigma_t} + \frac{\sigma_t}{\sqrt{2}}\right)$$

$$= \sqrt{\frac{2}{\pi}}\exp\left(-\frac{\mu_t^2}{2\sigma_t^2}\right)\text{r}\left(\sigma_t + \frac{\mu_t}{\sigma_t}\right) , \tag{128}$$

$$A_3(\mu_t, \sigma_t) = \frac{2\sqrt{2}}{\sqrt{\pi}}\exp\left(-\frac{\mu_t^2}{2\sigma_t^2}\right) . \tag{129}$$

Let $\widetilde{h}^{t+1}$ denote the un-normalized gradient descent update at iterate $t+1$. We have:

$$\widetilde{h}^{t+1} = h^t - \eta \cdot \frac{\partial \mathcal{L}_{\mathrm{st}}(h^t)}{\partial h}. \tag{130}$$

Now we will individually argue about the update of $\widetilde{h}^{t+1}$. First, we have:

$$\widetilde{h}_1^{t+1} = h_1^t - \eta \cdot \frac{\partial \mathcal{L}_{\mathrm{st}}(h^t)}{\partial h_1}$$

$$\widetilde{h}_1^{t+1} = h_1^t - \eta \cdot \underbrace{\left[ A_1 \cdot (\sigma_t c_3 \sigma_{\mathrm{sp}} - c_1 \gamma) + A_2 \cdot (\sigma_t c_3 \sigma_{\mathrm{sp}} + c_1 \gamma) - A_3 c_3 \sigma_{\mathrm{sp}} \right]}_{\delta_1}. \tag{131}$$

and second, we have:

$$\widetilde{h}_2^{t+1} = h_2^t - \eta \cdot \frac{\partial \mathcal{L}_{\mathrm{st}}(h^t)}{\partial h_2}$$

$$\widetilde{h}_2^{t+1} = h_2^t - \eta \cdot \underbrace{\left[ A_1 \cdot (\sigma_t c_4 \sigma_{\mathrm{sp}} - c_2 \gamma) + A_2 \cdot (\sigma_t c_4 \sigma_{\mathrm{sp}} + c_2 \gamma) - A_3 c_4 \sigma_{\mathrm{sp}} \right]}_{\delta_2}. \tag{132}$$

We will now argue the conditions under which $h_2^{t+1}$ increases till its value reaches $1/\sqrt{2}$. In particular, we will argue that when $h_2^t$ is negative, the norm $|h_2^t|$ decreases and when $h_2^t$ becomes positive, then its norm increases. We show that the following three conditions are sufficient to argue the increasing value of $h_2^t$: for all $t$, we have (i) $\mu_t \geqslant \mu_c$ and $|\sigma_t| < \sigma_c$ for constant $\mu_c = |c_1 \cdot \gamma|/2$ and $\sigma_c = |c_4 \sigma_{\mathrm{sp}}|$; (ii) $\delta_2 < 0$; (iii) $|\delta_2| \geqslant \delta_1$. In Lemma 18, we argue that our assumption on the initialization of the backbone learned with BT implies the previous three conditions.

**Case-1.** When $h_2^t$ is negative (and after the update, it remains negative). Then we want to argue the following:

$$\frac{(h_2^t - \eta \delta_2)^2}{(h_2^t - \eta \delta_2)^2 + (h_1^t - \eta \delta_1)^2} \leqslant (h_2^t)^2 \tag{133}$$

$$\Rightarrow \qquad \frac{(h_2^t - \eta \delta_2)^2}{(h_2^t)^2} \leqslant (h_2^t - \eta \delta_2)^2 + (h_1^t - \eta \delta_1)^2 \tag{134}$$

$$\Rightarrow \qquad \frac{h_2^{t\,2} + \eta^2 \delta_2^2 - 2\eta \delta_2 h_2^t}{(h_2^t)^2} \leqslant h_2^{t\,2} + \eta^2 \delta_2^2 - 2\eta h_2^t \delta_2 + h_1^{t\,2} + \eta^2 \delta_1^2 - 2\eta h_1^t \delta_1 \tag{135}$$

$$\Rightarrow \qquad 1 + \frac{\eta^2 \delta_2^2 - 2\eta \delta_2 h_2^t}{(h_2^t)^2} \leqslant 1 + \eta^2 \delta_2^2 - 2\eta h_2^t \delta_2 + \eta^2 \delta_1^2 - 2\eta h_1^t \delta_1 \tag{136}$$

$$\Rightarrow \qquad \eta^2 \delta_2^2 - 2\eta \delta_2 h_2^t \leqslant \left[ \eta^2 \delta_2^2 - 2\eta h_2^t \delta_2 + \eta^2 \delta_1^2 - 2\eta h_1^t \delta_1 \right] (h_2^t)^2 \tag{137}$$

$$\Rightarrow \qquad \eta^2 \delta_2^2 (h_1^t)^2 - 2\eta \delta_2 h_2^t (h_1^t)^2 \leqslant \eta^2 \delta_1^2 (h_2^t)^2 - 2\eta h_1^t \delta_1 (h_2^t)^2 \tag{138}$$

$$\Rightarrow \qquad \eta^2 \delta_2^2 (h_1^t)^2 - \eta^2 \delta_1^2 (h_2^t)^2 \leqslant 2\eta \delta_2 h_2^t (h_1^t)^2 - 2\eta h_1^t \delta_1 (h_2^t)^2 \tag{139}$$

$$\Rightarrow \quad \left[ \eta \delta_2 (h_1^t) - \eta \delta_1 (h_2^t) \right] \left[ \eta \delta_2 (h_1^t) + \eta \delta_1 (h_2^t) \right] \leqslant 2 h_2^t h_1^t \left[ \eta \delta_2 (h_1^t) - \eta \delta_1 (h_2^t) \right] \tag{140}$$

$$\Rightarrow \qquad \left[ \eta \delta_2 (h_1^t) + \eta \delta_1 (h_2^t) \right] \leqslant 2 h_2^t h_1^t \tag{141}$$

Since $\delta_2 < 0$, $|\delta_2| \geqslant |\delta_1|$ and $h_2^t < h_1^t < 0$, we have $\left[ \eta \delta_2 (h_1^t) - \eta \delta_1 (h_2^t) \right]$ as positive. This implies inequality (140) to (141) and for small enough $\eta$, (141) will continue to hold true.

**Case-2.** When $h_2^t$ is positive but less than $1/\sqrt{2}$. Then we want to argue the following:

$$\frac{(h_2^t - \eta\delta_2)^2}{(h_2^t - \eta\delta_2)^2 + (h_1^t - \eta\delta_1)^2} \geqslant (h_2^t)^2 \tag{142}$$

$$\Rightarrow \qquad \frac{(h_2^t - \eta\delta_2)^2}{(h_2^t)^2} \geqslant (h_2^t - \eta\delta_2)^2 + (h_1^t - \eta\delta_1)^2 \tag{143}$$

$$\Rightarrow \qquad \frac{{h_2^t}^2 + \eta^2\delta_2^2 - 2\eta\delta_2 h_2^t}{(h_2^t)^2} \geqslant {h_2^t}^2 + \eta^2\delta_2^2 - 2\eta h_2^t\delta_2 + {h_1^t}^2 + \eta^2\delta_1^2 - 2\eta h_1^t\delta_1 \tag{144}$$

$$\Rightarrow \qquad 1 + \frac{\eta^2\delta_2^2 - 2\eta\delta_2 h_2^t}{(h_2^t)^2} \geqslant 1 + \eta^2\delta_2^2 - 2\eta h_2^t\delta_2 + \eta^2\delta_1^2 - 2\eta h_1^t\delta_1 \tag{145}$$

$$\Rightarrow \qquad \eta^2\delta_2^2 - 2\eta\delta_2 h_2^t \geqslant \left[\eta^2\delta_2^2 - 2\eta h_2^t\delta_2 + \eta^2\delta_1^2 - 2\eta h_1^t\delta_1\right](h_2^t)^2 \tag{146}$$

$$\Rightarrow \qquad \eta^2\delta_2^2(h_1^t)^2 - 2\eta\delta_2 h_2^t(h_1^t)^2 \geqslant \eta^2\delta_1^2(h_2^t)^2 - 2\eta h_1^t\delta_1(h_2^t)^2 \tag{147}$$

$$\Rightarrow \qquad \eta^2\delta_2^2(h_1^t)^2 - \eta^2\delta_1^2(h_2^t)^2 \geqslant 2\eta\delta_2 h_2^t(h_1^t)^2 - 2\eta h_1^t\delta_1(h_2^t)^2 \tag{148}$$

$$\Rightarrow \quad \left[\eta\delta_2(h_1^t) - \eta\delta_1(h_2^t)\right]\left[\eta\delta_2(h_1^t) + \eta\delta_1(h_2^t)\right] \geqslant 2h_2^t h_1^t\left[\eta\delta_2(h_1^t) - \eta\delta_1(h_2^t)\right] \tag{149}$$

$$\Rightarrow \qquad \left[\eta\delta_2(h_1^t) + \eta\delta_1(h_2^t)\right] \geqslant 2h_2^t h_1^t \tag{150}$$

Since $\delta_2 < 0$, $|\delta_2| \geqslant |\delta_1|$, $h_1^t \leqslant -1/\sqrt{2}$ and $0 < h_2^t < 1/\sqrt{2}$, we have $\left[\eta\delta_2(h_1^t) - \eta\delta_1(h_2^t)\right]$ as positive. This implies inequality (149) to (150). Focusing on (150), we note that $h_1^t \cdot \delta_2$ is positive and greater in magnitude than $h_2^t \cdot \delta_1$. Moreover, since $h_2^t h_1^t$ is negative, (150) will continue to hold true.

Now, when $h_2^t$ is positive and greater than $1/\sqrt{2}$, then $h_2^t$ will stay in that region. Convergence of STOC together with conditions of convergence as in Lemma 17 will imply that the at convergence $h_2^t$ will remain greater than $1/\sqrt{2}$, such that $\frac{h_1^{t_c}}{h_2^{t_c}} = \frac{\delta_1}{\delta_2}$. Now we bound the target error of STOC.

**Part 2.** To bound the accuracy at any iterate $t$ when $h_2^t \geqslant 1/\sqrt{2}$, we have from Lemma 29:

$$\mathbb{E}_{\mathrm{P_T}}\left[y \cdot \left({h^t}^\top \phi_{\mathrm{cl}} x\right) > 0\right] = \mathbb{E}_{z \sim \mathcal{N}(0,1)}\left[z > -\frac{c_1\gamma h_1^t + c_2\gamma h_2^t}{|c_3\sigma_{\mathrm{sp}}h_1^t + c_4\sigma_{\mathrm{sp}}h_2^t|}\right]. \tag{151}$$

We now upper bound and lower bound the fraction $\frac{c_1\gamma h_1^t + c_2\gamma h_2^t}{|c_3\sigma_{\mathrm{sp}}h_1^t + c_4\sigma_{\mathrm{sp}}h_2^t|}$ in RHS in (151): (i) $c_1\gamma h_1^t + c_2\gamma h_2^t \geqslant c_2\gamma h_2^t$ since both $c_1\gamma h_1^t$ and $c_2\gamma h_2^t$ have same sign; (ii) $|c_3\sigma_{\mathrm{sp}}h_1^t + c_4\sigma_{\mathrm{sp}}h_2^t| \leqslant |c_4\sigma_{\mathrm{sp}}h_2^t|$ because $|c_4\sigma_{\mathrm{sp}}h_2^t| \geqslant |c_3\sigma_{\mathrm{sp}}h_1^t|$ and they have opposite signs. Hence, from (151), we have:

$$\mathbb{E}_{\mathrm{P_T}}\left[y \cdot \left({h^t}^\top \phi_{\mathrm{cl}} x\right) > 0\right] = \mathbb{E}_{z \sim \mathcal{N}(0,1)}\left[z > -\frac{c_2\gamma h_2^t}{|c_4\sigma_{\mathrm{sp}}h_2^t|}\right] = \mathbb{E}_{z \sim \mathcal{N}(0,1)}\left[z > -\frac{c_2\gamma}{|c_4\sigma_{\mathrm{sp}}|}\right]. \tag{152}$$

Substituting the definition of $\mathrm{erfc}$, the expression (152) gives us the required lower bound on the target accuracy.

$$\square$$

**Lemma 17** (Convergence of STOC). *Assume the gradient updates as in* (131) *and* (132). *Then STOC converges at $t = t_c$ when $\frac{h_1^{t_c}}{h_2^{t_c}} = \frac{\delta_1}{\delta_2}$. For $t > t_c$,* (131) *and* (132) *make no updates to the linear $h$.*

*Proof.* When the gradient updates $\delta_1$ and $\delta_2$ are such that $h_1^{t+1}$ matches $h_1^t$, we have convergence of STOC.

$$\frac{(h_2^t - \eta\delta_2)^2}{(h_2^t - \eta\delta_2)^2 + (h_1^t - \eta\delta_1)^2} = (h_2^t)^2 \tag{153}$$

$$\Rightarrow \qquad \frac{(h_2^t - \eta\delta_2)^2}{(h_2^t)^2} = (h_2^t - \eta\delta_2)^2 + (h_1^t - \eta\delta_1)^2 \tag{154}$$

$$\Rightarrow \qquad \frac{{h_2^t}^2 + \eta^2\delta_2^2 - 2\eta\delta_2 h_2^t}{(h_2^t)^2} = {h_2^t}^2 + \eta^2\delta_2^2 - 2\eta h_2^t\delta_2 + {h_1^t}^2 + \eta^2\delta_1^2 - 2\eta h_1^t\delta_1 \tag{155}$$

$$\Rightarrow \qquad 1 + \frac{\eta^2\delta_2^2 - 2\eta\delta_2 h_2^t}{(h_2^t)^2} = 1 + \eta^2\delta_2^2 - 2\eta h_2^t\delta_2 + \eta^2\delta_1^2 - 2\eta h_1^t\delta_1 \tag{156}$$

$$\Rightarrow \qquad \eta^2\delta_2^2 - 2\eta\delta_2 h_2^t = \left[\eta^2\delta_2^2 - 2\eta h_2^t\delta_2 + \eta^2\delta_1^2 - 2\eta h_1^t\delta_1\right](h_2^t)^2 \tag{157}$$

$$\Rightarrow \qquad \eta^2\delta_2^2(h_1^t)^2 - 2\eta\delta_2 h_2^t(h_1^t)^2 = \eta^2\delta_1^2(h_2^t)^2 - 2\eta h_1^t\delta_1(h_2^t)^2 \tag{158}$$

$$\Rightarrow \qquad \eta^2\delta_2^2(h_1^t)^2 - \eta^2\delta_1^2(h_2^t)^2 = 2\eta\delta_2 h_2^t(h_1^t)^2 - 2\eta h_1^t\delta_1(h_2^t)^2 \tag{159}$$

$$\Rightarrow \quad \left[\eta\delta_2(h_1^t) - \eta\delta_1(h_2^t)\right]\left[\eta\delta_2(h_1^t) + \eta\delta_1(h_2^t)\right] = 2h_2^t h_1^t\left[\eta\delta_2(h_1^t) - \eta\delta_1(h_2^t)\right] \tag{160}$$

Thus either $\left[\eta\delta_2(h_1^t) - \eta\delta_1(h_2^t)\right] = 0$ or $\left[\eta\delta_2(h_1^t) + \eta\delta_1(h_2^t)\right] = 2h_2^t h_1^t$. Since $\eta$ is such that $h_1 - \eta\delta_1 < 0$, $\left[\eta\delta_2(h_1^t) + \eta\delta_1(h_2^t)\right] \neq 2h_2^t h_1^t$ implying that $\left[\eta\delta_2(h_1^t) - \eta\delta_1(h_2^t)\right] = 0$ giving us the required condition. $\qquad\square$

**Lemma 18.** *Under the initialization conditions assumed in Theorem 16, for all $t$, we have: (i) $\mu_t \geqslant \mu_c$ and $|\sigma_t| \leqslant \sigma_c$ for constant $\mu_c = |c_1 \cdot \gamma|/2$ and $\sigma_c = |c_4\sigma_{\mathrm{sp}}|$; (ii) $\delta_2 < 0$; (iii) $|\delta_2| \geqslant \delta_1$, where $\delta_1 = A_1 \cdot (\sigma_t c_3\sigma_{\mathrm{sp}} - c_1\gamma) + A_2 \cdot (\sigma_t c_3\sigma_{\mathrm{sp}} + c_1\gamma) - A_3 c_3\sigma_{\mathrm{sp}}$ and $\delta_2 = A_1 \cdot (\sigma_t c_4\sigma_{\mathrm{sp}} - c_2\gamma) + A_2 \cdot (\sigma_t c_4\sigma_{\mathrm{sp}} + c_2\gamma) - A_3 c_4\sigma_{\mathrm{sp}}$ for $A_1, A_2$ and $A_3$ defined in (127), (128), and (129).*

*Proof.* Recall, $\mu_t = c_1\gamma h_1^t + c_2\gamma h_2^t$ and $\sigma_t = c_3\sigma_{\mathrm{sp}} h_1^t + c_4\sigma_{\mathrm{sp}} h_2^t$. First, we argue that $\mu_t$ increases from the initialization value. Notice that $\mu_0 = c_1\gamma h_1^0 + c_2\gamma h_2^0$. Due to Corollary 15, we have $h_2^0 \geqslant 0$. And since $|c_2| > |c_1|$, we get $\mu_0 \geqslant |c_1\gamma|$ as both $c_1$ and $h_1^0$ are of same sign. Moreover, as training progresses with $h_1^t$ remaining negative and $h_2^t$ remaining positive, we have $\mu_t$ stays greater than $\mu_0$.

Recall the definition of $A_1, A_2$, and $A_3$ in (127), (128), and (129). Moreover, recall the definition of $\alpha_1(\mu_t, \sigma_t)$ and $\alpha_2(\mu_t, \sigma_t)$:

$$\alpha_1(\mu_t, \sigma_t) = \sqrt{\frac{2}{\pi}}\exp\left(-\frac{\mu_t^2}{2\sigma_t^2}\right)\left[\mathrm{r}\left(\sigma_t + \frac{\mu_t}{\sigma_t}\right) - \mathrm{r}\left(\sigma_t - \frac{\mu_t}{\sigma_t}\right)\right]. \tag{161}$$

and

$$\alpha_2(\mu_t, \sigma_t) = \sqrt{\frac{2}{\pi}}\exp\left(-\frac{\mu_t^2}{2\sigma_t^2}\right)\left[\mathrm{r}\left(\sigma_t + \frac{\mu_t}{\sigma_t}\right) + \mathrm{r}\left(\sigma_t - \frac{\mu_t}{\sigma_t}\right) - \frac{2}{\sigma_t}\right]. \tag{162}$$

Thus, we have $\alpha_1(\mu_t, \sigma_t) \cdot A_3 = A_1 \cdot \sigma_t$ and $\alpha_2(\mu_t, \sigma_t) \cdot A_3 = \sigma_t \cdot \left(A_2 \cdot -\frac{2}{\sigma_t}A_3\right)$. Replacing the definition of $A_1, A_2$, and $A_3$ in $\delta_1$ and $\delta_2$, we get:

$$\delta_1 = \sigma_t c_3\sigma_{\mathrm{sp}} \cdot \alpha_2(\mu_t, \sigma_t) + c_1\gamma\alpha_1(\mu_t, \sigma_t) \quad \text{and} \quad \delta_2 = \sigma_t c_4\sigma_{\mathrm{sp}} \cdot \alpha_2(\mu_t, \sigma_t) + c_2\gamma\alpha_1(\mu_t, \sigma_t) \tag{163}$$

We now upper bound and lower bound $\alpha_1$ and $\alpha_2$ by using the properties of $\mathrm{r}(\cdot)$. We use Taylor's expansion on $\mathrm{r}(\cdot)$ and we get:

$$\mathrm{r}(\sigma_t) + \mathrm{r}'(\sigma_t) \cdot \left(\frac{\mu_t}{\sigma_t}\right) \leqslant \mathrm{r}\left(\sigma_t + \frac{\mu_t}{\sigma_t}\right) \leqslant \mathrm{r}(\sigma_t) + \mathrm{r}'(\sigma_t) \cdot \left(\frac{\mu_t}{\sigma_t}\right) + \mathrm{r}''(\sigma_t) \cdot \left(\frac{\mu_t}{\sigma_t}\right)^2 \tag{164}$$

and similarly, we get:

$$\mathrm{r}(\sigma_t) - \mathrm{r}'(\sigma_t) \cdot \left(\frac{\mu_t}{\sigma_t}\right) + \mathrm{r}''(\sigma_t) \cdot \left(\frac{\mu_t}{\sigma_t}\right)^2 \leqslant \mathrm{r}\left(\sigma_t - \frac{\mu_t}{\sigma_t}\right) \leqslant \mathrm{r}(\sigma_t) - \mathrm{r}'(\sigma_t) \cdot \left(\frac{\mu_t}{\sigma_t}\right) + R''\left(\frac{\mu_t}{\sigma_t}\right)^2 \tag{165}$$

where $R'' = \text{r}''(\sigma_0)$. This is because $\text{r}''(\cdot)$ takes positive values and is a decreasing function in $\sigma_t$ (refer to Lemma 21). We now lower bound $\alpha_1(\mu_t, \sigma_t)$ and upper bound $\alpha_2(\mu_t, \sigma_t)$:

$$\frac{\alpha_1(\mu_t, \sigma_t)}{\sqrt{\frac{2}{\pi}}\exp\left(-\frac{\mu_t^2}{2\sigma_t^2}\right)} \leqslant 2\text{r}'(\sigma_t) \cdot \left(\frac{\mu_t}{\sigma_t}\right) \tag{166}$$

$$\frac{\alpha_2(\mu_t, \sigma_t)}{\sqrt{\frac{2}{\pi}}\exp\left(-\frac{\mu_t^2}{2\sigma_t^2}\right)} \geqslant 2\text{r}(\sigma_t) + \text{r}''(\sigma_t) \cdot \left(\frac{\mu_t}{\sigma_t}\right)^2 - \frac{2}{\sigma_t} \tag{167}$$

**Part-1.** We first prove that $\delta_2 \leqslant 0$. Substituting the lower bound and upper bound in (163) gives us the following as stricter a sufficient condition (i.e., (168) implies $\delta_2 \leqslant 0$):

$$\left[2\text{r}(\sigma_t) + \text{r}''(\sigma_t) \cdot \left(\frac{\mu_t}{\sigma_t}\right)^2 - \frac{2}{\sigma_t}\right] \cdot \frac{\sigma_{\text{sp}} \cdot (-c_4)}{\gamma \cdot c_2} \geqslant 2\text{r}'(\sigma_t) \cdot \left(\frac{\mu_t}{\sigma_t}\right) \tag{168}$$

$$\Longleftrightarrow \left[2\text{r}(\sigma_t) + \text{r}''(\sigma_t) \cdot \left(\frac{\mu_t}{\sigma_t}\right)^2 - \frac{2}{\sigma_t}\right] \geqslant 2\text{r}'(\sigma_t) \cdot \left(\frac{\mu_t}{\sigma_t}\right) \cdot \frac{\gamma \cdot c_2}{\sigma_{\text{sp}} \cdot (-c_4)} \tag{169}$$

$$\Longleftrightarrow 2\text{r}(\sigma_t) + \text{r}''(\sigma_t) \cdot \left(\frac{\mu_t}{\sigma_t}\right)^2 - \frac{2}{\sigma_t} - 2\text{r}'(\sigma_t) \cdot \left(\frac{\mu_t}{\sigma_t}\right) \cdot \frac{\gamma \cdot c_2}{\sigma_{\text{sp}} \cdot (-c_4)} \geqslant 0 \tag{170}$$

$$\Longleftrightarrow 2\text{r}(\sigma_t) \cdot \sigma_t + \text{r}''(\sigma_t) \cdot \frac{\mu_t^2}{\sigma_t} - 2 - 2\text{r}'(\sigma_t) \cdot \mu_t \cdot \frac{\gamma \cdot c_2}{\sigma_{\text{sp}} \cdot (-c_4)} \geqslant 0 \tag{171}$$

$$\Longleftrightarrow 2\text{r}'(\sigma_t) + \text{r}''(\sigma_t) \cdot \frac{\mu_t^2}{\sigma_t} - 2\text{r}'(\sigma_t) \cdot \mu_t \cdot \frac{\gamma \cdot c_2}{\sigma_{\text{sp}} \cdot (-c_4)} \geqslant 0 \tag{172}$$

$$\Longleftrightarrow \text{r}''(\sigma_t) \cdot \frac{\mu_t^2}{\sigma_t} + 2\text{r}'(\sigma_t) \cdot \left[1 - \mu_t \cdot \frac{\gamma \cdot c_2}{\sigma_{\text{sp}} \cdot (-c_4)}\right] \geqslant 0 \tag{173}$$

Thus, if we have $\mu_t \geqslant \frac{\sigma_{\text{sp}} \cdot (-c_4)}{\gamma \cdot c_2}$, then (168) holds true.

**Part-2.** Next, we prove that $|\delta_2| \geqslant \delta_1$. Substituting the lower bound and upper bound in (163) gives us the following as stricter a sufficient condition (i.e., (174) implies $|\delta_2| \geqslant \delta_1$):

$$\left[2\text{r}(\sigma_t) + \text{r}''(\sigma_t) \cdot \left(\frac{\mu_t}{\sigma_t}\right)^2 - \frac{2}{\sigma_t}\right] \cdot \frac{\sigma_{\text{sp}} \cdot (-c_4 - c_3)}{\gamma \cdot (c_2 + c_1)} \geqslant 2\text{r}'(\sigma_t) \cdot \left(\frac{\mu_t}{\sigma_t}\right) \tag{174}$$

$$\Longleftrightarrow \left[2\text{r}(\sigma_t) + \text{r}''(\sigma_t) \cdot \left(\frac{\mu_t}{\sigma_t}\right)^2 - \frac{2}{\sigma_t}\right] \geqslant 2\text{r}'(\sigma_t) \cdot \left(\frac{\mu_t}{\sigma_t}\right) \cdot \frac{\gamma \cdot (c_2 + c_1)}{\sigma_{\text{sp}} \cdot (-c_4 - c_3)} \tag{175}$$

$$\Longleftrightarrow 2\text{r}(\sigma_t) + \text{r}''(\sigma_t) \cdot \left(\frac{\mu_t}{\sigma_t}\right)^2 - \frac{2}{\sigma_t} - 2\text{r}'(\sigma_t) \cdot \left(\frac{\mu_t}{\sigma_t}\right) \cdot \frac{\gamma \cdot (c_2 + c_1)}{\sigma_{\text{sp}} \cdot (-c_4 - c_3)} \geqslant 0 \tag{176}$$

$$\Longleftrightarrow 2\text{r}(\sigma_t) \cdot \sigma_t + \text{r}''(\sigma_t) \cdot \frac{\mu_t^2}{\sigma_t} - 2 - 2\text{r}'(\sigma_t) \cdot \mu_t \cdot \frac{\gamma \cdot (c_2 + c_1)}{\sigma_{\text{sp}} \cdot (-c_4 - c_3)} \geqslant 0 \tag{177}$$

$$\Longleftrightarrow 2\text{r}'(\sigma_t) + \text{r}''(\sigma_t) \cdot \frac{\mu_t^2}{\sigma_t} - 2\text{r}'(\sigma_t) \cdot \mu_t \cdot \frac{\gamma \cdot (c_2 + c_1)}{\sigma_{\text{sp}} \cdot (-c_4 - c_3)} \geqslant 0 \tag{178}$$

$$\Longleftrightarrow \text{r}''(\sigma_t) \cdot \frac{\mu_t^2}{\sigma_t} + 2\text{r}'(\sigma_t) \cdot \left[1 - \mu_t \cdot \frac{\gamma \cdot (c_2 + c_1)}{\sigma_{\text{sp}} \cdot (-c_4 - c_3)}\right] \geqslant 0 \tag{179}$$

Thus, if we have $\mu_t \geqslant \frac{\sigma_{\text{sp}} \cdot (-c_4 - c_3)}{\gamma \cdot (c_2 + c_1)}$, then (174) holds true which in-turn implies $|\delta_2| \geqslant \delta_1$. Plugging in $\mu_t \geqslant \mu_0$, we get the required condition.

$\square$

### E.3 Analysis for SSL

For SSL analysis, we argue that the projection learned by contrastive pretraining can significantly improve the generalization of the linear head learned on top, leaving little to no room for improvement for self-training. Our analysis leverages the margin-based bound for linear models from Kakade et al. [45]. Before introducing the result, we present some additional notation. Let $\mathrm{Err}_D(w)$ denote 0-1 error of a classifier on a distribution $D$. Define 0-1 error with margin $\gamma$ as $\widehat{\mathrm{Err}}^\gamma(w) = \sum_{i=1}^{n} \frac{\mathbb{I}\left[y_i w^\top x_i \leqslant \gamma\right]}{n}$.

**Theorem 19** (Corollary 6 in Kakade et al. [45]). *For all classifiers $w$ and margin $\gamma$, we have with probability at least $1 - \delta$:*

$$\mathrm{Err}_T(w) \leqslant \widehat{\mathrm{Err}}^\gamma(w) + 4\frac{B}{\gamma}\sqrt{\frac{1}{n}} + \sqrt{\frac{\log(1/\delta)}{n}} + \sqrt{\frac{\log(\log_2(4B/\gamma))}{n}}, \qquad (180)$$

*where $B$ is an upper bound on the $\ell_2$ norm of the input points $x$.*

When $\widetilde{\mathrm{Err}}^\gamma(w)$ is close to zero, the denominating term in RHS of (180) is $4\frac{B}{\gamma}\sqrt{\frac{1}{n}}$. With Proposition 3, CL solution $\phi$ obtained on the target domain alone (for SSL setup) is $w_{\mathrm{in}}$ when $k = 1$. For larger $k$'s the CL solution is dominated by the $\phi_1 = w_{\mathrm{in}}$. Thus, SSL mainly reduces the B on the projected data by reducing the dependency from order $\sqrt{d}$ to 1 where 1 is the dimensionality of the output of $\phi$ without altering the margin. Thus, we get a tighter upper bound for linear probing performed on top CL features when compared with linear probing done on inputs directly.

Intuitively, since the target data has only one predictive feature (along $w_{\mathrm{in}}$), CL directly recovers this predictive feature as it is the predominant direction that minimizes invariance loss.

## F  Limitations of Prior Work

### F.1  Contrastive learning analysis

Prior works that analyze contrastive learning show that minimizers of the CL objective recover clusters in the augmentation graph, which weights pairs of augmentations with their probability of being sampled as a positive pair [39, 11, 73, 44]. When there is no distribution shift in the downstream task, assumptions made on the graph in the form of consistency of augmentations with downstream labels, is sufficient to ensure that a linear probed head has good ID generalization. Under distribution shift, these assumptions are not sufficient and stronger ones are needed. *E.g.*, some works assume that same-domain/class examples are weighted higher that cross-class cross-domain pairs [40, 76].

Using notation defined in [76], the assumption on the augmentation graph requires cross-class and same-domain weights ($\beta$) to be higher than cross-class and cross-domain weights ($\gamma$). It is unclear if examples from different classes in the same domain will be "connected" if strong spurious features exist in the source domain and augmentations fail to mask them completely (*e.g.*, image background may not be completely masked by augmentations but it maybe perfectly predictive of the label on source domain). In such cases, the linear predictor learnt over CL would fail to generalize OOD. In our toy setup as well, the connectivity assumption fails since on source $x_{\mathrm{sp}}$ is perfectly predictive of the label and the augmentations are imperfect, *i.e.*, augmentations do not mask $x_{\mathrm{sp}}$ and examples of different classes do not overlap in source (*i.e.*, $\beta = 0$). On the other hand, since $x_{\mathrm{sp}}$ is now random on target, augmentations of different classes may overlap, *i.e.*, $\gamma > 0$, thus breaking the connectivity assumption. This is also highlighted in our empirical findings of CL furnishing representations that do not fully enable linear transferability from source to target (see Sec. 5). These empirical findings also call into question existing assumptions on data augmentations, highlighting that perfect linear transferability may not typically hold in practice. It is in this setting that we believe self-training can improve over contrastive learning by unlearning source-only features and improving linear transferability.

### F.2  Self-training analysis

Some prior works on self-training view it as consistency regularization that constrain pseudolabels of original samples to be consistent with all their augmentations [12, 87, 79]. This framework

abstracts the role played by the optimization algorithm and instead evaluates the global minimizer of a population objective that enforces consistency of pseudolabels. In addition, certain expansion assumptions on class-conditional distributions are needed to ensure that pseudolabels have good accuracy on source and target domains. This framework does not account for challenges involved in propagating labels iteratively. For *e.g.*, when augmentation distribution has long tails, the consistency of pseudolabels depends on the sampling frequency of "favorable" augmentations. As an illustration, consider our augmentation distribution in the toy setup in Sec. 4. If it were not uniform over dimensions, but instead something that was highly skewed, then a large number of augmentations need to be sampled for every data point to propagate pseudolabels successfully from source labeled samples to target unlabeled samples during self-training. This might hurt the performance of ST when we are optimizing for only finitely many iterations and over finitely many datapoints. This is why in our analysis we instead adopt the iterative analysis of self-training [17].

## G   Additional Lemmas

In this section we define some additional lemmas that we use in our theoretical analysis in E.

**Lemma 20** (Upper bound and lower bounds on $\mathrm{erfc}$; Kschischang [49]). *Define* $\mathrm{erfc}(x) = \frac{2}{\sqrt{\pi}} \cdot \int_x^\infty \exp(-z^2) \cdot dz$. *Then we have:*

$$\frac{2}{\sqrt{\pi}} \cdot \frac{\exp(-x^2)}{x + \sqrt{x^2 + 2}} < \mathrm{erfc}(x) \leqslant \frac{2}{\sqrt{\pi}} \cdot \frac{\exp(-x^2)}{x + \sqrt{x^2 + 4/\pi}}$$

**Lemma 21** (Properties of Mill's ratio [5]). *Define the Mill's ratio as* $\mathrm{r}(x) = \exp\left(x^2/2\right) \cdot \mathrm{erfc}\left(x/\sqrt{2}\right) \cdot \sqrt{\pi/2}$. *Then following assertions are true: (i)* $\mathrm{r}(x)$ *is a strictly decreasing log-convex function; (ii)* $\mathrm{r}'(x) = x \cdot \mathrm{r}(x) - 1$ *is an increasing function with* $\mathrm{r}'(x) < 0$ *for all x; (iii)* $\mathrm{r}''(x) = \mathrm{r}(x) + x^2 \cdot \mathrm{r}(x) - x$ *is a decreasing function with* $\mathrm{r}''(x) > 0$ *for all x; (iv)* $x^2 \cdot \mathrm{r}'(x)$ *is a decreasing function of* $x$.

**Lemma 22** (invariance loss as product with operator $L$). *The invariance loss for some* $\phi \in \mathbb{R}^d$ *is given as:* $2 \cdot \int_{\mathcal{A}} \phi(a) \cdot L(\phi)(a) \, \mathrm{dP_A}$ *where the operator $L$ is defined as:*

$$L(\phi)(a) = \phi(a) - \int_{\mathcal{A}} \frac{A_+(a, a')}{p_\mathsf{A}(a)} \cdot \phi(a') \, \mathrm{d}a'$$

*Proof.* The invariance loss for $\phi$ is given by:

$$\mathbb{E}_{x \sim \mathrm{P_U}} \mathbb{E}_{a_1, a_2 \sim \mathrm{P_A}(\cdot|x)} (a_1^\top \phi - a_2^\top \phi)^2 = 2 \mathbb{E}_{x \sim \mathrm{P_U}} \mathbb{E}_{a \sim \mathrm{P_A}(\cdot|x)} \left[\phi(a)^2\right]$$
$$- 2 \mathbb{E}_{a_1, a_2 \sim A_+(\cdot, \cdot)} \left[\phi(a_1)\phi(a_2)\right] \quad (181)$$

$$= 2 \cdot \int_{\mathcal{A}} \phi(a)^2 \, \mathrm{dP_A} - 2 \cdot \int_{\mathcal{A}} \phi(a) \left(\int_{\mathcal{A}} \frac{A_+(a, a_2)}{p_\mathsf{A}(a)} \cdot \phi(a_2) \, \mathrm{d}a_2\right) \mathrm{dP_A} \quad (182)$$

$$= 2 \cdot \int_{\mathcal{A}} \phi(a) \cdot L(\phi)(a) \, \mathrm{dP_A} \quad (183)$$

$\square$

**Lemma 23.** *If $\mathcal{W}$ is the space spanned by $w_{\mathrm{in}}$ and $w_{\mathrm{sp}}$, and $\mathcal{W}_\perp$ is the null space for $\mathcal{W}$, then for any $u \in \mathcal{W}$ and any $v \in \mathcal{W}_\perp$, the covariance along these directions $\mathbb{E}_{a \sim \mathrm{P_A}}[a^\top u v^\top a] = 0$.*

*Proof:* We can write the covariance over augmentations after we break down the augmentation $a$ into two projections: $a = \Pi_{\mathcal{W}}(a) + \Pi_{\mathcal{W}_\perp}(a)$

$$\mathbb{E}_{a \sim \mathrm{P_A}}[a^\top u v^\top a] = \mathbb{E}_{a \sim \mathrm{P_A}} \left[\left(u^\top (\Pi_{\mathcal{W}}(a) + \Pi_{\mathcal{W}_\perp}(a))\right)\left(v^\top (\Pi_{\mathcal{W}}(a) + \Pi_{\mathcal{W}_\perp}(a))\right)\right] \quad (184)$$

$$= \mathbb{E}_{a \sim \mathrm{P_A}} \left[\left(u^\top \Pi_{\mathcal{W}}(a)\right)\left(v^\top \Pi_{\mathcal{W}_\perp}(a)\right)\right] \quad (185)$$

$$= u^\top \left(\mathbb{E}_{a \sim \mathrm{P_A}} \left[\Pi_{\mathcal{W}}(a) \Pi_{\mathcal{W}_\perp}(a)^\top\right]\right) v = 0 \quad (186)$$

where the last inequality follows from the fact that $\mathbb{E}_{a \sim \mathrm{P_A}} \left[\Pi_{\mathcal{W}}(a) \Pi_{\mathcal{W}_\perp}(a)^\top\right] = \mathbb{E}_{a \sim \mathrm{P_A}} \left[\Pi_{\mathcal{W}}(a)\right] \mathbb{E}_{a \sim \mathrm{P_A}} \left[\Pi_{\mathcal{W}_\perp}(a)\right]^\top$, since the noise in the null space of $\mathcal{W}$ is drawn independent of the component along $\mathcal{W}$, and furthermore the individual expectations evaluate to zero.

**Lemma 24** (closed-form expressions for eigenvalues and eigenvectors of $\Sigma_A, \widetilde{\Sigma}$). *For a $2 \times 2$ real symmetric matrix $\begin{bmatrix} a, & b \\ c, & d \end{bmatrix}$ the eigenvalues $\lambda_1, \lambda_2$ are given by the following expressions:*

$$\lambda_1 = \frac{(a+b+\delta)}{2}, \quad \lambda_2 = \frac{(a+b-\delta)}{2},$$

*where $\delta = \sqrt{4c^2 + (a-b)^2}$. Further, the eigenvectors are given by $U = \begin{bmatrix} \cos(\theta), & \sin(\theta) \\ \sin(\theta), & -\cos(\theta) \end{bmatrix}$, where:*

$$\tan(\theta) = \frac{b - a + \delta}{2c}.$$

*For full proof of these statements see [23]. Here, we will use these statements to arrive at closed form expressions for the eigenvalues and eigenvectors of $\Sigma_A, \widetilde{\Sigma}$.*

*Proof.* We can now substitute the above formulae with $a, b, c, d$ taken from the expressions of $\Sigma_A$ and $\widetilde{\Sigma}$, to get the following values: $\lambda_1, \lambda_2$ are the eigenvalues of $\Sigma_A$, with $\alpha$ determining the corresponding eigenvectors $[\cos(\alpha), \sin(\alpha)], [\sin(\alpha), -\cos(\alpha)]$; and $\widetilde{\lambda}_1, \widetilde{\lambda}_2$ are the eigenvalues of $\widetilde{\Sigma}$, with $\beta$ determining the corresponding eigenvectors: $[\cos(\beta), \sin(\beta)], [\sin(\beta), -\cos(\beta)]$.

$$\lambda_1 = \frac{1}{8}\left(\gamma^2\left(1 + \frac{1}{3d_{\text{in}}}\right) + \frac{\sigma_{\text{in}}^2}{3}\left(1 - \frac{1}{d_{\text{in}}}\right) + \frac{d_{\text{sp}}}{2} + \frac{2\sigma_{\text{sp}}^2}{3} + \frac{1}{6}\right.$$
$$\left. + \sqrt{\gamma^2 d_{\text{sp}} + \left(\left(\gamma^2\left(1 + \frac{1}{3d_{\text{in}}}\right) + \frac{\sigma_{\text{in}}^2}{3}\left(1 - \frac{1}{d_{\text{in}}}\right)\right) - \left(\frac{d_{\text{sp}}}{2} + \frac{2\sigma_{\text{sp}}^2}{3} + \frac{1}{6}\right)\right)^2}\right) \quad (187)$$

$$\lambda_2 = \frac{1}{8}\left(\gamma^2\left(1 + \frac{1}{3d_{\text{in}}}\right) + \frac{\sigma_{\text{in}}^2}{3}\left(1 - \frac{1}{d_{\text{in}}}\right) + \frac{d_{\text{sp}}}{2} + \frac{2\sigma_{\text{sp}}^2}{3} + \frac{1}{6}\right.$$
$$\left. - \sqrt{\gamma^2 d_{\text{sp}} + \left(\left(\gamma^2\left(1 + \frac{1}{3d_{\text{in}}}\right) + \frac{\sigma_{\text{in}}^2}{3}\left(1 - \frac{1}{d_{\text{in}}}\right)\right) - \left(\frac{d_{\text{sp}}}{2} + \frac{2\sigma_{\text{sp}}^2}{3} + \frac{1}{6}\right)\right)^2}\right) \quad (188)$$

$$\widetilde{\lambda}_1 = \frac{1}{8}\left(\gamma^2 + \frac{d_{\text{sp}}}{2} + \frac{\sigma_{\text{sp}}^2}{2} + \sqrt{\gamma^2 d_{\text{sp}} + \left(\gamma^2 - \left(\frac{d_{\text{sp}}}{2} + \frac{\sigma_{\text{sp}}^2}{2}\right)\right)^2}\right) \quad (189)$$

$$\widetilde{\lambda}_2 = \frac{1}{8}\left(\gamma^2 + \frac{d_{\text{sp}}}{2} + \frac{\sigma_{\text{sp}}^2}{2} - \sqrt{\gamma^2 d_{\text{sp}} + \left(\gamma^2 - \left(\frac{d_{\text{sp}}}{2} + \frac{\sigma_{\text{sp}}^2}{2}\right)\right)^2}\right) \quad (190)$$

$$\tan(\alpha) = \frac{1}{\gamma\sqrt{d_{\text{sp}}}}\left(\frac{d_{\text{sp}}}{2} + \frac{2\sigma_{\text{sp}}^2}{3} + \frac{1}{6} - \left(\gamma^2\left(1 + \frac{1}{3d_{\text{in}}}\right) + \frac{\sigma_{\text{in}}^2}{3}\left(1 - \frac{1}{d_{\text{in}}}\right)\right)\right.$$
$$\left. + \sqrt{\gamma^2 d_{\text{sp}} + \left(\left(\gamma^2\left(1 + \frac{1}{3d_{\text{in}}}\right) + \frac{\sigma_{\text{in}}^2}{3}\left(1 - \frac{1}{d_{\text{in}}}\right)\right) - \left(\frac{d_{\text{sp}}}{2} + \frac{2\sigma_{\text{sp}}^2}{3} + \frac{1}{6}\right)\right)^2}\right) \quad (191)$$

$$\tan(\beta) = \frac{1}{\gamma\sqrt{d_{\text{sp}}}}\left(\frac{d_{\text{sp}}}{2} + \frac{\sigma_{\text{sp}}^2}{2} - \gamma^2 + \sqrt{\gamma^2 d_{\text{sp}} + \left(\gamma^2 - \left(\frac{d_{\text{sp}}}{2} + \frac{\sigma_{\text{sp}}^2}{2}\right)\right)^2}\right) \quad (192)$$

Consider the subclass of problem parameters, $d_{\text{sp}} = z, \gamma = K_1/\sqrt{z}$ and $\sigma_{\text{sp}} = K_2\sqrt{z}$ for fixed constants $K_1, K_2 > 0$ and some variable $z > 0$, which we can vary to give us different problem instances for our toy model in (6).

$$\lambda_1 = \frac{1}{8}\left(\frac{K_1^2}{z}\left(1+\frac{1}{3d_{\text{in}}}\right) + \frac{\sigma_{\text{in}}^2}{3}\left(1-\frac{1}{d_{\text{in}}}\right) + \frac{z}{2} + \frac{2K_2^2 z}{3} + \frac{1}{6}\right.$$

$$\left.+ \sqrt{K_1^2 + \left(\left(\frac{K_1^2}{z}\left(1+\frac{1}{3d_{\text{in}}}\right) + \frac{\sigma_{\text{in}}^2}{3}\left(1-\frac{1}{d_{\text{in}}}\right)\right) - \left(\frac{z}{2} + \frac{2K_2^2 z}{3} + \frac{1}{6}\right)\right)^2}\right) \quad (193)$$

$$\lambda_2 = \frac{1}{8}\left(\frac{K_1^2}{z}\left(1+\frac{1}{3d_{\text{in}}}\right) + \frac{\sigma_{\text{in}}^2}{3}\left(1-\frac{1}{d_{\text{in}}}\right) + \frac{z}{2} + \frac{2K_2^2 z}{3} + \frac{1}{6}\right.$$

$$\left.- \sqrt{K_1^2 + \left(\left(\frac{K_1^2}{z}\left(1+\frac{1}{3d_{\text{in}}}\right) + \frac{\sigma_{\text{in}}^2}{3}\left(1-\frac{1}{d_{\text{in}}}\right)\right) - \left(\frac{z}{2} + \frac{2K_2^2 z}{3} + \frac{1}{6}\right)\right)^2}\right) \quad (194)$$

$$\widetilde{\lambda}_1 = \frac{1}{8}\left(\frac{K_1^2}{z} + \frac{z}{2} + \frac{K_2^2 z}{2} + \sqrt{K_1^2 + \left(\frac{K_1^2}{z} - \left(\frac{z}{2} + \frac{K_2^2 z}{2}\right)\right)^2}\right) \quad (195)$$

$$\widetilde{\lambda}_2 = \frac{1}{8}\left(\frac{K_1^2}{z} + \frac{z}{2} + \frac{K_2^2 z}{2} - \sqrt{K_1^2 + \left(\frac{K_1^2}{z} - \left(\frac{z}{2} + \frac{K_2^2 z}{2}\right)\right)^2}\right) \quad (196)$$

$$\tan(\alpha) = \frac{1}{K_1}\left(\frac{z}{2} + \frac{2K_2^2 z}{3} + \frac{1}{6} - \left(\frac{K_1^2}{z}\left(1+\frac{1}{3d_{\text{in}}}\right) + \frac{\sigma_{\text{in}}^2}{3}\left(1-\frac{1}{d_{\text{in}}}\right)\right)\right.$$

$$\left.+ \sqrt{K_1^2 + \left(\frac{K_1^2}{z}\left(1+\frac{1}{3d_{\text{in}}}\right) + \frac{\sigma_{\text{in}}^2}{3}\left(1-\frac{1}{d_{\text{in}}}\right) - \left(\frac{z}{2} + \frac{2K_2^2 z}{3} + \frac{1}{6}\right)\right)^2}\right) \quad (197)$$

$$\tan(\beta) = \frac{1}{K_1}\left(\frac{z}{2} + \frac{K_2^2 z}{2} - \frac{K_1^2}{z} + \sqrt{K_1^2 + \left(\frac{K_1^2}{z} - \left(\frac{z}{2} + \frac{K_2^2 z}{2}\right)\right)^2}\right) \quad (198)$$

From Stewart [80], we can use the closed form expression for the singular vectors of a $2 \times 2$ full rank asymmetric matrix $\begin{bmatrix} a, & b \\ c, & d \end{bmatrix}$. The singular vectors are given by

$$\begin{bmatrix} \cos\theta, & \sin\theta \\ \sin\theta, & -\cos\theta \end{bmatrix},$$

where, $\tan(2\theta)$ is given by:

$$\tan(2\theta) = \frac{2ac + 2bd}{a^2 + b^2 - c^2 - d^2}.$$

Now, substituting the values in the expression from (100), we get singular vectors of the above form where $\theta \in [0, \pi/2]$ satisfies:

$$\theta = \frac{1}{2}\tan^{-1}\left(\frac{2\tan(\beta - \alpha); \cdot (\widetilde{\lambda}_1 - \widetilde{\lambda}_2) \cdot \sqrt{\lambda_1 \lambda_2}}{(\lambda_2 \widetilde{\lambda}_1 - \lambda_1 \widetilde{\lambda}_2) - (\lambda_1 \widetilde{\lambda}_1 - \lambda_2 \widetilde{\lambda}_2) \cdot \tan^2(\alpha - \beta)}\right) \quad (199)$$

$\square$

**Lemma 25** (asymptotic behavior of $\tau \tan\theta$). *For $\gamma = K_1/\sqrt{z}$, $\sigma_{\text{sp}} = K_2\sqrt{z}$,*

$$\lim_{z \to \infty} \tau \tan\theta = \frac{K_1 K_2^2}{(1 + K_2^2)2\sigma_{\text{in}}^2(1 - 1/d_{\text{in}})}$$

*Proof.* In order to determine the asymptotic nature of $\tan(\theta)$ as $z \to \infty$, we take the limit of a slightly different term first, since we have the closed form expression of $\tan(2\theta)$.

$$\lim_{z\to\infty} \tau \tan(2\theta) = \sqrt{\frac{\lambda_1}{\lambda_2}} \cdot \frac{2\tan(\alpha-\beta)\cdot(\tilde{\lambda}_1/\widetilde{\lambda}_2 - 1)}{(\tilde{\lambda}_1/\widetilde{\lambda}_2 - \lambda_1/\lambda_2) - (\lambda_1\tilde{\lambda}_1/\lambda_2\tilde{\lambda}_2 - 1)\cdot\tan^2(\alpha-\beta)}$$

$$= 2\tan(\alpha-\beta)\cdot\frac{\widetilde{\lambda}_1/\widetilde{\lambda}_2 - 1}{\widetilde{\lambda}_1/\widetilde{\lambda}_2 \cdot \lambda_2/\lambda_1 - 1},$$

since it is easy to see that $\lim_{z\to\infty}\tan^2(\alpha-\beta)\cdot\left(\frac{\lambda_1\widetilde{\lambda}_1}{\lambda_2\widetilde{\lambda}_2} - 1\right) = 0$.

If we use $\tan(\alpha-\beta) = \frac{\tan\alpha - \tan\beta}{1+\tan\alpha\tan\beta}$, and substitute the functions of $z$, for all the quantities in the above expression using Lemma 24, we derive: $\lim_{z\to\infty}\tau\tan 2\theta = 2K_1K_2^2/(1+K_2^2)2\sigma_{\text{in}}^2(1-1/d_{\text{in}})$.

Since $\tau \to \infty$, $\tan(2\theta) \to 0$, and further from Taylor approximation of $\tan(2\theta)$, $\tan(2\theta) \to 2\theta$. We can use this to derive the limit for $\tau\tan\theta$, which would just be $1/2 \cdot 2K_1K_2^2/(1+K_2^2)2\sigma_{\text{in}}^2(1-1/d_{\text{in}}) = K_1K_2^2/(1+K_2^2)2\sigma_{\text{in}}^2(1-1/d_{\text{in}})$.

$\square$

**Lemma 26** (asymptotic behaviors of $\cot\alpha, \tan\theta$). *For $\gamma = K_1/\sqrt{z}$, $\sigma_{\text{sp}} = K_2\sqrt{z}$ following the expressions in Lemma 24,*

$$\lim_{z\to\infty}\cot\alpha = 0, \qquad \lim_{z\to\infty}\tan\theta = 0.$$

*Proof.* For $\tan\theta$, since $\tau \to \infty$, and $\tau\tan\theta$ approaches a constant (from Lemma 25), we conclude $\lim_{z\to\infty}\tan\theta = 0$. For $\cot\alpha$,

$$\lim_{z\to\infty}\frac{z}{2} + \frac{2K_2^2 z}{3} + \frac{1}{6} - \left(\frac{K_1^2}{z}\left(1+\frac{1}{3d_{\text{in}}}\right) + \frac{\sigma_{\text{in}}^2}{3}\left(1-\frac{1}{d_{\text{in}}}\right)\right) = \infty,$$

and,

$$\lim_{z\to\infty}\sqrt{K_1^2 + \left(\frac{K_1^2}{z}\left(1+\frac{1}{3d_{\text{in}}}\right) + \frac{\sigma_{\text{in}}^2}{3}\left(1-\frac{1}{d_{\text{in}}}\right) - \left(\frac{z}{2} + \frac{2K_2^2 z}{3} + \frac{1}{6}\right)\right)^2} = \infty.$$

Thus, $\cot\alpha \to 0$. $\square$

**Lemma 27** (asymptotic behavior of $z\cot\alpha$). *For $\gamma = K_1/\sqrt{z}$, $\sigma_{\text{sp}} = K_2\sqrt{z}$ following the expressions in Lemma 24,*

$$\lim_{z\to\infty} z\cot\alpha = \frac{K_1}{1 + 4/3K_2^2}.$$

*Proof.* The expression for $z\cot\alpha$ or $z/\tan\alpha$ follows from Lemma 24:

$$\lim_{z\to\infty} z\cot\alpha = \frac{zK_1}{p + \sqrt{p^2 + K_1^2}},$$

where $p = \frac{K_1^2}{z}\left(1+\frac{1}{3d_{\text{in}}}\right) + \frac{\sigma_{\text{in}}^2}{3}\left(1-\frac{1}{d_{\text{in}}}\right) - \left(\frac{z}{2} + \frac{2K_2^2 z}{3} + \frac{1}{6}\right)$. Applying L'Hôpital's (relevant expressions are continuous in $z$) rule we get: $\lim_{z\to\infty} z\cot\alpha = \frac{K_1}{1+4/3K_2^2}$.

$\square$

**Lemma 28** (asymptotic behavior of $z/\tau^2$). *For $\gamma = K_1/\sqrt{z}$, $\sigma_{\text{sp}} = K_2\sqrt{z}$ following the expressions in Lemma 24,*

$$\lim_{z\to\infty} z/\tau^2 = \frac{2\sigma_{\text{in}}^2/3(1 - 1/d_{\text{in}})}{1 + 4/3K_2^2}.$$

*Proof.* For $\tau = \lambda_1/\lambda_2$, substituting the relevant expressions from Lemma 24, we get:

$$z/\tau^2 = \frac{z\lambda_2}{\lambda_1}$$

$$= z \cdot \frac{2K_1^2/z\left(1+1/3d_{\text{in}}\right) + 2\sigma_{\text{in}}^2\left(1-1/d_{\text{in}}\right) + p - \sqrt{K_1^2 + p^2}}{2K_1^2/z\left(1+1/3d_{\text{in}}\right) + 2\sigma_{\text{in}}^2\left(1-1/d_{\text{in}}\right) + p + \sqrt{K_1^2 + p^2}},$$

where $p = z/2 + 2K_2^2 z/3 + 1/6$. Applying L'Hôpital's (relevant expressions are continuous in $z$) rule we get: $\lim_{z\to\infty} z/\tau^2 = \frac{2\sigma_{\text{in}}^2/3(1-1/d_{\text{in}})}{1+4/3K_2^2}$. $\qquad\square$

**Lemma 29** (0-1 error of a classifier on target). *Assume a classifier of the form $w = l_1 \cdot w_{\text{in}} + l_2 \cdot w_{\text{sp}}$ where $l_1, l_2 \in \mathbb{R}$ and $w_{\text{in}} = [w^\star, 0, ..., 0]^\top$, and $w_{\text{sp}} = [0, ..., 0, 1_{d_{\text{sp}}}/\sqrt{d_{\text{sp}}}]^\top$. Then the target accuracy of this classifier is given by $0.5 \cdot \text{erfc}\left(-\frac{l_1 \cdot \gamma}{\sqrt{2} \cdot l_2 \cdot \sigma_{\text{sp}}}\right)$.*

*Proof.* Assume $(x, y) \sim P_{\mathsf{T}}$. Accuracy of $w$ is given by $\mathbb{E}_{P_{\mathsf{T}}}\left[(\text{sign}\left(w^\top x\right) = y)\right]$.

$$
\begin{aligned}
\mathbb{E}_{P_{\mathsf{T}}}\left[\text{sign}\left(w^\top x\right) = y\right] &= \mathbb{E}_{P_{\mathsf{T}}}\left[y \cdot \text{sign}\left(w^\top x\right) = 1\right] \\
&= \mathbb{E}_{P_{\mathsf{T}}}\left[y \cdot (w^\top x) > 0\right] \\
&= \mathbb{E}_{P_{\mathsf{T}}}\left[y \cdot (x^\top (l_1 \cdot w_{\text{in}} + l_2 \cdot w_{\text{sp}})) > 0\right] \\
&= \mathbb{E}_{P_{\mathsf{T}}}\left[y \cdot (\gamma \cdot l_1 \cdot y + l_2 \cdot \sigma_{\text{sp}}) > 0\right] \\
&= \mathbb{E}_{z\sim\mathcal{N}(0,1)}\left[(\gamma \cdot l_1 + y \cdot l_2 \cdot \sigma_{\text{sp}} \cdot z) > 0\right] \\
&= \mathbb{E}_{z\sim\mathcal{N}(0,1)}\left[y \cdot l_2 \cdot \sigma_{\text{sp}} \cdot z > -\gamma \cdot l_1\right] \\
&= \mathbb{E}_{z\sim\mathcal{N}(0,1)}\left[l_2 \cdot \sigma_{\text{sp}} \cdot z > -\gamma \cdot l_1\right] \\
&= \mathbb{E}_{z\sim\mathcal{N}(0,1)}\left[z > -\frac{\gamma \cdot l_1}{l_2 \cdot \sigma_{\text{sp}}}\right]
\end{aligned}
$$

Using the definition of $\text{erfc}$ function, we get the aforementioned accuracy expression. $\qquad\square$

**Lemma 30.** *For $\sigma > 0$ and $\mu \in \mathbb{R}$, we have*

$$
g(\mu, \sigma) := \mathbb{E}_{z\sim\mathcal{N}(0,\sigma)}\left[\exp\left(-|\mu + z|\right)\right] \tag{200}
$$
$$
= \frac{1}{2}\left(\exp\left(\sigma^2/2 - \mu\right) \cdot \text{erfc}\left(-\mu/\sqrt{2}\sigma + \sigma/\sqrt{2}\right) + \exp\left(\sigma^2/2 + \mu\right) \cdot \text{erfc}\left(\mu/\sqrt{2}\sigma + \sigma/\sqrt{2}\right)\right) \tag{201}
$$

*Proof.* The proof uses simple algebra and the definition of $\text{erfc}$ function.

$$
\begin{aligned}
g(\mu, \sigma) := \ &\mathbb{E}_{z\sim\mathcal{N}(0,\sigma)}\left[\exp\left(-|\mu + z|\right)\right] \\
= \ &\frac{1}{\sqrt{2\pi}}\int_z \exp\left(-|\mu + z|\right) \cdot \exp\left(-\frac{z^2}{2\sigma^2}\right) dz \\
= \ &\frac{1}{\sqrt{2\pi}}\int_{-\infty}^{\infty} \exp\left(-|\mu + z|\right) \cdot \exp\left(-\frac{z^2}{2\sigma^2}\right) dz \\
= \ &\frac{1}{\sqrt{2\pi}}\int_{-\mu}^{\infty} \exp\left(-\mu + z\right) \cdot \exp\left(-\frac{z^2}{2\sigma^2}\right) dz + \frac{1}{\sqrt{2\pi}}\int_{-\infty}^{-\mu} \exp\left(\mu + z\right) \cdot \exp\left(-\frac{z^2}{2\sigma^2}\right) dz \\
= \ &\exp\left(\sigma^2/2 - \mu\right)\int_{\frac{-\mu}{\sqrt{2}\sigma}+\frac{\sqrt{2}\sigma}{2}}^{\infty} \exp(-z^2)dz + \exp\left(\sigma^2/2 + \mu\right)\int_{-\infty}^{\frac{-\mu}{\sqrt{2}\sigma}-\frac{\sqrt{2}\sigma}{2}} \exp(-z^2)dz \\
= \ &\frac{1}{2}\left(\exp\left(\sigma^2/2 - \mu\right) \cdot \text{erfc}\left(-\mu/\sqrt{2}\sigma + \sigma/\sqrt{2}\right) + \exp\left(\sigma^2/2 + \mu\right) \cdot \text{erfc}\left(\mu/\sqrt{2}\sigma + \sigma/\sqrt{2}\right)\right)
\end{aligned}
$$

$\qquad\square$

