# OpenReview forum: "Complementary Benefits of Contrastive Learning and Self-Training Under Distribution Shift"
_NeurIPS.cc/2023/Conference — NeurIPS 2023 poster_

### Official Review · Reviewer_C2y1 · 2023-07-05

**Soundness:** 2 fair
**Presentation:** 3 good
**Contribution:** 3 good
**Rating:** 6
**Confidence:** 3

**Summary:**

This paper proposes to combine contrastive learning and self-training for unsupervised domain adaptation. Experimental results on UDA benchmarks demonstrate the empirical effectiveness of the approach and a thorough study demonstrates the theoretical benefits.

**Strengths:**

- The results on UDA tasks are consistently better than simple contrastive learning and self-training approaches.

- The theoretical analysis is very thorough and demonstrates the fundamental benefits of contrastive learning and self-training individually and of combining them in the case of UDA.

**Weaknesses:**

- The proposed method STOC is vaguely described in L104-108. It would be clearer to describe it formally in more details.

- The contributions are not clearly stated. Among the methods presented in L90-108, please clarify that STOC is your contribution.

- Some popular UDA benchmarks are missing, many concurrent work are using Office-31 and Office-Home. Evaluation on these datasets would be appreciated.

- Concurrent work are missing, such as [1, 2, 3]. Please position your work against these paper, and compare your approach both conceptually and empirically to the approaches presented in these papers. More generally, it seems that the presented method is only compared to simple baselines and not to methods that are specifically designed to tackle UDA tasks.

[1] DeepJDOT: Deep Joint Distribution Optimal Transport for Unsupervised Domain Adaptation, Damodaran et al., ECCV 2018

[2] Implicit Class-Conditioned Domain Alignment for Unsupervised Domain Adaptation, Jiang et al., ICML 2020

[3] Semantic-aware Message Broadcasting for Efficient Unsupervised Domain Adaptation, Li et al., ArXiv: 2212.02739

**Questions:**

- In Table 1, are concurrent models your reimplementation, or pretrained models from the SwAV and FixMatch papers ?

**Limitations:**

Some limitations are discussed in the analysis, for exemple regarding the limited performance improvement on semi-supervised tasks.

---

> ### Author Rebuttal · Authors · 2023-08-10
>
> We thank the reviewer for their positive assessment of our work and for their thoughtful feedback. We will improve the exposition of the final version as per your suggestions (e.g., a bulleted list of contributions, algorithmic description of STOC).
>
>
> > **The proposed method STOC is vaguely described in L104-108. It would be clearer to describe it formally in more details.**
>
> We have updated the draft to include an Algorithmic description of STOC.
>
> > **The contributions are not clearly stated. Among the methods presented in L90-108, please clarify that STOC is your contribution.**
>
> Thanks for your suggestion; we have updated the draft to list our main contributions:
> We propose Self-Training Over Contrastive learning (STOC) to explore benefits of combining self-training and contrastive learning.
> - Our experiments across eight benchmark datasets highlight that: (i) in domain adaptation settings, self-training and contrastive learning offer significant complementary gains; and (ii) in semi-supervised learning settings, surprisingly, the benefits are not synergistic.
> - We theoretically analyze these techniques to understand why self-training improves over contrastive learning under distribution shift in a simplified model of distribution shift.
>
> Overall, our findings demonstrate that under distribution shift features produced by contrastive learning can yield a good initialization for self-training to further amplify gains and achieve improved performance, even when either method alone would yield comparatively worse performance.
>
> > **Some popular UDA benchmarks are missing, many concurrent work are using Office-31 and Office-Home. Evaluation on these datasets would be appreciated.**
>
> We would like to clarify that we indeed perform experiments on Office-Home and results are included in Table 1 and Table 2 (columns abbreviated with header OH). We do not include results on Office-31 because (i) the task is similar to Officehome; and (ii) Office-31 dataset is much smaller in size than OfficeHome.
>
> > **Concurrent work are missing, such as [1, 2, 3]. Please position your work against these paper ... More generally ... only compared to simple baselines and not to... specifically designed to tackle UDA tasks.**
>
> We thank the reviewer for pointers to the related work. We will update the draft to include discussion on these papers in our related work.
>
> Please note that the main focus of our paper is to investigate when constrastive learning and self-training benefits are complementary and not to establish a new state-of-the-art for unsupervised domain adaptation. For this reason, we also do not use imagenet pretraining to avoid confounding our conclusions about contrastive pretraining. Fair comparisons to other domain adaptation algorithms would require repeating experiments with Imagenet pretrained models which we leave for future work. Nevertheless, recent large-scale studies [4,5,6] hint that self-training (e.g. FixMatch) and constrastive learning methods (e.g. SwAV) empirically outperform other existing approaches specifically proposed for domain adaptation (e.g., DANN, CDANN).
>
> Specifically, for [1, 2, 3] we will include the following discussion. Jiang et al. proposes an alternative of psuedolabling to handle label imbalances. In our study, we do not focus on simulating label proportion shifts and datasets included in our study mainly focus on shifts in the covariate distribution. Hence, the approach proposed by Jiang et al. will be similar to pseudolabeling in absence of label proportion shifts, which we compare with in our additional experiments (see general response, where our trends continue to hold). Damodaran et al. proposed DeepJDOT which uses optimal transport loss to align the features on target data with features on source data, and as part of future work it would be interesting to see if distribution matching methods also benefit from contrastive learning. Li et al. proposed a message passing variant to perform domain adaptation specifically for vision transformer architectures (i.e., ViT). In our work, we perform experiments with ResNet architectures and it remains unclear how to extend Li et al. to our setting.
>
>
> > **In Table 1, are concurrent models your reimplementation, or pretrained models from the SwAV and FixMatch papers ?**
>
> All models included in our paper are trained again in our experimental setup with (adaptation) of original code from SwAV and FixMatch papers.
>
> [4] Sagawa, Shiori, et al. "Extending the WILDS benchmark for unsupervised adaptation." arXiv preprint arXiv:2112.05090 (2021).
> [5] Garg, Saurabh, et al. "Rlsbench: Domain adaptation under relaxed label shift." International Conference on Machine Learning. PMLR, 2023.
> [6] Shen, Kendrick, et al. "Connect, not collapse: Explaining contrastive learning for unsupervised domain adaptation." International Conference on Machine Learning. PMLR, 2022.

---

> > ### Comment · Reviewer_C2y1 · 2023-08-16
> >
> > Thank you for your response. I have read the rebuttal and my concerns have been addressed. I am increasing my score to weak accept.

---

### Official Review · Reviewer_Srqn · 2023-07-07

**Soundness:** 3 good
**Presentation:** 3 good
**Contribution:** 3 good
**Rating:** 6
**Confidence:** 3

**Summary:**

The paper explores the synergy of self-training and contrastive learning in semi-supervised learning (SSL) and unsupervised domain adaptation (UDA). It discovers their complementary effect in UDA. Furthermore, it proposes Self-Training Over Contrastive learning (STOC) to combine the benefits of the two approach. The STOC algorithm pretrains the self-training model by contrastive learning on source and target data.Finally, it theoretically analyzes the outcome of initializing with contrastive learning under a specific distribution shift.

**Strengths:**

- The paper provides good theoretical support and empirical evidence to defend their hypothesis. It provides valuable insights into the factors that contribute to the success of contrastive learning and self-training.
- The paper proposes a novel method that can improve unsupervised domain adaptation.
- The findings of this paper have important implications for incorporating unlabeled data.
- The paper is well-written.

**Weaknesses:**

- The combination of semi-supervised learning with self-supervised learning has been proposed in prior work [1], which is not included in the related work. An updated variety of this method can constitute as a relevant benchmark.
- The assumptions about infinite unlabeled data, linear classifier $h$, and scaling the magnitude of the coordinates for augmentation, are restrictive.
- The benchmarks are not competitive. There are several self-supervised methods such as DINO that can supplement the benchmarks.
- There are typos such as in line 40: 'the strong strong results'.

[1] Zhai, Xiaohua, Avital Oliver, Alexander Kolesnikov, and Lucas Beyer. "S4l: Self-supervised semi-supervised learning." In Proceedings of the IEEE/CVF international conference on computer vision, pp. 1476-1485. 2019.

**Questions:**

- The role of the classification head architecture and deep fine-tuning is missing from the experiments.

**Limitations:**

The paper does not discuss the limitations and societal impacts of this work.

---

> ### Author Rebuttal · Authors · 2023-08-10
>
> We thank the reviewer for their detailed comments and positive feedback. In the next revision, we will **add discussion on Zhai et al. [1]** and correct typos. We hope our responses address any outstanding concerns. Please let us know if there are any additional questions.
>
> > **The combination of semi-supervised learning with self-supervised learning has been proposed in prior work [1], which is not included in the related work.**
>
> Zhai et al. [1] differs from our work significantly in two ways: (1) they only focus on the semi-supervised learning (SSL) setting where there is no distribution shift between the input distribution of the unlabeled and labeled sets, while we focus more on the distribution shift setting (DA), and further contrast our findings in SSL and DA settings; and (2) they use rotation prediction as their pre-training objective while we analyze contrastive learning objectives. We will include this discussion in our related work section (currently in Appendix B).
>
> > **The assumptions about infinite unlabeled data, linear classifier , and scaling the magnitude of the coordinates for augmentation, are restrictive.**
>
> Since the aim of our study is to mainly explain the effects of distribution shift between labeled source and unlabeled target distributions, we avoid complicating the analysis stemming from the finite sample nature of the data. Further the main aim of our theoretical analysis is to explain our empirical findings (complementary nature of CL and ST) in a specific setup for UDA and SSL, as opposed to describing generic conditions for CL+ST to outperform CL/ST. Our choice of a linear classifier is common in theoretical analysis on contrastive learning and self-training [2,3,4], and our augmentation distribution has also been studied in prior work [3]. We note that our augmentations assume no knowledge of spurious features, and are similar in principle to augmentations like cropping or blurring that effectively reduce the magnitude of image features (see L191).
>
> > **The benchmarks are not competitive. There are several self-supervised methods such as DINO that can supplement the benchmarks.**
>
> In DINO the self-supervised learning objective used is derived from knowledge distillation and is very different from the contrastive pretraining objective we study. Therefore, we do not include experiments with DINO. Extending our study from CL to other pretraining techniques like DINO  is an interesting direction of further research. For our study on studying CL/ST and their combinations we include results on 8 datasets, each with two settings (SSL/UDA), and four methods (ERM, CL, ST, CL+ST). During the rebuttal period, we also ran experiments with additional combinations of CL+ST methods (beyond SWaV+FixMatch) that we will append to the revised paper. Please see the general response for these results.
>
> > **The role of the classification head architecture and deep fine-tuning is missing from the experiments.**
>
> We are not sure if we understand this question completely. If the question is about ''classification head architecture'', then we default to using a linear layer head for the classification, as is common practice. In Section 4.5 and Appendix E.4 we provide experimental results for full finetuning for our simulations in Sec 4, and for real world experiments we always finetune the full backbone. Please let us know if we misunderstood your question in which case we would be happy to provide further clarifications.
>
> > **The paper does not discuss the limitations and societal impacts of this work.**
>
> We discuss limitations of our work in Appendix A. For the final version, we will move this discussion to the main paper.
>
>
>
> [2] Shen, Kendrick, et al. "Connect, not collapse: Explaining contrastive learning for unsupervised domain adaptation." International Conference on Machine Learning. PMLR, 2022.
> [3] Saunshi, Nikunj, et al. "Understanding contrastive learning requires incorporating inductive biases." International Conference on Machine Learning. PMLR, 2022.
> [4] Chen, Yining, et al. "Self-training avoids using spurious features under domain shift." Advances in Neural Information Processing Systems 33 (2020): 21061-21071.

---

> > ### Comment · Reviewer_Srqn · 2023-08-17
> > **Post Rebuttal Response**
> >
> > Thank you for your response. Overall, the paper seems beneficial to be presented to the community.

---

### Official Review · Reviewer_fFuY · 2023-07-09

**Soundness:** 3 good
**Presentation:** 3 good
**Contribution:** 3 good
**Rating:** 7
**Confidence:** 4

**Summary:**

This paper empirically explores and theoretically studies the complementary benefits of using self-training (ST) and contrastive learning (CL) for unsupervised domain adaptation (UDA), where unlabeled data is available from source and target domains, whereas labels are only available from the source domain. Firstly, experiments on many distribution shift datasets show that combining self-training over contrastive learning (STOC) yields significant benefits for target domain accuracy (~5% on average) over just doing CL or ST. However the benefit in-distribution is shown to be much lower (<1%).

Motivated by these findings, the paper delves into the theoretical study of this phenomenon through a toy Gaussian data model, where each example is composed of domain-invariant (useful for source and target) and spurious features (only useful for source). Experiments on the toy model are shown to follow similar trends to real data. Through analysis on this toy model, the paper provides the following intuitions:

(a) ST can learn a good target classifier if it starts from a good enough initial classifier

(b) ERM on source domain cannot provide a good initializer because it focusses too much on spurious features

(c) CL (with some "generic" augmentations) learns representations that focus more on domain-invariant features, thus providing better features for ST to improve. Although CL classifier itself is not sufficient for good target performance

This theoretical model provides many insights into the role of CL and ST. Finally the paper empirically verifies some of these claims through further probing experiments on real data.

**Strengths:**

- The paper studies an unexplored (to my knowledge) space of combining two popular ideas for domain adaptation. The idea is natural and seemingly effective

- Experiments on standard datasets are convincing enough on the efficacy of the proposed approach. Theoretical analysis on the toy example seems solid, and provides useful intuitions & insights into why the proposed approach could work well. Overall the quality of the technical work seems good. The insights about "CL provides better features while ST improves the head" is a clean contribution

- The paper is clearly written and easy to follow. Sufficient intuitions are provided for many of the theoretical claims.

I didn't read through all the proofs, but the intuitions made sense and the contrastive learning results seemed believable. On the whole, I think this is a solid contribution and vote for accepting.

**Weaknesses:**

- One minor criticism is about the presentation of Section 5. That one required multiple passes to follow, and it might help to include a clearer description of what is being tested. For instance in L377, it would help to explain how the 14% number was calculated. Some more questions about this in the next section

- A bit more discussion about prior work, in the main paper, would have been useful to see. Sections 5 and 6 do a bit of this in some parts, but I only partially understand how this analysis different from previous ones, especially for self-training

- The paper discusses contrastive learning and self-training in general, but only tests on one pair of methods (SwaV and FixMatch). Including more methods (even in 1 setting) would be useful.

**Questions:**

(Q1) How many labeled examples were used for experiments in Section 5? I'm wondering if the target probe results in Fig. 3 are due to worse sample complexity rather than expressivity of the features themselves

(Q2) What are the limitations of the proposed theoretical framework? Does the toy model fail in crucial ways to capture realistic distributions? Does the theory for this model predict something that does not hold in practice. A discussion about this would be useful for future work on this topic

(Q3) Does the choice of contrastive learning/self-training method affect the findings?

(Q4) Does linear probe (instead of fine-tuning everything) also work well in practice?

**Limitations:**

See Q2 above.

---

> ### Author Rebuttal · Authors · 2023-08-10
>
> We thank the reviewer for their detailed comments and positive feedback. In our revision, we will elaborate on the setup in Sec 5, and move the discussion on related work from App B to the main paper. We **add experiments with two additional combinations of CL+ST** algorithms (see general response) and hope that our responses address any outstanding concerns. Please let us know if there are any additional questions.
>
> > **In L377 … explain how the 14% number was calculated.**
>
> The 14% target probe performance difference is calculated based on the right half of Fig 3. The target linear probe performance of CL+ST (80.7%) is about 14% greater than that of ST (67.1%). The difference is higher when comparing CL with ERM. This leads us to believe that contrastive pretraining significantly raises the ceiling on target performance. We apologize for the confusion and will make this clearer in our final version.
>
> >  **More discussion about prior work, especially for self-training.**
>
> We deferred discussion on extended related works to App  B. Further, in App G.2 we specifically focus on distinguishing our analysis from prior self-training analyses that treat self-training as a consistency regularization objective. In principle, these works assume strong expansion assumptions on class conditional distributions, and ignore the challenges involved in propagating pseudolabels iteratively when the expansion assumptions are difficult to satisfy. If we get an extra page for the final version, we will move this discussion to the main paper.
>
> > **Only one pair of CL+ST algorithms. Does the choice of contrastive learning/self-training method affect the findings?**
>
> Please see the general response for our reply to this question. We have included results with BT as a CL algorithm and Pseudolabeling as self-training algorithm in the general response. This gives us two additional pairs for  CL + ST. Overall, we observe that results with these two new algorithms match the trends we observed with our default choices. This highlights that the complementary benefits of self-training and contrastive learning hold across different variations of each of them. We thank the reviewer for their suggestion.
>
> In theory, we analyze Barlow Twins which has been shown to be equivalent to spectral contrastive loss, and even other non-contrastive objectives (see L949 in App E.2 for more). Similarly, the self-training method we analyze in Sec 4 captures generic iterative pseudolabeling methods done in two stages.
>
>
> > **How many labeled examples were used for experiments in Section 5?**
>
> For each dataset the number of training examples for linear probes are of order $10^4$ (except for OfficeHome where the target dataset is of order 3000) obtained by 80:20 split of all the target data. The impact of finite sample nature of target samples is negligible because we only train a linear head over the representations, so the generalization error is very small $O(\sqrt{1/n})$ where $n$ is number of samples used to learn head.
>
>
> > **What are the limitations of the proposed theoretical framework? Does the toy model fail in crucial ways to capture realistic distributions?**
>
> Following are some limitations of our theoretical setup for analyzing the complementary nature of CL and ST, which we are actively exploring as directions of future work. We mention some of these in Appendix A.
>
> -  We only analyze the case of spurious correlations in source data, as opposed to more general shifts in the covariate distribution.
>
> - Our theory assumes general augmentations that scale each feature independently (similar to [2]) and doesn’t capture more specific conditions for the augmentations.
>
> - While empirically we see our trends hold with full finetuning (Sec 4.5, App E.4), we only analyze linear probing (as done in prior works [1, 3]).
>
>
> > **Does the theory for this model predict something that does not hold in practice**
>
> The only important trend that shows a deviation between theory and practice is the following: self-training performs as worse as ERM (Fig 2. and Thm. 2), whereas in practice self-training is not necessarily worse (Sec 3). This is mainly due to the normalization step in the self-training iteration (Eq. (2)), which is not done in practice. This normalization is done for mathematical convenience (similar to analysis [3]), so that we can control the $l_2$ norm of the self-training iterates. We thank the reviewer for bringing this up and will add this discussion to our theory section.
>
>
>
>
> > **Does linear probe (instead of fine-tuning everything) also work well in practice**
>
> We observe that training linear head with ST (source labeled + target unlabeled) improves performance over just training head with source labeled data. In fact, if we used the optimal stopping criterion in hindsight (using target labeled data), the benefits are almost as much as full finetuning. However, since we do not have target labeled data in practice, we use source hold data for early stopping and resort to full finetuning which usually yields slightly better performance than only training the linear head.
>
> In our theoretical setup, we observe that source early stopping works and yields optimal performance. Hence, to analyze CL and ST, we perform linear probing which is amenable to theoretical treatment (as in prior works [1, 2, 3]) .
>
> [1] Shen et al. Connect, not collapse: Explaining contrastive learning for unsupervised domain adaptation.
>
> [2] Saunshi et al. Understanding contrastive learning requires incorporating inductive biases
>
> [3] Chen et al. Self-training avoids using spurious features under domain shift.

---

> > ### Comment · Reviewer_fFuY · 2023-08-19
> > **Thank you for the response**
> >
> > Thanks for the response and clarifications. The new experiments with more contrastive and self-training methods would certainly be a useful additions. I maintain my positive opinion of this paper.

---

### Official Review · Reviewer_272G · 2023-07-29

**Soundness:** 3 good
**Presentation:** 3 good
**Contribution:** 3 good
**Rating:** 7
**Confidence:** 4

**Summary:**

This paper investigates the complementary benefits of combining self-training and contrastive pretraining for domain adaptation under distribution shifts. Through an empirical study on 8 benchmarks, the authors demonstrate that applying self-training (FixMatch) after contrastive pretraining (SwAV) yields substantial accuracy gains over either approach alone. To understand this synergy, the authors analyze a simplified theoretical setup indicating that while contrastive pretraining can amplify signal along invariant features, it may retain dependence on spurious source-only correlations. On the other hand, self-training can effectively unlearn these spurious dependencies and improve linear transfer of representations, achieving optimal target performance. The authors further apply linear probing on representations to verify their theoretical findings empirically. Overall, this work highlights the potential of combining self-training and contrastive learning to address challenges posed by distribution shifts, grounded by both empirical evidence and theoretical analysis.

**Strengths:**

**Originality**: The paper explores the promising yet underexplored direction of combining self-training and contrastive learning to address distribution shifts. The extensive empirical study systematically investigates their complementary benefits across diverse benchmarks. The theoretical analysis offers unique insights into their synergistic effects.

**Quality**: The empirical study is relatively extensive, spanning 8 datasets with careful controls. Despite simplifying assumptions, the theory makes falsifiable predictions that are precisely analyzed and align well with observations. Additional probing experiments further validate the theoretical intuitions.

**Clarity**: The paper is clearly presented overall. The problem setup and motivation are lucid, and the methods and experiments are thoroughly explained. The authors effectively convey the core conceptual messages and insights from their theoretical analysis.

**Significance**: Distribution shifts ubiquitously challenge ML model generalization in practice. This paper highlights the significant potential of combining self-training and contrastive learning to address this problem, demonstrating substantial empirical gains and proposing explanations grounded in both theory and experiments. Overall, I think this paper will be of interest to the ML research community of distribution shifts.

**Weaknesses:**

**Small Pre-training Set**: The authors have chosen to train models using contrastive learning (CL) from scratch on a selection of small datasets from BREEDS and WILDS. The authors have justified this approach by stating, "We have opted not to use off-the-shelf pretrained models (e.g., on Imagenet [67]) to avoid confounding our conclusions about contrastive pretraining." However, in practice, pretraining CL models on small datasets is not a common practice, given the availability of powerful off-the-shelf CL models that have been pretrained on larger datasets (e.g., from SimCLR, iBot to DINOv2). If the paper's results only apply to CL models pretrained on small datasets and do not generalize to more powerful contemporary CL models, the utility of this work could be limited. I would recommend that the authors conduct studies of modern off-the-shelf CL models to strengthen this work. For instance, off-the-shelf ImageNet-pretrained CL models could be used as initialization, followed by self-training, and then evaluation on various shifted datasets (numerous ImageNet shifted datasets are available). Please note, it is not necessary to strive for results during the rebuttal period. This paper is already commendable in my view, I merely want to offer suggestions that could further enhance its impact.

**Limited Combination of CL+ST Algorithms** In this study, the authors have exclusively utilized SwAV+FixMatch, which are indeed effective algorithms. However, the presentation would be more compelling if a wider range of combinations were explored. Given that the authors employ Barlow Twins in their theoretical analysis, would it not be more fitting to also include Barlow Twins as a CL algorithm in the empirical studies?

**Questions:**

See Weaknesses.

**Limitations:**

See Weaknesses.

---

> ### Author Rebuttal · Authors · 2023-08-10
>
> We thank the reviewer for their positive assessment of our work and their detailed and constructive feedback. For the rebuttal, we **added new results with Barlow Twins** as the pretraining algorithm and some **preliminary results with Imagenet pretrained networks**.  Please let us know if this addresses all outstanding concerns or if there are any additional questions.
>
> > **Small Pretraining Set**
>
> Thanks for your suggestion; we agree this would be an interesting direction of study that could further strengthen the results. We performed preliminary experiments where finetune Imagenet pretrained networks and performed experiments with FixMatch training (e.g., we replace contrastive pretraining with imagenet pretraining) and we observe that FixMatch continues to improve over ERM (source-only) models both pretrained with Imagenet.
>
> | Dataset      | ERM (Imagenet pretrained)   |FixMatch (Imagenet pretrained) |
> |--------------|-------|------------------|
> | OfficeHome (avg on 3 shifts)   | 47.1 | 57.4            |
> | Visda | 44.9 | 72.8           |
>
> Note, we did not perform these experiments with BREEDs datasets, because they are subsets of Imagenet and Imagenet-pretraining leaks information about target. We will plan to explore the impact of leveraging large off-the-shelf pretrained models, e.g. DINOv2 and CLIP for our revision.
>
>
> > **Results with Barlow Twins as a CL algorithm**
>
> We have included results with BT as a CL algorithm and Pseudolabeling as self-training algorithm in the general response. Overall, we observe that results with these two new algorithms match the trends we observed with our default choices. This highlights that the complementary benefits of self-training and contrastive learning hold across different variations of each of them. We thank the reviewer for their suggestion.

---

> > ### Comment · Reviewer_272G · 2023-08-13
> >
> > Thank you for your response. While the complementary ERM result is commendable, I believe there has been a misunderstanding. My intention was to suggest that you use Contrastive Learning (CL) models pre-trained on ImageNet instead of ERM models. For example, the official Barlow Twins model was pre-trained on ImageNet (https://github.com/facebookresearch/barlowtwins), and you can directly fine-tune it using FixMatch.

---

> > > ### Author Response · Authors · 2023-08-17
> > > **Updated results with Barlow Twins model pre-trained on Imagenet**
> > >
> > > We thank the reviewer for engaging in the discussion and apologize for confusing your suggestion. We re-ran experiments as you suggested, i.e., with the Barlow Twins model pre-trained on Imagenet on 4 datasets: Entity13, Nonliving26, Officehome, and Visda. We tabulate the results below. We continue to observe that FixMatch improves over ERM (source-only) models for Barlow Twins pre-trained models on Imagenet.
> > >
> > >
> > > |                           | ERM (Imagenet BT) | ST (Imagenet BT) |
> > > |---------------------------|-------------------|------------------|
> > > | Entity13                  | 81.0              | **85.4**             |
> > > | NonLiving26               | 62.3              | **69.7**            |
> > > | Visda (avg 2 shifts)      | 52.5              | **69.8**             |
> > > | Officehome (avg 3 shifts) | 46.4              | **49.3**             |
> > >
> > >
> > > We will include these results in the updated draft, and we believe these results strengthen our findings as we can hope to leverage off-the-shelf contrastively pre-trained models to combine the benefits of CL and ST.

---

> > > > ### Comment · Reviewer_272G · 2023-08-21
> > > >
> > > > Thank you for the new results. I quite appreciate this work, and I will keep my rating (clear accept).

---

### Author Rebuttal · Authors · 2023-08-10

We are grateful to the reviewers for their thoughtful feedback and are glad to see all of them recommending acceptance. Per their feedback we have **added experiments on more combinations of CL+ST** where we find our empirical findings on SWaV+FixMatch continue to hold. In the general response, we address one common concern shared by reviewers, and address others as individual responses to each reviewer. Please let us know if you have any additional concerns or if all outstanding concerns are addressed.


> **Limited combination of contrastive learning and self-training algorithms.**

In our experiments, we default to using the SWaV backbone for contrastive pretraining and FixMatch for self-training mainly because prior works [1, 2, 3] note that SWaV outperforms other contrastive pretraining methods like SimCLR, BYOL, etc. Having said that, for the rebuttal, we provide experimental results for two additional combinations: (1) SWaV + pseudolabeling (different self-training method), and (2) Barlow Twins + FixMatch (different contrastive pretraining method). Due to rebuttal time constraints we could only run on two datasets for each, but will add results on the remaining six in our revision.

| Dataset      | ERM   | ST (FixMatch) | CL (Barlow Twins) | STOC (Barlow Twins + FixMatch) |
|--------------|-------|---------------|-------------------|--------------------------------|
| Entity-13    | 68.32 | 77.93         | 81.04             | 86.23                          |
| Nonliving-26 | 45.54 | 56.79         | 62.17             | 71.46                          |


| Dataset      | ERM   | ST (Pseudolabel) | CL (SwAV)    | STOC (SwAV+Pseudolabel) |
|--------------|-------|------------------|-------|------------------------|
| Living-17    | 60.31 | 69.34            | 74.14 | 79.81                  |
| Nonliving-26 | 45.54 | 58.25            | 57.02 | 67.87                  |


Results with these two new combinations match the trends in Sec 3 (i.e., we observe that STOC improves over CL and ST alone), similar to what we observed with SWaV+FixMatch. In fact, on these datasets Barlow Twins pretraining improves over SwAV pretraining. This reinforces that the complementary benefits of self-training and contrastive learning hold across different variations of pretraining and self-training objectives. We thank the reviewers for their suggestion and will include these ablations on all datasets in our revision.

[1] Sagawa et al. "Extending the WILDS benchmark for unsupervised adaptation." arXiv preprint arXiv:2112.05090 (2021).
[2] Garg et al. "Rlsbench: Domain adaptation under relaxed label shift." International Conference on Machine Learning. PMLR, 2023.
[3] Shen et al. "Connect, not collapse: Explaining contrastive learning for unsupervised domain adaptation." International Conference on Machine Learning. PMLR, 2022.

---

### Decision · Program_Chairs · 2023-09-21

**Decision:**

Accept (poster)

**Comment:**

The paper proposed an approach to combine both self-training and contrastive pretraining for domain adaptation under distribution shifts. The main strength of the work lies in its entensive empirical studies, including consistent accuracy gains over 8 benchmark datasets. All the reviewers like the paper and recommend acceptance. I have read the rebuttal and the reviews. I agree with the reviewers that the rebuttal addressed most of the concerns. I recommend acceptance.